# A spatiotemporally separated framework for reconstructing the source of atmospheric radionuclide releases

Yuhan Xu[1], Sheng Fang[1,*], Xinwen Dong[1], Shuhan Zhuang[1]

[1]Institute of Nuclear and New Energy Technology, Collaborative Innovation Centre of Advanced Nuclear Energy Technology, Key Laboratory of Advanced Reactor Engineering and Safety of Ministry of Education, Tsinghua University, Beijing 100084, China

*Correspondence to*: Sheng Fang (fangsheng@tsinghua.edu.cn)

**Abstract.**   Determining the source location and release rate are critical tasks in assessing the environmental consequences of atmospheric radionuclide releases, but remain challenging because of the huge multi-dimensional solution space. We propose a spatiotemporally separated two-step framework that reduces the dimension of the solution space in each step and improves the source reconstruction accuracy. The separation process applies a temporal sliding-window average filter to the observations, thereby reducing the influence of temporal variations in the release rate on the observations and ensuring that the features of the filtered data are dominated by the source location. A machine learning model is trained to link these features to the source location, enabling independent source location estimations. The release rate is then determined using projected alternating minimization with the L1-norm and total variation regularization algorithm. This method is validated against the local-scale SCK-CEN $^{41}$Ar field experiment and the first release of the continental-scale European Tracer Experiment, for which the lowest source location errors are 4.52 m and 5.19 km, respectively. This presents higher accuracy and a smaller uncertainty range than the correlation-based and Bayesian methods in estimating the source location. The temporal variations in release rates are accurately reconstructed, and the mean relative errors of the total release are 65.09% and 72.14% lower than the Bayesian method for the SCK-CEN experiment and European Tracer Experiment, respectively. A sensitivity study demonstrates the robustness of the proposed method to different hyperparameters. With an appropriate site layout, low error levels can be achieved from only a single observation site or under meteorological errors.

## 1. Introduction

Atmospheric radionuclide release is a major environmental concern of the nuclear industry, including nuclear energy and its heat applications, isotope production, and the post-processing of radioactive waste. Such releases occurred after the Chernobyl nuclear accident (Anspaugh et al., 1988) and the Fukushima nuclear explosion (Katata et al., 2012), with partially known source information, i.e. the location. Recently, there have been several atmospheric radionuclide leaks from unknown sources, such as the 2017 $^{106}$Ru leakage (Masson et al., 2019) and the 2020 $^{134/137}$Cs detection in northern Europe (Ingremeau and Saunier, 2022), which have raised global concerns regarding the subsequent hazard to public health. Identification of source

information in these events is critical for the safe operation of nuclear facilities, consequence assessment, and emergency response.

During these events, source data often cannot be directly measured or determined because of the lack of information on the source of the leak. Instead, source information can only be reconstructed through inversion methods, which identify the optimal solution by comparing the environmental observations with atmospheric dispersion simulations using different estimates of the source location and release rate. Such reconstructions simultaneously identify the source location and release rate because the observations are intuitively determined by both parameters. In this case, the reconstruction searches for a solution over a large multi-dimensional space, where the dimension is the sum of the number of space coordinates and the length of the estimated release window. Therefore, the inversion is weakly constrained and can become ill-posed in the case of spatiotemporally limited observations and uncertainties in the atmospheric dispersion models. Unfortunately, this is quite often the case for atmospheric radionuclide releases.

To reduce the problem of ill-posedness, most previous studies have attempted to constrain the reconstruction by imposing assumptions on the model–observation discrepancies or release characteristics. Assumptions on model–observation discrepancies are widely used in Bayesian methods to simultaneously reconstruct the posterior distributions of spatiotemporal source parameters (De Meutter et al., 2021; Meutter and Hoffman, 2020; Xue et al., 2017a). This assumes that the model–observation discrepancies follow a certain statistical distribution (i.e. the likelihood of Bayesian methods), with the normal (Eslinger and Schrom, 2016; Guo et al., 2009; Keats et al., 2007, 2010; Rajaona et al., 2015; Xue et al., 2017a, b; Yee, 2017; Yee et al., 2008; Zhao et al., 2021) and log-normal (Chow et al., 2008; Dumont Le Brazidec et al., 2020; KIM et al., 2011; Monache et al., 2008; Saunier et al., 2019; Senocak, 2010; Senocak et al., 2008) distributions being two popular choices. Other candidates include the t-distribution (with degrees of freedom ranging from 3–10), Cauchy distribution, and log-Cauchy distribution, all of which were compared against the normal and log-normal distributions in terms of reconstructing the source parameters of the Prairie Grass field experiment (Wang et al., 2017). The results demonstrate that the likelihoods are sensitive to both the dataset and the target source parameters. Several studies have constructed the likelihood based on multiple metrics that measure the model–observation discrepancies in an attempt to better constrain the solution (Lucas et al., 2017; Jensen et al., 2019). More sophisticated methods involve the use of different statistical distributions for the likelihoods of non-detections and detections (De Meutter et al., 2021; Meutter and Hoffman, 2020). Recent studies have suggested the use of log-based distributions and tailored parameterization of the covariance matrix as a means of better quantifying the uncertainties in the reconstruction (Dumont Le Brazidec et al., 2021). These Bayesian methods have been applied to real atmospheric radionuclide releases, such as the 2017 [106]Ru event, and have provided important insights into the source and release process (Dumont Le Brazidec et al., 2020; Saunier et al., 2019; Dumont Le Brazidec et al., 2021; De Meutter et al., 2021). However, these studies have also revealed that the likelihood in Bayesian methods must be exquisitely designed and parameterized to achieve satisfactory spatiotemporal source reconstruction (Dumont Le Brazidec et al., 2021; Wang et al., 2017). With suboptimal design, the reconstruction may exhibit a bimodal posterior distribution (Meutter and Hoffman, 2020), which remains a challenge for robust applications in different scenarios.

Assumptions on the release characteristics aim to reduce the dimension of the solution space to 4 or 5, namely the two source location coordinates, the total release, and the release time (or the release start and end time), i.e. an instantaneous release at one time or constant release over a period (Kovalets et al., 2020, 2018; Efthimiou et al., 2018, 2017; Tomas et al., 2021; Andronopoulos and Kovalets, 2021; Ma et al., 2018). Under these assumptions, the correlation-based method exhibits high accuracy for ideal cases under stationary meteorological conditions, such as synthetic simulation experiments (Ma et al., 2018) and wind tunnel experiments (Kovalets et al., 2018; Efthimiou et al., 2017). However, previous studies have also demonstrated that real-world applications may be much more challenging, (Kovalets et al., 2020; Tomas et al., 2021; Andronopoulos and Kovalets, 2021; Becker et al., 2007) because the release usually exhibits temporal variations and may experience non-stationary meteorological fields. In addition, inaccurate calculation of the meteorological field input can further intensify these challenges. The interaction between the time-varying release characteristics and non-stationary meteorological fields is neglected in the instantaneous-release and constant-release assumptions, leading to inaccurate reconstruction.

Given the assumption-related reconstruction deviations in complex scenarios, we propose a spatiotemporally separated source reconstruction method that is less dependent on such assumptions. Our approach reduces the complexity of the source reconstruction using the simple fact that the source location is fixed during the atmospheric radionuclide release process. In this case, the spatiotemporal variations of observations are influenced by the time-varying release rate, source location, and meteorology, of which the last variable is generally known. The proposed method reduces the influence of the release rate through a temporal sliding-window average filter, making the filtered observations more sensitive to the source location than to the release rate. After filtering, existing methods based on direct observation–simulation comparisons may be unable to locate the source. Thus, the response features of the filtered observations are extracted and mapped to the source location by training a data-driven machine learning model using the extreme gradient boosting (XGBoost) algorithm (Chen and Guestrin, 2016). To fully capture the response features at each observation site, tailored time- and frequency-domain features are designed and optimized using the feature selection technique of XGBoost. Using this optimized model, the source location is estimated based on the filtered observations. Once the source location has been retrieved, the non-constant release rate is determined using the Projected Alternating MInimization with L1-norm and Total variation regularization (PAMILT) algorithm (Fang et al., 2022), which is robust to model uncertainties. The sequential spatiotemporal reconstruction reduces the dimension of the solution space at each step, which helps to improve the accuracy and reliability of the reconstruction.

The proposed method is validated using the data from multi-scales field experiments, namely the local-scale SCK-CEN [41]Ar experiment (Rojas-Palma et al., 2004), and the first release of the continental scale European Tracer Experiment (ETEX-1) (Nodop et al., 1998), which traced emissions of Perfluoro-Methyl-Cyclo-Hexane (PMCH). The performance of the proposed method is compared with the correlation-based method in terms of source location estimation and the Bayesian method in terms of spatiotemporal accuracy. The sensitivity of the source location estimation to the spatial search range, size of the sliding window, feature type, number and combination of sites, and meteorological errors is also investigated for the SCK-CEN [41]Ar experiment.

## 2. Materials and Methods

### 2.1 Source reconstruction models

For an atmospheric radionuclide release, Eq. (1) relates the observations at each observation site to the source parameters:

$$\boldsymbol{\mu} = \mathbf{F}(\mathbf{r}, \mathbf{q}) + \boldsymbol{\varepsilon}, \tag{1}$$

where $\boldsymbol{\mu} = [\mu_1, \mu_2, \cdots, \mu_N]^T \in \mathbb{R}^N$ is an observation vector composed of $N$ observations, the function $\mathbf{F}$ maps the source parameters to the observations, i.e. an atmospheric dispersion model, $\mathbf{r}$ refers to the source location, $\mathbf{q} \in \mathbb{R}^S$ is the temporally varying release rate, and $\boldsymbol{\varepsilon} \in \mathbb{R}^N$ is a vector containing both model and measurement errors.

In most source reconstruction models, $\mathbf{F}$ is simplified to the product of $\mathbf{q}$ and a source–receptor matrix $\mathbf{A}$ that depends on the source location:

$$\boldsymbol{\mu} = \mathbf{A}(\mathbf{r})\mathbf{q} + \boldsymbol{\varepsilon}, \tag{2}$$

where $\mathbf{A}(\mathbf{r}) = [A_1(\mathbf{r}), A_2(\mathbf{r}), \cdots, A_N(\mathbf{r})]^T \in \mathbb{R}^{N \times S}$ and each row describes the sensitivity of an observation to the release rate $\mathbf{q}$ given the source location $\mathbf{r}$.

### 2.2 Observation filtering for spatiotemporally separated reconstruction

A straightforward way to solve Eq. (2) is to simultaneously retrieve the source location and release rate; however, the solution space is huge and difficult to constrain. Several studies have noted that the source location can be retrieved separately without knowledge of the exact release rate, on the condition that the release rate is constant (Efthimiou et al., 2018; Kovalets et al., 2018; Efthimiou et al., 2017; Ma et al., 2018). The key reason is that, in constant-release cases, the relative spatiotemporal distribution of radionuclides is determined by the meteorological conditions and the relative positions between the source and receptors, and the constant release rate only changes the absolute values. Although the release rate may counteract the influence of the meteorological conditions and relative position at a single observation site, it cannot change the whole spatiotemporal distribution at multiple observation sites. Therefore, by analysing the spatiotemporal distribution of radionuclides at multiple observation sites, it is possible to locate the source without knowing the release rate under the constant-release assumption.

To provide a more general method, we take advantage of the fact that the source location has been fixed during all known atmospheric radionuclide releases, such as the Chernobyl nuclear accident (Anspaugh et al., 1988), Fukushima nuclear explosion (Katata et al., 2012), and 2017 [106]Ru leakage (Masson et al., 2019). With a fixed source location, the release rate and meteorology jointly determine the temporal variations of the observations (Li et al., 2019b). The influence of meteorology can be pre-calculated as the source–receptor sensitivities and subsequently stored in matrix $\mathbf{A}(\mathbf{r})$. By reducing the influence of the release rate, the constant-release case can be approximated and the sensitivity of the observations to the source location can be improved, enabling separate source location and release rate estimations and reducing the solution space at each step. For this purpose, we introduce an operator matrix $\mathbf{P} \in \mathbb{R}^{N \times N}$ to reduce the temporal variations of $\mathbf{A}(\mathbf{r})\mathbf{q}$:

$\boldsymbol{\mu}_p = \mathbf{P}\boldsymbol{\mu} = \mathbf{P}\mathbf{A}(\mathbf{r})\mathbf{q} + \mathbf{P}\boldsymbol{\varepsilon}$,      (3)
where $\boldsymbol{\mu}_p$ refers to the filtered observations. In this study, the following operator matrix is constructed to impose a one-sided
temporal sliding-window average filter (Eamonn Keogh, Selina Chu, 2004):
$$\mathbf{P} = \frac{1}{T}\begin{bmatrix} 1 & & & & & & & & \\ 1 & 1 & & & & & & & \\ \vdots & & \ddots & & & & & & \\ 1 & 1 & \cdots & 1 & & & & & \\ 1 & 1 & \cdots & 1 & 1 & & & & \\ & 1 & 1 & \cdots & 1 & 1 & & & \\ & & 1 & 1 & \cdots & 1 & 1 & & \\ & & & \ddots & \ddots & \ddots & \ddots & \ddots & \\ & & & & 1 & 1 & \cdots & 1 & 1 \\ & & & & 1 & 1 & 1 & 1 & 1 \end{bmatrix},$$
     (4)

where $T$ is the size of the sliding window. This one-sided filter involves the current and previous observations in the window,
acknowledging that future observations are not available for filtering in practice. Although a sliding-window average filter is
used in this study, Eq. (3) is compatible with more advanced processing methods.
**2.3 Source location estimation without knowing the exact release rates**
After applying the filter in Eq. (4), the peak observations, primarily shaped by the temporal release profile, are smoothed out.
However, the influences of the source position and meteorology remain relatively unchanged, as they determine the long-term
temporal trends of observations and are less affected by the filter. The meteorology is known, so it becomes possible to locate
the source using the filtered observations. Nevertheless, the specificity of source location estimation methods that rely on direct
observation–simulation comparisons may be substantially compromised because the peak amplitude is reduced. A better choice
for locating the source would be to use the response features of the filtered observations, which preserve most of the location
information. Therefore, it is necessary to establish a link between the response features of the filtered observations and the
source location. To achieve this, we train an XGBoost model that maps the response features of the filtered observations to the
coordinates of the source.
XGBoost is an optimized distributed gradient boosting library. Suppose $D = \{(\mathbf{X}_i, \mathbf{r}_i)\}(|D| = n, \mathbf{X}_i \in \mathbb{R}^p, \mathbf{r}_i \in \mathbb{R}^2)$, where
the number of samples is $n$ and each sample contains $p$ features. $\mathbf{X}_i$ is the given input feature vector of the $i$-th sample and
$\mathbf{r}_i = (x_i, y_i)$ is the location vector. XGBoost typically uses multiple decision trees (Fig. 1) to fit the target, which can be
formulated as:
$\hat{\mathbf{r}}_i = G(\mathbf{X}) = \sum_{k=1}^{K} f_k(\mathbf{X}_i), f_k \in \mathcal{F}$,      (5)
where $K$ is the number of trees, $\mathcal{F} = \{f(x) = \boldsymbol{\omega}_{Q(x)}\}(Q: \mathbb{R}^p \rightarrow M, \boldsymbol{\omega} \in \mathbb{R}^M)$ is the space of the decision trees, and $Q$
represents the structure of each tree, mapping the feature vector to $M$ leaf nodes. Each $f_k$ corresponds to an independent tree
structure $Q$ with leaf node weights $\boldsymbol{\omega} = (\omega_1, \omega_2, \cdots, \omega_M)$. Equation (5) is then used to predict $\hat{\mathbf{r}}_i = (\hat{x}_i, \hat{y}_i)$ for the $i$-th sample.

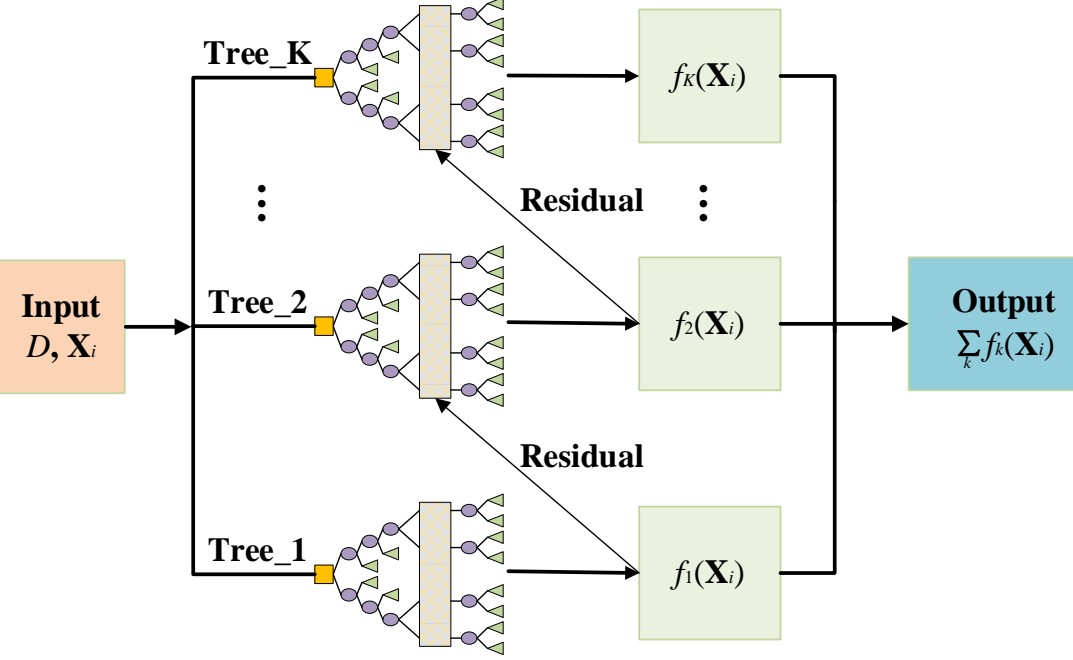


**Figure 1.** Flowchart of XGBoost for predicting $\hat{r}_i$ based on decision tree model. The yellow squares are the root nodes within each tree, representing the input features in this paper. The purple ellipses denote the child nodes where the model evaluates input features and make decisions to split the data. The green rectangles depict the leaf nodes and refer to the prediction results. The vertical rectangles abstract the internal splitting processes of the trees, indicating decision-making not explicitly detailed in the diagram.

XGBoost trains $G(\mathbf{X})$ in Eq. (5) by continuously fitting the residual error until the following objective function is minimized:
$$Obj^{(t)} = \sum_{i=1}^{n} \left( \mathbf{r}_i - \left( \hat{\mathbf{r}}_i^{(t-1)} + f_t(\mathbf{X}_i) \right) \right)^2 + \sum_{i=1}^{t} \Omega(f_i) \,, \tag{6}$$
where $t$ represents the training of the $t$-th tree and $\Omega(f_i)$ is the regularization term, given by:
$$\Omega(f) = YM + \frac{1}{2}\lambda \sum_{j=1}^{M} \omega_j^2 \,, \tag{7}$$
where $M$ is the number of leaf nodes, $\omega_j$ is the leaf node weight for the $j$-th leaf node, and $Y$, $\lambda$ are penalty coefficients. The
minimization of Eq. (6) provides a parametric model $G(\mathbf{X})$ that maps the feature ensemble $\mathbf{X}$ extracted from $\boldsymbol{\mu}_p$ to the source
location $\mathbf{r}$.
To comprehensively evaluate the influence of the source location, both time- and frequency-domain features (as outlined in
Table 1) are considered during the training process and mapped to the source location by $G(\mathbf{X})$. Among the time-domain
features, the wave rate quantifies the fluctuations of $\boldsymbol{\mu}_p$ over time, while the temporal mean and median values are measures
of the central tendency of $\boldsymbol{\mu}_p$ (Witte and Witte, 2017). The sample entropy measures the complexity of $\boldsymbol{\mu}_p$, with a lower sample
entropy indicating greater self-similarity and less randomness in $\boldsymbol{\mu}_p$. The frequency-domain features are calculated based on

the fast Fourier transform (FFT). The FFT mean is the mean value of the Fourier coefficients for $\boldsymbol{\mu}_p$ and the FFT shape mean describes the shape of the Fourier coefficients. These quantities are formulated as follows:

$$\text{FFT mean} = \frac{1}{N}\sum_{k=1}^{N}|\mu_{ik}|, \tag{8}$$

$$\text{FFT shape mean} = \frac{1}{\sum_{k=1}^{N}|\mu_{ik}|}\sum_{k=1}^{N}k|\mu_{ik}|, \tag{9}$$

where $\mu_{ik}$ is the Fourier coefficient and $N$ is the length of $\boldsymbol{\mu}_p$. These features are calculated from the simulated observations at each site and provided to XGBoost as initial inputs.

**Table 1.** Summary of the basic information on the observation series features.

| Attribute | Feature | Description |
|---|---|---|
| | Wave rate | Difference between 90-th and 10-th quantile of normalized observation series |
| Time domain | Mean | Temporal mean value of observation series |
| | Median | Temporal median value of observation series |
| | Sample entropy | Complexity of observation series |
| Frequency domain | FFT mean | Amplitude of power spectral density by FFT |
| | FFT shape mean | Shape of power spectral density by FFT |

## 2.4 Release rate estimation

Once the source location has been retrieved, many existing methods can be used to inversely estimate the release rate. In this study, we choose the recently developed PAMILT method (Fang et al., 2022) because it can correct the intrinsic model errors of the release rate estimation and accurately retrieve the temporal variations in the release rates.

## 2.5 Numerical implementation

### 2.5.1 Pre-screening of potential source locations

To reduce the computational cost and remove low-quality samples, the search range for the source location is pre-screened by evaluating the correlation coefficients between the observations and atmospheric dispersion model simulations, where the candidate source locations are randomly sampled in the considered calculation domain. Because the release rate is unknown, it is assumed to be 1 for all simulations. Source locations corresponding to the highest 40% of correlation coefficients are selected as the search range of the subsequent refined source location estimation using XGBoost.

### 2.5.2 Samples for training XGBoost

The samples for training $G(\mathbf{X})$ in Eq. (5) are generated based on the simulations described in Sect. 2.5.1, and the source locations of these simulations are within the search range determined according to Sect. 2.5.1. The simulation data are scaled by a constant factor (the ratio between the median value of all observations and that of the simulations using a unit release rate), which ensures that the simulations and observations have the same order of magnitude. Gaussian noise is added to the simulation data to simulate the statistical fluctuations of the measurements. The simulations between the first and last data points above the noise level are filtered by a temporal sliding-window average filter with a window size of 5, yielding samples for feature extraction as described in Sect. 2.3.

### 2.5.3 Automatic optimization of XGBoost model

The XGBoost model for source location estimation is automatically optimized with respect to the hyperparameters and feature selection. Specifically, the Bayesian optimization algorithm is used to optimize the hyperparameters by minimizing the following generalization coefficient (GC) defined under the five-fold cross-validation framework:

$$\text{GC} = (1 - \text{MCV})^2 + Var(R_k^2) , \tag{10}$$

$$\text{MCV} = \frac{1}{5}\sum_k R_k^2 , \tag{11}$$

where $R_k^2$ is the goodness of fit and $k$ is the index of each fold ($k = 1, 2, …, 5$). MCV is the mean cross-validation score $R_k^2$ among the five folds and $Var(R_k^2)$ measures the variance of $R_k^2$. This function aims to balance the average and the variance of $R_k^2$, thus enhancing the generalization ability of the XGBoost model. In this study, the optimized hyperparameters include *max_depth* (maximum depth of a decision tree), *learning_rate* (step size shrinkage when updating), *n_estimators* (number of decision trees), *min_child_weight* (minimum sum of sample weight of a child node), *subsample* (subsample ratio of the training samples), *colsample_bytree* (subsample ratio of columns when constructing a decision tree), *reg_lambda* (L2 regularization term on weights), and *gamma* (minimum loss reduction required to split the decision tree).

The initial input features (Table 1) are optimized through a feature selection step, where MCV serves as the selection criterion. The selection is implemented by recursively removing the feature with the least importance, and reassessing the MCV based on cross-validation (Akhtar et al., 2019). Initially, an XGBoost model is trained with all features, and the importance of each feature is assessed based on its contribution to the model accuracy. The feature with the least importance is removed and the XGBoost model is retrained using the remaining features. The feature importance and MCV are updated accordingly and another feature is removed. This iterative process continues until the optimal number of features is identified, corresponding to the highest MCV achieved during the process. The overall flowchart of the proposed spatiotemporally separated source reconstruction model is shown in Fig. S1.

**2.6 Validation case**

**2.6.1 Field experiments**

The proposed methodology was validated against the observations of the SCK-CEN [41]Ar and ETEX-1 field experiments. The SCK-CEN [41]Ar experiment was carried out at the BR1 research reactor in Mol, Belgium, in October 2001 as a collaboration between NKS and the Belgian Nuclear Research Centre (SCK-CEN) (Rojas-Palma et al., 2004). The major part of the experiment was conducted on 3–4 October, during which time [41]Ar was emitted from a 60-m stack with a release rate of approximately $1.5 \times 10^{11}$ Bq h$^{-1}$. Meteorological data such as wind speed and direction were provided by the on-site weather mast. For most of the experimental period, the atmospheric stability was neutral, and the wind was blowing from the southwest. As illustrated in Fig. 2(a), the source coordinates were (650 m, 210 m). The 60-s-average ground-level fluence rates were continuously collected by an array of NaI (Tl) gamma detectors, with different observation sites used on the two days. To convert the measured fluence rates to gamma dose rates (mSv/h), we used the [41]Ar parameters of a previous study (Li et al., 2019a): $E_\gamma = 1.2938$ MeV, $f^n(E_\gamma) = 0.9921$, $\mu_a = 2.05 \times 10^{-3}$ m$^{-1}$, and $\omega = 7.3516 \times 10^{-1}$ Sv Gy$^{-1}$. More details of these measurements can be found in (Rojas-Palma et al., 2004).

The ETEX-1 experiment took place at Monterfil in Brittany, France, on 23 October 1994 (Nodop et al., 1998). During ETEX-1, a total of 340 kg of PMCH was released into the atmosphere on 23 October 1994 at 16:00:00 UTC and 24 October 1994 at 03:50:00 UTC. As illustrated in Fig. 2(b), the source coordinates were (-2.0083°E, 48.058°N). A total of 3104 available observations (3-h-averaged concentrations) were collected at 168 ground sites. ETEX-1 has been widely used as a validation scenario for reconstructing atmospheric radionuclide releases (Ulimoen and Klein, 2023; Tomas et al., 2021). The candidate source locations are uniformly sampled from the green shaded zone. We choose two groups of observation sites: the first comprises four sites (i.e. B05, D10, D16, F02) randomly selected from the sites within the sample zone (Group 1, with a total of 92 available observations), and the second involves four sites (i.e. CR02, D15, DK08, S09) randomly selected from the sites beyond the sample zone boundaries (Group 2, with a total of 90 available observations). Compared with the SCK-CEN [41]Ar experiment, the ETEX-1 observations exhibit temporal sparsity, lower temporal resolution, and increased complexity in meteorological conditions.

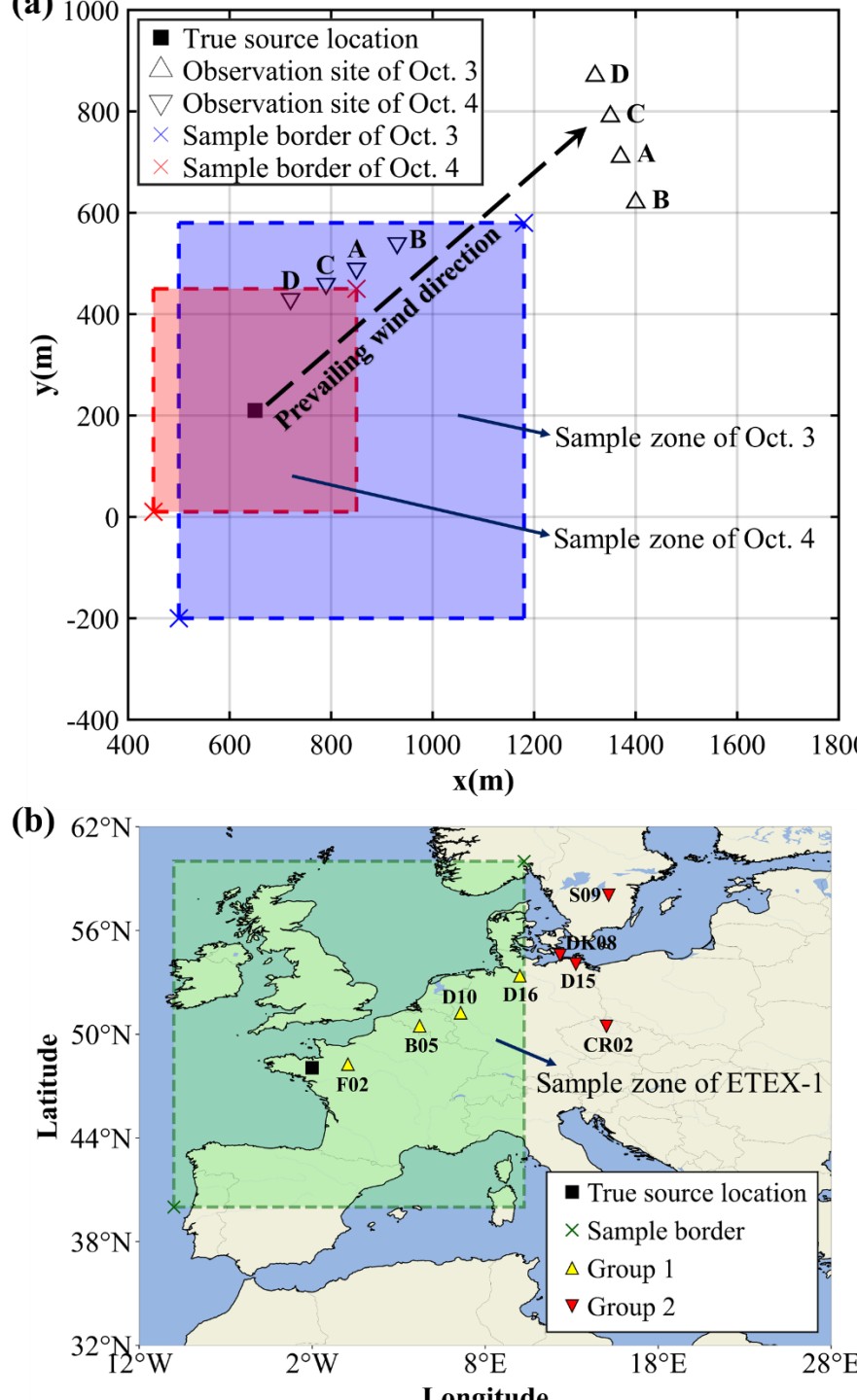


**Figure 2.** Release location and observation sites of two field experiments. (a) SCK-CEN $^{41}$Ar experiment. The map was created based on
the relative positions of the release source and observation sites (Drews et al., 2002). The coordinates of the sample border are (500 m, −200
m) and (1180 m, 580 m) on Oct. 3, and (450 m, 10 m) and (850 m, 450 m) on Oct. 4. This figure was plotted using MATLAB 2016b, rather
than created by a map provider; (b) ETEX-1 experiment. The map was created based on the real longitudes and latitudes of the release source
and observation sites (Nodop et al., 1998). The coordinates of the sample border are (10°W, 40°N) and (10°E, 60°N). This figure was plotted
using the cartopy function of Python, rather than created by a map provider.
**2.6.2 Simulation settings of atmospheric dispersion model**
For the SCK-CEN $^{41}$Ar field experiment, the Risø Mesoscale PUFF (RIMPUFF) model was employed to simulate the
dispersion of radionuclides and calculate the dose rates at each observation site (Thykier-Nielsen et al., 1999). The simulations
used on-site measured meteorological data and the modified Karlsruhe–Jülich diffusion coefficients. The calculation domain
measured 1800 m×1800 m and the grid resolution was 10 m×10 m. The release height of $^{41}$Ar was assumed to be 60 m. Other
RIMPUFF calculation settings followed those of a previous study (Li et al., 2019a), and have been validated against the
observations. To establish the datasets for the XGBoost model, 2000 simulations and 1000 simulations with different source
locations were performed by RIMPUFF for the experiments on Oct. 3 and Oct. 4, respectively. Candidate source locations
were randomly sampled from the shaded zones in Fig. 2(a), which were determined according to the positions of the
observation sites and the upwind direction. Each simulation, along with its corresponding source location, forms one
sample. As described in Sect. 2.5.1, we calculated the correlation coefficient for each sample and preserved the 40% of samples
with the highest 40% of correlation coefficients (i.e. 800 samples for Oct. 3 and 400 samples for Oct. 4). The constant factors
mentioned in Sect. 2.5.2 are $1.53×10^{11}$ and $1.48×10^{11}$ for Oct. 3 and Oct. 4, respectively.
For the ETEX-1 experiment, the FLEXible PARTicle (FLEXPART) model (version 10.4) was applied to simulate the
dispersion of PMCH (Pisso et al., 2019). The meteorological data were obtained from the United States National Centers of
Environmental Prediction Climate Forecast System Reanalysis, and have a spatial resolution of 0.5°×0.5° and time resolution
of 6 h. To rapidly establish the relationship between the varying source locations and the observations, 182 backward
simulations were performed using FLEXPART with a time interval of 3 h, grid size of 0.25°×0.25°, and 8 vertical levels (from
100–50000 m). Only the lowest model output layer was used for source reconstruction. Candidate source locations were
uniformly sampled from the shaded zone in Fig. 2(b), resulting in a total of 6561 source locations. As described in Sect. 2.5.1,
2624 candidate source locations were preserved following the pre-screening step. The constant factors mentioned in Sect. 2.5.2
are $5.60×10^{12}$ and $2.86×10^{13}$ for Group 1 and Group 2, respectively.
**2.7 Sensitivity study**
(1) Search range
The search range is controlled by the pre-screening threshold, which is the top proportion of the correlation coefficients in
the pre-screening step. Specifically, we use source locations corresponding to the highest 20%, 40%, 50%, 60%, 80%, and
100% of correlation coefficients to define the search ranges, with a lower proportion indicating a narrower and more focused
search area.
(2) Size of the sliding window

276       Temporal filtering with different sliding-window sizes is applied to separate the source location estimation from the release

rate estimation. In this study, the size of the sliding window ranges from 3–10. With these filtered data, the XGBoost model is
trained using the same pattern for the source location estimation.
(3) Feature type

280       The XGBoost model is trained using only time-domain features and only frequency-domain features to investigate the

influence of these features on the source location estimation. The performance of the time-feature-only and frequency-feature-
only models is compared with the all-features result.
(4) Number and combination of observation sites

284       The XGBoost model is trained and applied to the source location estimation with different numbers of observation sites,

namely a single site, two sites, and three sites. For the two- and three-site cases, the model is trained using different
combinations of sites and the source location is estimated accordingly.
(5) Meteorological errors

288       Meteorological errors are important uncertainties in source reconstruction, especially the random errors in the wind field

(Mekhaimr and Abdel Wahab, 2019). To simulate such uncertainties, a stochastic perturbation of ±10% is introduced to the
observed wind speeds in the x and y components, and a ±1 stability class perturbation is applied to the stability parameters
(e.g., from C to B or D). For both days, 50 meteorological groups are generated based on these random perturbations.

292       In all the sensitivity tests, the source location is estimated 50 times with randomly initialized hyperparameters to demonstrate

the uncertainty range of the proposed method under different circumstances. The performance of source location estimation is
compared quantitatively using the metrics specified in Sect. 2.8.3.
**2.8 Performance evaluation**
**2.8.1 Observation filtering**
The feasibility of filtering is demonstrated using both the synthetic and real observations of the SCK-CEN [41]Ar experiment
and the real observations of the ETEX-1 experiment. The synthetic observations are generated by a simulation using a synthetic
temporally varying release profile with sharp increase, stable, and gradual decrease phases (as illustrated in Fig. S2), which is
typical for an atmospheric radionuclide release (Davoine and Bocquet, 2007). Because several temporal observations are
missing at some observation sites, we only choose observations sampled between 24 October 1994 09:00:00 UTC and 26
October 1994 03:00:00 UTC for the source location estimation. The simulations corresponding to the synthetic and real
observations should first be processed following the procedure described in Sect. 2.5.2. The filtering performance is evaluated
by comparing the simulation–observation differences before and after the filtering step. Several statistical metrics can be used
to quantify this difference, including the normalized mean square error (NMSE), Pearson's correlation coefficient (PCC), and
the fraction of predictions within a factor of 2 and 5 of the observations (FAC 2 and FAC 5, respectively) (Chang and Hanna,

307    2004).

**2.8.2 Optimization of the XGBoost model**

The hyperparameters are optimized with respect to the GC in Eq. (10) and the features are optimized with respect to the MCV
in Eq. (11). Larger values of MCV and smaller values of GC indicate better optimization performance. In addition, the
importance of each feature to the XGBoost training is evaluated with the built-in *feature importance* measure of the XGBoost
model.

**2.8.3 Source reconstruction**

The relative errors in the source location ($\delta_{\mathbf{r}}$) and total release ($\delta_Q$) are calculated to evaluate the source reconstruction accuracy:
$$\delta_{\mathbf{r}} = \frac{|\mathbf{r}_{true} - \mathbf{r}_{est}|}{L_D} \times 100\% ,\qquad(12)$$
$$\delta_Q = \frac{Q_{true} - Q_{est}}{Q_{true}} \times 100\% ,\qquad(13)$$
where $\mathbf{r}_{true}$ and $Q_{true}$ refer to the real source location and total release of the field experiment and $\mathbf{r}_{est}$ and $Q_{est}$ are the
estimated location and total release, respectively. $L_D$ represents the range of the source domain, which is the distance between
the lower and upper borders of the sampled zone (Fig. 2). The values of $\mathbf{r}_{true}$, $L_D$, and $Q_{true}$ are listed in Table 2. In addition
to the total release, the reconstructed release rates are also compared with the true temporal release profile.
**Table 2.** Parameter settings of field experiments.

| Experiment | Case | Parameters | | |
| --- | --- | --- | --- | --- |
| | | $\mathbf{r}_{true}$ | $L_D$ | $Q_{true}$ |
| SCK-CEN [41]Ar | Oct. 3 | (650 m, 210 m) | 1034.8 m | 423.10 GBq |
| | Oct. 4 | (650 m, 210 m) | 565.7 m | 1045.09 GBq |
| ETEX-1 | Group 1 | (2.0083°W, 48.058°N) | 2620.5 km | 340 kg |
| | Group 2 | (2.0083°W, 48.058°N) | 2620.5 km | 340 kg |

**2.8.4 Comparison with the Bayesian method**

The proposed method is compared with the popular Bayesian method based on the SCK-CEN [41]Ar and ETEX-1 experiments,
with the same search range used for locating the source in both methods (Fig. 2). The Bayesian method is augmented with an
in-loop inversion of the release rate at each iteration of the Markov chain Monte Carlo sampling. The prior distribution of the
Bayesian method is a uniform distribution and the likelihood is a log-Cauchy distribution. More detailed information is
presented in Supplementary Note S1.

### 2.8.5 Uncertainty range

The uncertainty ranges are calculated and compared for the correlation-based method, the Bayesian method, and the proposed method. For the correlation-based method, the uncertainty range is calculated using the source locations with the top-50 correlation coefficients. For the proposed method, the uncertainty range is calculated from 50 Monte Carlo runs with randomly initialized hyperparameters. The Bayesian method provides the uncertainty range directly through the posterior distribution. For consistency with the other two methods, the results with the top-50 frequencies are selected for the comparison.

## 3. Results and Discussion

### 3.1 Filtering performance

Figure S3 displays the original and filtered observations at different observation sites for both days. The results demonstrate that the peak values have been smoothed out and the long-term trends are preserved to a large degree. Figure 3 compares the filtering performance for both the synthetic and real observations, where the constant-release simulations are plotted against the observations before and after filtering. For the synthetic observations, the filtered data are more concentrated along the 1:1 line for both days, and all filtered data fall within the 2-fold lines for Oct. 3. For the real observations, the dots before filtering in Fig. 3 have a dispersed distribution for both Oct. 3 and Oct. 4, indicating limited correlations with the simulations. After filtering, the dots are more concentrated towards the 1:1 line for both the SCK-CEN $^{41}$Ar and ETEX-1 experiments. These phenomena indicate a noticeably increased agreement between the filtered observations and the constant-release simulations.

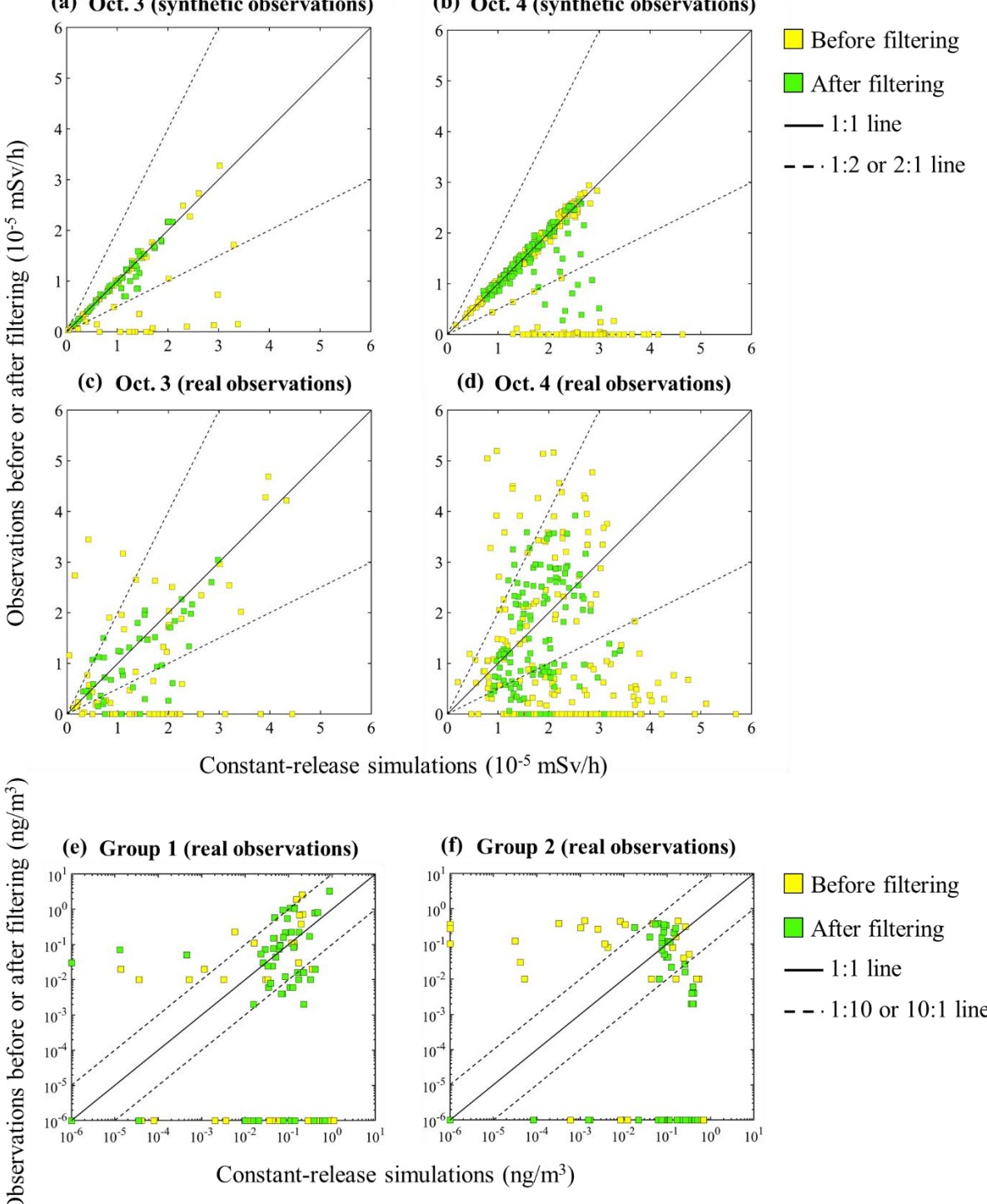

**Figure 3.** Scatter plots of the original (yellow squares) and filtered (green squares) observations versus the constant-release simulation results.

SCK-CEN $^{41}$Ar experiment: (a) Oct. 3 (synthetic observations); (b) Oct. 4 (synthetic observations); (c) Oct. 3 (real observations); (d) Oct. 4
(real observations); ETEX-1 experiment: (e) Group 1 (real observations); (f) Group 2 (real observations).
Table 3 quantitatively compares the results presented in Fig. 3. For each case, all metrics are greatly improved after filtering,
confirming the better agreement between the filtered observations and the constant-release simulations. The improved
agreement indicates that the filtering step significantly reduces the influence of temporal variations in release rates across the
observations. The filtering performs better with the synthetic observations than with the real observations because the synthetic
observations are free of measurement errors. The filtering process produces a better effect with the SCK-CEN $^{41}$Ar experiment
than with the ETEX-1 experiment, owing to the sparser observations in the ETEX-1 experiment (Fig. S3).
**Table 3.** Quantitative metrics for the filtering validation.

| Experiment | Case | | NMSE | PCC | FAC2 | FAC5 |
|---|---|---|---|---|---|---|
| SCK-CEN $^{41}$Ar | Oct. 3 (synthetic observations) | Before filtering | 0.6970 | 0.5315 | 0.7647 | 0.8235 |
| | | After filtering | 0.0239 | 0.9514 | 1 | 1 |
| | Oct. 4 (synthetic observations) | Before filtering | 0.9290 | -0.0267 | 0.7292 | 0.7292 |
| | | After filtering | 0.0956 | 0.6179 | 0.9412 | 0.9779 |
| | Oct. 3 (real observations) | Before filtering | 1.4437 | 0.3572 | 0.3824 | 0.5147 |
| | | After filtering | 0.2730 | 0.6976 | 0.7273 | 0.8864 |
| | Oct. 4 (real observations) | Before filtering | 1.9290 | -0.2099 | 0.3073 | 0.4948 |
| | | After filtering | 0.3668 | 0.2802 | 0.6552 | 0.9310 |
| ETEX-1 | Group 1 (real observations) | Before filtering | 10.9936 | 0.3414 | 0.1000 | 0.2167 |
| | | After filtering | 6.6769 | 0.5145 | 0.2500 | 0.3667 |
| | Group 2 (real observations) | Before filtering | 5.8705 | -0.2824 | 0.0667 | 0.1167 |
| | | After filtering | 4.9799 | -0.2695 | 0.1167 | 0.2500 |

## 3.2 Optimization of XGBoost model

### 3.2.1 Hyperparameters

Table S1 summarizes the optimal hyperparameters and corresponding GCs used for source location estimation in this study;
Tables S2–S5 includes all the optimal hyperparameters used in the 50 runs of the SCK-CEN $^{41}$Ar and ETEX-1 experiments.
The optimal GCs of the SCK-CEN $^{41}$Ar experiment are smaller than those of the ETEX-1 experiment, indicating better fitting
performance. This is because the sparse observations of the ETEX-1 experiment (Fig. S3) are more sensitive to the added
Gaussian noise (see Sect. 2.5.2).
**3.2.2 Feature selection**
Figure 4 compares the importance of the selected features at each site for the two experiments. The time-domain features are
dominant for both days in the SCK-CEN [41]Ar experiment (Fig. 4a and 4b). For Oct. 3, Site B is the most important, possibly
because it is farthest away in the crosswind direction. For Oct. 4, the four sites provide redundant feature information, and
many features are removed. This is because the distribution of observation sites is almost parallel to the wind direction on this
day. According to Fig. S3(b), the measurements from Sites A and B have a high correlation, thus leading to the removal of
features from Site A on Oct. 4. In summary, the feature selection process adapts XGBoost to different application scenarios.
Figure S4(a) and S4(b) shows the variations in MCV with the number of features for the x and y coordinates. The MCV first
increases with the number of features, and then decreases slightly after reaching the maximum. The optimal number of features
for Oct. 4 is noticeably smaller than for Oct. 3. In addition, the selected features for Oct. 3 involve all four sites, whereas those
for Oct. 4 involve three sites. The reduced features and site numbers indicate a high level of redundancy in the observations
acquired on Oct. 4. This is because the observation sites are parallel to the downwind direction and provide similar location
information in the crosswind direction.

375        For the ETEX-1 experiment, Fig. 4c and d shows that the features of Group1 and Group2 are largely preserved after the

feature selection process (only one feature is removed for each case), indicating less redundancy than that in the SCK-CEN
[41]Ar experiment. The time-domain features are dominant, but the frequency-domain features at some sites (e.g. D16 and S09)
also play important roles. The MCVs of the ETEX-1 experiment have similar variation trends as those for the SCK-CEN [41]Ar
experiment (Fig. S4c and S4d).

**(a)**

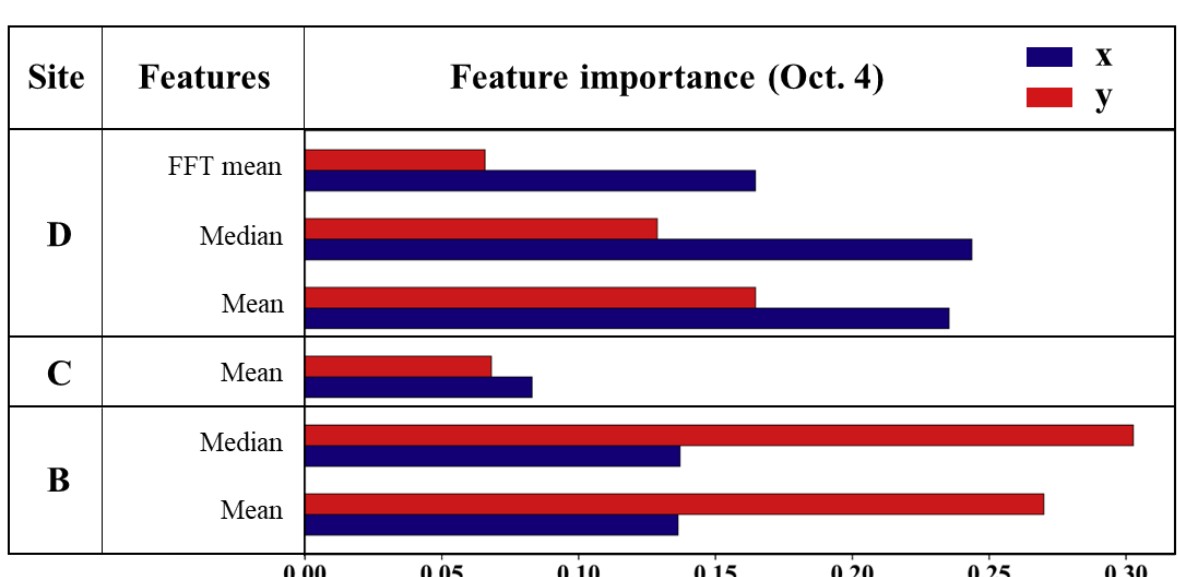

**(b)**

**(c)**

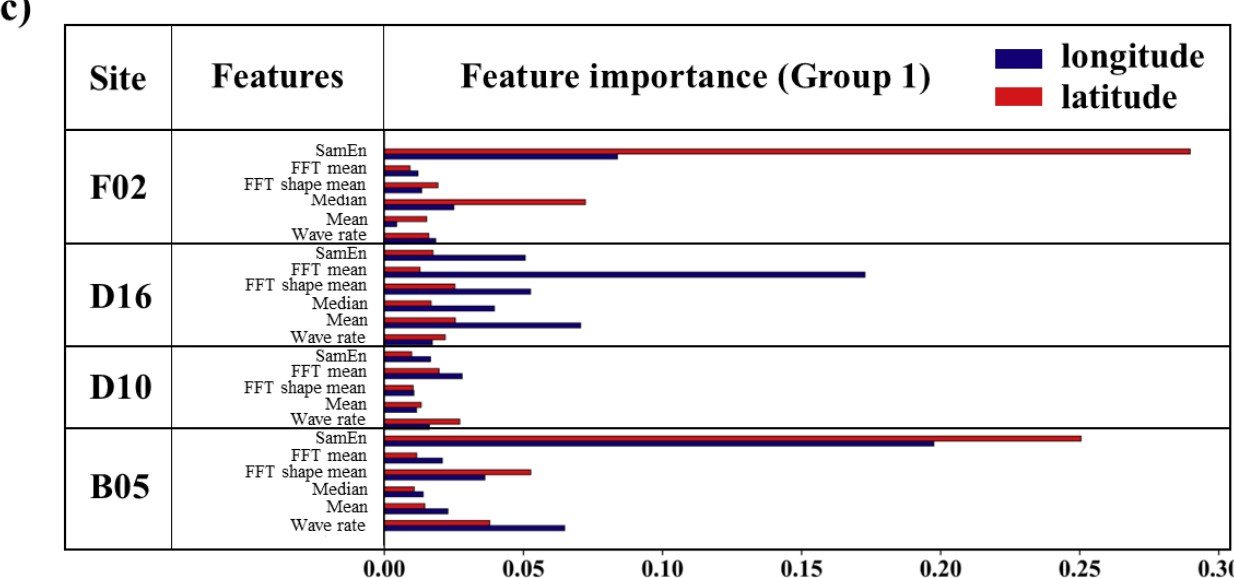

**(d)**

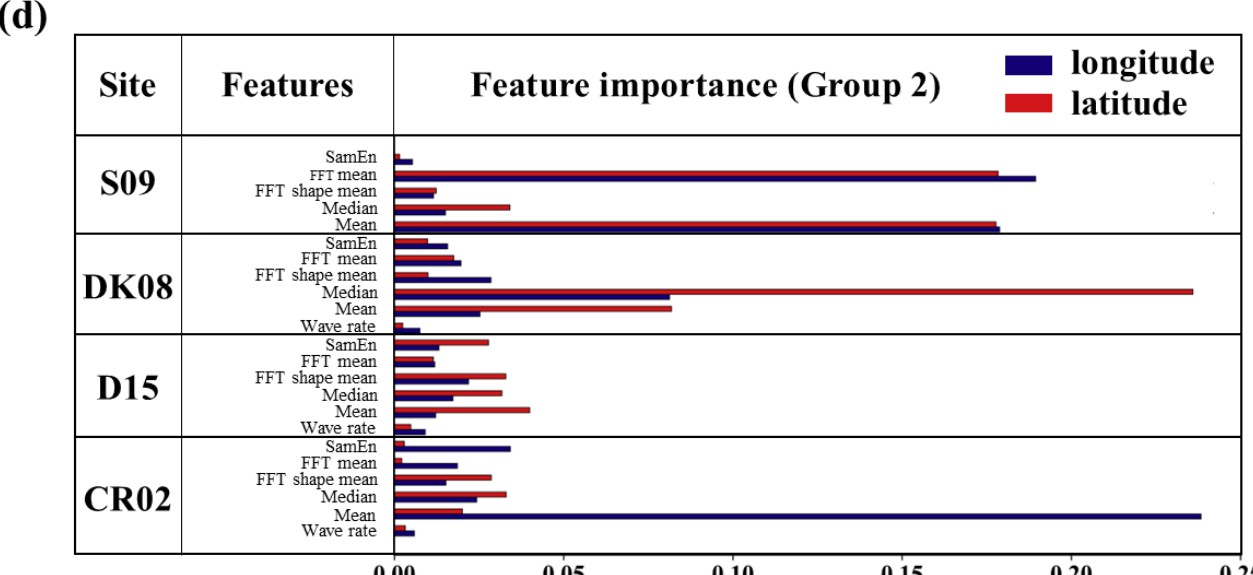

**Figure 4.** Feature importance of SCK-CEN [41]Ar experiment: (a) Oct. 3; (b) Oct. 4; and ETEX-1 experiment: (c) Group 1; (d) Group 2.

**3.3 Source reconstruction**

**3.3.1 Source locations**

Figure 5 compares the best-estimated source locations of the correlation-based method, the Bayesian method, and the proposed method with the ground truth. The pre-screening zone covers the true source location for both days, but the areas with the

highest correlation coefficients are still too large for the point source to be accurately located. The locations with the maximum correlation exhibit errors of 270.19 m and 36.06 m for Oct. 3 and Oct. 4, respectively, indicating that the correlation-based method may produce biased results in the case of non-constant releases. The Bayesian method estimates the location with errors of 19.62 m and 52.81 m for Oct. 3 and Oct. 4, respectively. In comparison, the proposed method achieves the best performance. The estimates without feature selection are only 10.65 m (Oct. 3) and 20.62 m (Oct. 4) away from the true locations. Feature selection further reduces these errors to 6.19 m (Oct. 3, a relative error of 0.60%) and 4.52 m (Oct. 4, a relative error of 0.80%), which are below the grid size (10 m×10 m) of the atmospheric dispersion simulation. The ability to estimate the source location with accuracy surpassing the grid size can be attributed to the strong fitting capability of the optimized XGBoost model (Chen and Guestrin, 2016; Grinsztajn et al., 2022). However, this capability, although inherent, is not present across all optimized XGBoost models, as external factors such as observation noises and meteorological data inaccuracies can also impact the accuracy of source location estimation.

For the ETEX-1 experiment, the pre-screening zone also covers the true source location for Group 1 and Group 2. The source locations estimated by the correlation-based method are 411.85 km and 486.41 km away from the ground truth for Group 1 and Group 2, respectively. The location error of the Bayesian method estimates is only 30.50 km for Group 1, but increases to 520.77 km for Group 2, indicating the sensitivity of this method to the observations. In contrast, the proposed method achieves much lower source location errors of 5.19 km for Group 1 (a relative error of 0.20%) and 17.65 km for Group 2 (a relative error of 0.70%). Group1 exhibits a lower source location error than Group 2, because the observation sites of Group 1 are closer to the sampled source locations than those of Group 2 and better characterize the plume. Feature selection did not remove many features (Fig. 4c and 4d), so the estimated source locations with and without feature selection basically overlap for both groups.

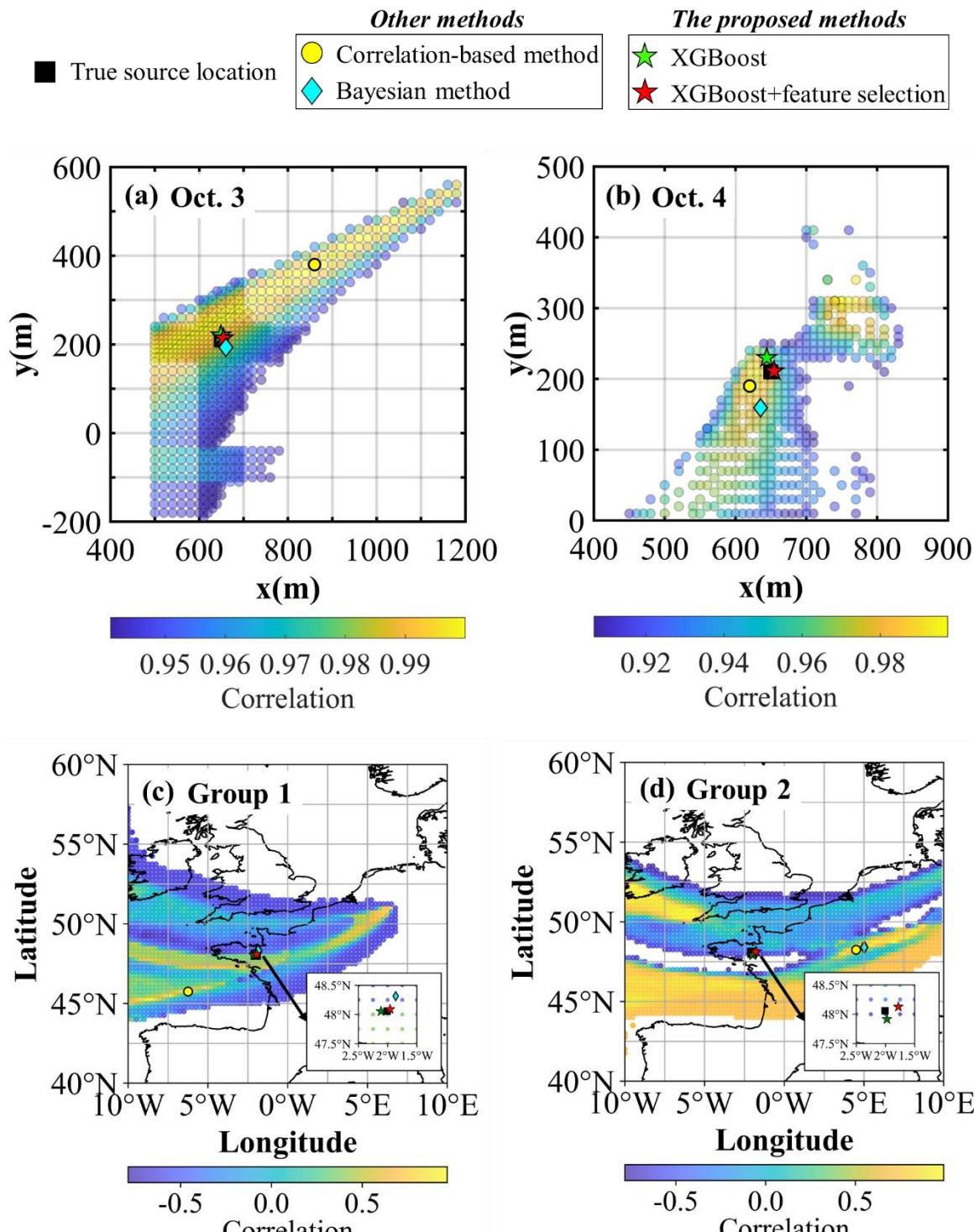

**Figure 5.** Source location estimation results of SCK-CEN [41]Ar experiment: (a) Oct. 3; (b) Oct. 4; and ETEX-1 experiment: (c) Group 1; (d)

407

408

Group 2. A detailed enlargement of the region around (2.5°W, 47.5°N) to (1.5°W, 48.5°N) is shown in the bottom right corner in (c) and (d) to highlight the source location estimation results of the proposed method. The yellow dots denote the maximum correlation points, which are the results of the correlation-based method. The green and red stars represent the results based on XGBoost before and after feature selection, respectively. The cyan diamonds represent the results based on the Bayesian method.

### 3.3.2 Release rates

Figure 6 displays the release rates estimated by the Bayesian and PAMILT methods based on the source location estimates in Fig. 5. For the SCK-CEN $^{41}$Ar experiment (Fig. 6a and 6b), the release rates provided by the Bayesian method present several sharp peaks, corresponding to overestimates of up to 269.03% (Oct. 3) and 532.35% (Oct. 4). Furthermore, the Bayesian estimates exhibit unrealistic oscillations in the stable release phase. In contrast, the PAMILT method successfully retrieves the peak releases without oscillations for both days. Both the Bayesian and PAMILT estimates give delayed release start times, but accurately estimate the end times, especially for Oct. 3. The PAMILT estimate underestimates the total release by 30.01% and 45.95% for Oct. 3 and Oct. 4, respectively; these values decrease to about 23.83% and 30.60%, respectively, after feature selection. The Bayesian method gives better total releases because of the overestimated peaks.

For the ETEX-1 experiment (Fig. 6c and 6d), the Bayesian estimates exhibit notable fluctuations, leading to underestimations of 58.11% for Group1 and 51.44% for Group 2. Furthermore, the temporal profile of the Bayesian estimates for Group 2 falls completely outside the true release window. In contrast, most releases using the PAMILT estimates are within the true release time window, especially for Group 2, despite the overestimations reaching 52.38% for Group 1 and 57.65% for Group 2, after the feature selection process. Compared with the SCK-CEN $^{41}$Ar experiment, the increased deviation in the ETEX-1 experiment is caused by the sparsity of observations at the four sites (Fig. S3).

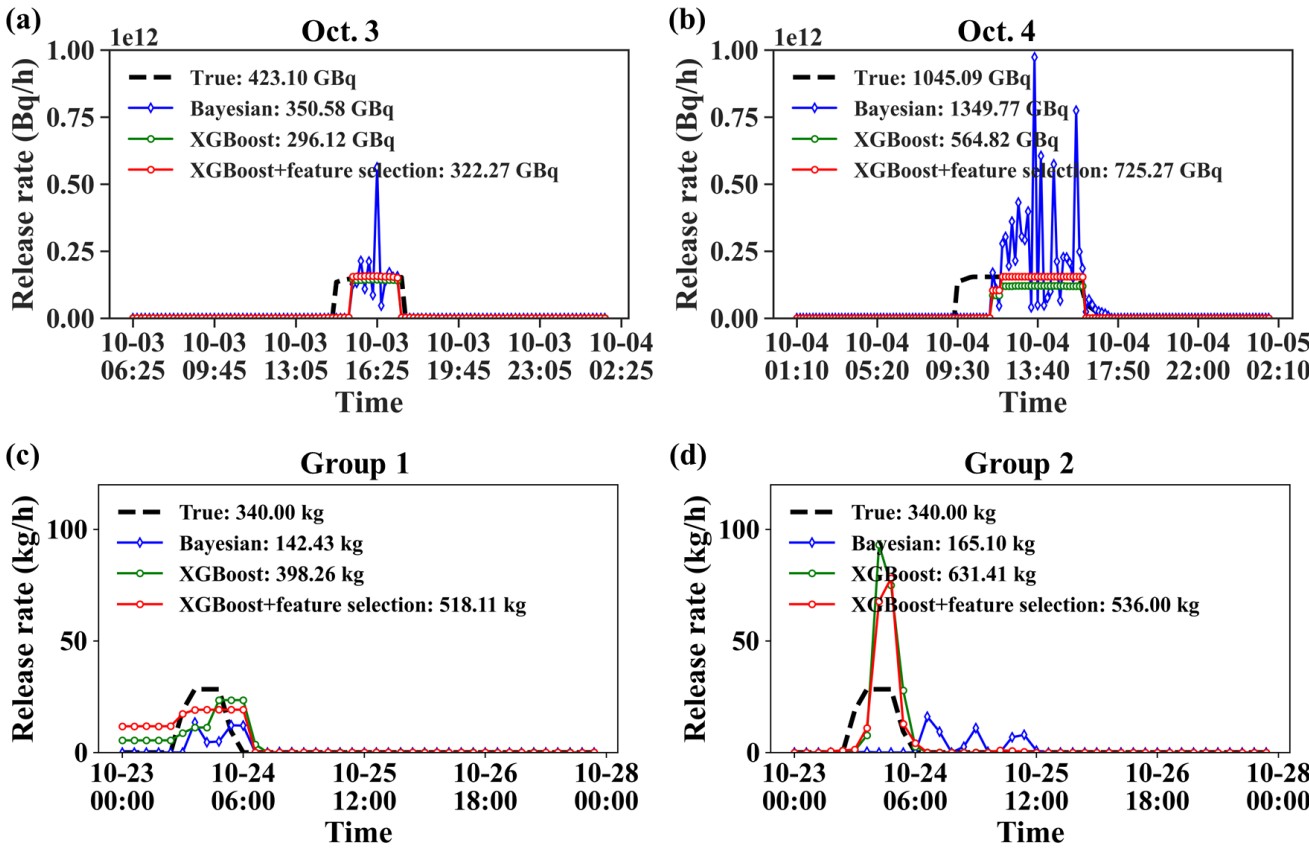

428

**Figure 6.** Release rate estimation results with different location estimates of SCK-CEN [41]Ar experiment: (a) Oct. 3; (b) Oct. 4; and ETEX-1 experiment: (c) Group 1; (d) Group 2. The release rates labelled XGBoost or XGBoost+feature selection are estimated using the PAMILT method.

### 3.3.3 Uncertainty range

Figure 7 compares the spatial distribution of 50 estimates produced by different methods. For the SCK-CEN [41]Ar experiment, the estimates of the correlation-based method are highly dispersed for both days, leading to a very uniform distribution of the x coordinate for Oct. 3 and two separate distributions of both the coordinates for Oct. 4. The Bayesian method produces a multimodal distribution for both days, in which the estimates are more concentrated than those of the correlation-based method. The corresponding full posteriori distributions in Fig. S5(a) and S5(b) better reveal the multimodal feature of the Bayesian method, with several peaks of similar probabilities in the estimates of both coordinates on Oct. 3 and the y coordinate on Oct. 4. The multimodal feature indicates the difficulty of constraining the solution in simultaneous spatiotemporal reconstruction, as reported in a previous study (Meutter and Hoffman, 2020). In comparison, the proposed method provides the most concentrated source location estimates. The feature selection moves the centre of the distribution closer to the true location and narrows the distribution of the estimates, especially for Oct. 4.

For the ETEX-1 experiment, the estimates of the correlation-based method are quite dispersed, whereas those of the

Bayesian method are more concentrated. The Bayesian estimates are close to the truth for Group 1, but deviate noticeably for
Group 2. This phenomenon indicates that the Bayesian method is sensitive to the observations, especially when the
observations are sparse. Figure S5(c) and S5(d) reveals that the Bayesian-estimated posterior distribution is multimodal for
both ETEX-1 groups; this can be avoided by using additional observations (Fig. S5e). In contrast, the proposed method
provides estimates that are concentrated around the truth for both Group 1 and Group 2, indicating its efficiency in the case of
sparse observations. Due to the shorter distance between observation sites and the sampled source locations, the uncertainty
range of source location for Group 1 is narrower than that for Group 2.

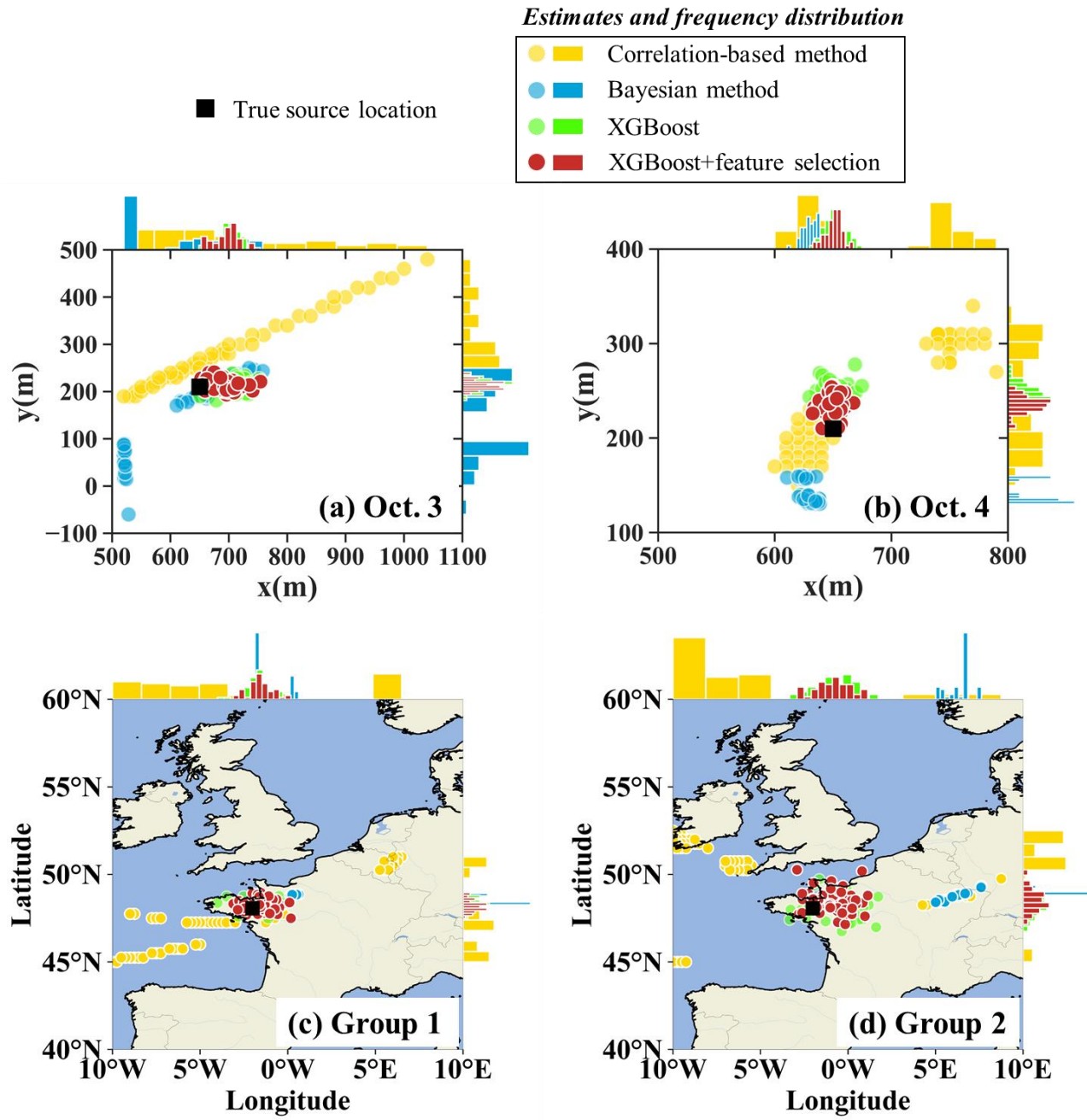


**Figure 7.** Spatial distribution of 50 source location estimates of SCK-CEN [41]Ar experiment: (a) Oct. 3; (b) Oct. 4; and ETEX-1 experiment: (c) Group 1; (d) Group 2. Each circle denotes an individual estimate as detailed in Sect. 2.8.5, with colour variations indicating the respective method employed. Histograms along the axes represent the frequency distribution of the estimates along the respective axis.

Figure 8 compares the uncertainty range and mean total release of the release rate estimations for the SCK-CEN [41]Ar experiment. For Oct. 3, the Bayesian estimates significantly overestimate the mean values and have a large uncertainty range,

whereas the mean PAMILT estimate is very close to the true release and the uncertainty range is smaller than that of the Bayesian method. For Oct. 4, the mean Bayesian estimate exhibits greater deviations than the mean PAMILT estimate. Feature selection improves the mean estimate and reduces the uncertainty range of PAMILT because it improves the source location estimation, thus reducing the deviation in the inverse model of the release rate. On Oct. 3 and Oct. 4, the PAMILT method underestimates the total release by 18.30% and 47.42%, respectively, whereas the Bayesian method gives overestimations of 153.61% and 42.29%, respectively.

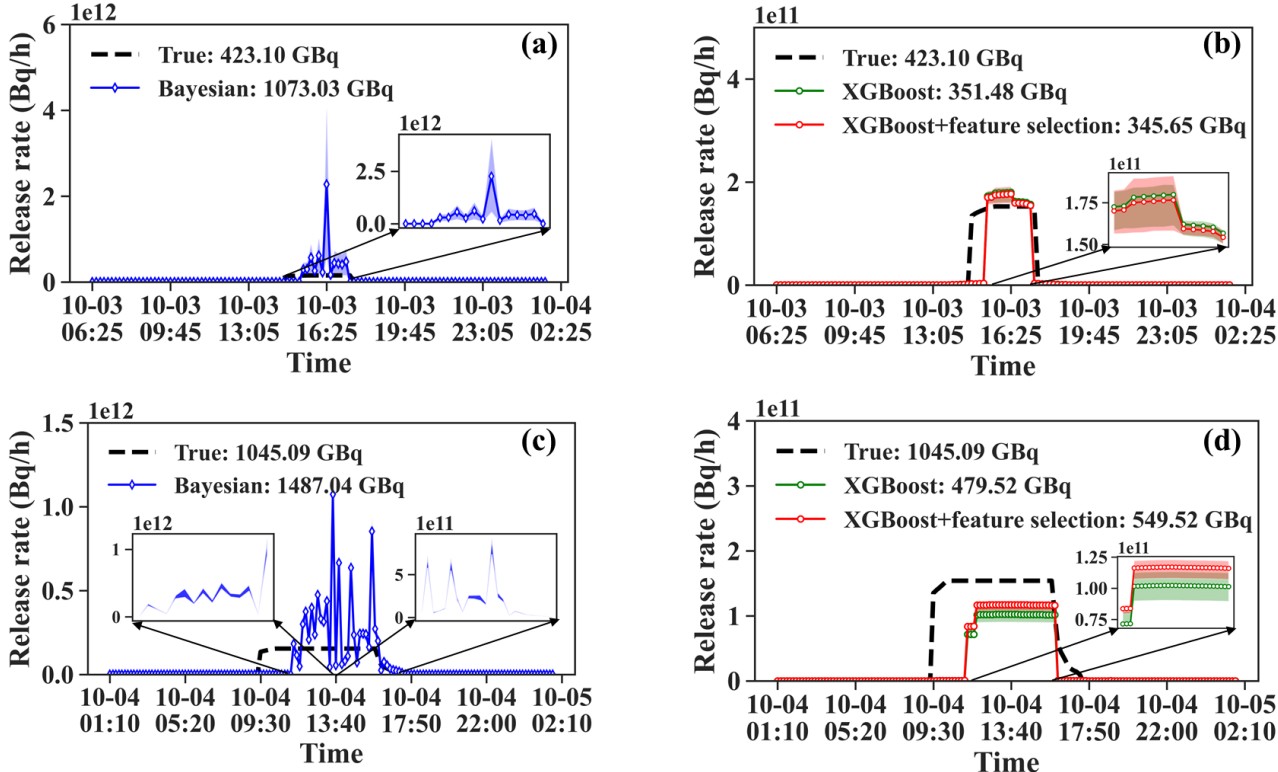

**Figure 8.** Release rate estimates over 50 calculations of SCK-CEN $^{41}$Ar experiment. (a) Oct. 3-Bayesian method; (b) Oct. 3-PAMILT method; (c) Oct. 4-Bayesian method; (d) Oct. 4-PAMILT method. The shadow represents the uncertainty range between the lower quartile and the upper quartile. The shadow of each figure is amplified by an enlarged subgraph. The legends in each figure provide the mean estimates for the total release.

Figure 9 compares the uncertainty ranges of the release rate estimates for the two ETEX-1 groups. For both groups, the Bayesian estimates exhibit noticeable underestimations (including the mean estimate) and small uncertainty ranges (Fig. 9a and 9c). The Bayesian estimates fall completely outside the true release window for Group 2 (Fig. 9c). The mean PAMILT estimates are more accurate than the mean Bayesian estimates, with most releases within the true release window (Fig. 9b and 9d). However, the PAMILT estimates have a large uncertainty range for the ETEX-I experiment than for the SCK-CEN $^{41}$Ar experiment, implying that the source–receptor matrices of the ETEX-1 experiment are more sensitive to errors in source location than those of the SCK-CEN $^{41}$Ar experiment. This greater sensitivity originates from the complex meteorology in the

ETEX-1 experiment. As for the mean total releases, the Bayesian method produces underestimations of 70.93% for Group1
and 74.15% for Group2. In comparison, the proposed method gives deviations of only 0.71% for Group 1 and 0.09% for Group
2, after feature selection.

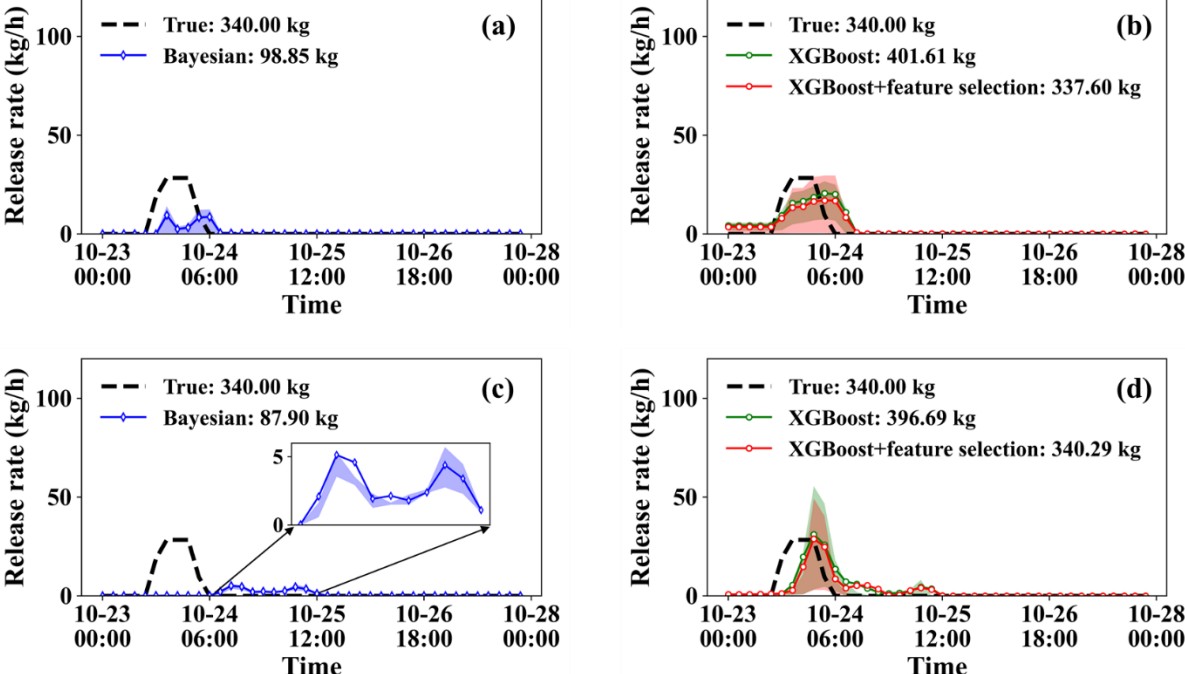

**Figure 9.** Release rate estimates over 50 calculations of ETEX-1 experiment. (a) Group 1-Bayesian method; (b) Group 1-PAMILT method; (c) Group 2-Bayesian method; (d) Group 2-PAMILT method.

Table 4 lists the mean and standard deviation of the relative errors for the 50 estimates given by different methods. The
correlation-based method produces the largest mean relative error and standard deviation for source location estimation, except
for Group 2 of ETEX-I. For the SCK-CEN [41]Ar experiment, the proposed method gives the smallest mean error, about half of
that of the Bayesian method. Its standard deviation is around one-quarter of that of the Bayesian method for Oct. 3, but is
slightly larger for Oct. 4. For the total release, the PAMILT method gives a better standard deviation of the relative error for
both days and a better mean relative error for Oct. 3, whereas the Bayesian method produces a better mean relative error for
Oct. 4. Feature selection reduces the mean relative error, except for the total release for Oct. 3, and slightly increases the
standard deviation of the source location and total release results for Oct. 3. The mean relative error of the total release averaged
on the two days is 65.09% lower than that of the Bayesian method.
For the ETEX-1 experiment, the Bayesian method exhibits case-sensitive performances with respect to the mean relative
error of source location estimation, whereas the proposed method gives the most accurate source locations with small
uncertainties for both groups. As for the total release, the proposed method gives smaller mean relative errors than the Bayesian
methods, but the Bayesian method has a smaller standard deviation. Feature selection significantly reduces the mean relative

error for the two groups. The mean relative error of the total release averaged over the two groups is 72.14% lower than that

of the Bayesian method.

**Table 4.** Relative errors of source reconstruction. $\delta_r$ represents the relative error of source location, which is positive and $\delta_Q$ denotes the relative error of total release, where a positive value indicates overestimation and a negative value denotes underestimation.

| Experiment | Case | Statistical parameters (Relative error) | | Correlation-based method | Bayesian method | The proposed method | |
|---|---|---|---|---|---|---|---|
| | | | | | | XGBoost | XGBoost+ feature selection |
| SCK-CEN $^{41}$Ar | Oct. 3 | $\delta_r$ | Mean | 14.10% | 11.88% | 5.18% | 4.68% |
| | | | Std | 11.37% | 7.53% | 1.79% | 2.05% |
| | | $\delta_Q$ | Mean | - | 153.61% | -16.93% | -18.30% |
| | | | Std | - | 189.76% | 9.45% | 8.01% |
| | Oct. 4 | $\delta_r$ | Mean | 14.30% | 12.83% | 6.83% | 4.71% |
| | | | Std | 9.60% | 1.68% | 1.76% | 1.53% |
| | | $\delta_Q$ | Mean | - | 42.29% | -54.12% | -47.42% |
| | | | Std | - | 15.05% | 6.47% | 5.85% |
| ETEX-I | Group 1 | $\delta_r$ | Mean | 16.95% | 3.22% | 2.32% | 2.42% |
| | | | Std | 7.46% | 2.75% | 1.43% | 1.43% |
| | | $\delta_Q$ | Mean | - | -70.93% | 18.12% | -0.71% |
| | | | Std | - | 17.87% | 99.85% | 102.01% |
| | Group 2 | $\delta_r$ | Mean | 21.9% | 23.97% | 5.21% | 4.97% |
| | | | Std | 5.05% | 1.97% | 2.42% | 2.35% |
| | | $\delta_Q$ | Mean | - | -74.15% | 16.67% | 0.09% |
| | | | Std | - | 11.68% | 93.50% | 109.56% |

## 3.4 Sensitivity analysis results

### 3.4.1 Sensitivity to the search range

Figure 10 displays the source location errors obtained using different pre-screening thresholds to determine the search range. The error is smaller with a lower threshold, implying that a small search range helps reduce the mean and median errors. As the threshold increases, the mean and median errors, as well as the error range, show an overall tendency to increase, but not

in a strictly monotonic way. The mean/median error is less than 12% for Oct. 3 and less than 22% for Oct. 4, indicating robust
performance in these tests. Feature selection reduces the mean/median, range, and the lower bound of the errors in most tests,
demonstrating its efficiency.

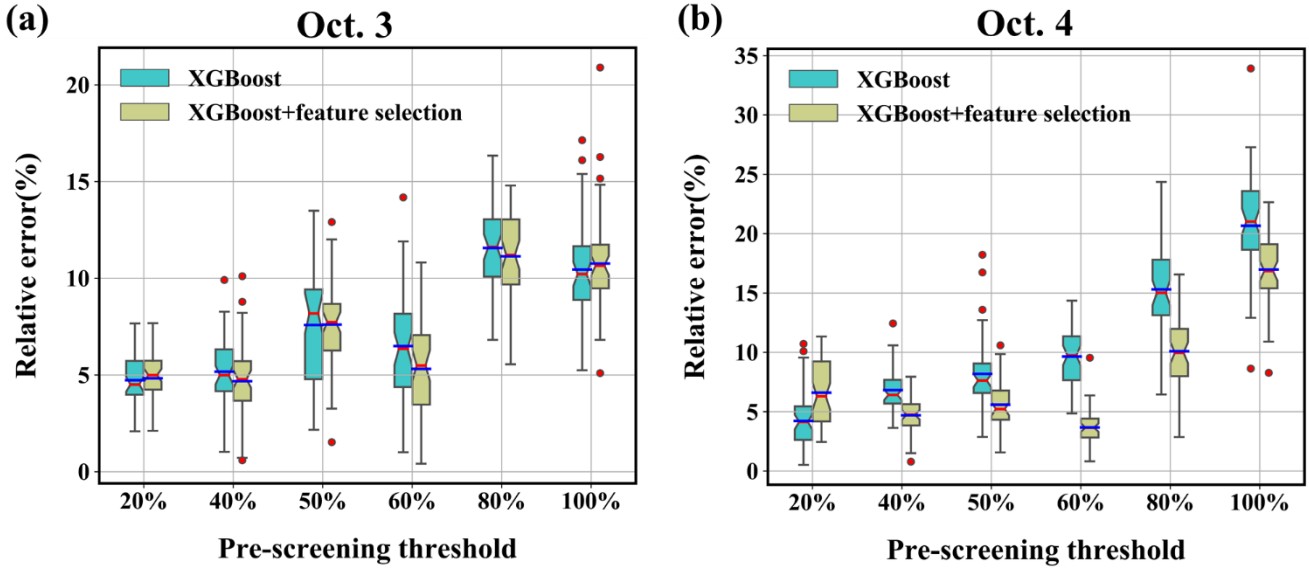


**Figure 10.** Distribution of relative error (%) over 50 runs with different search ranges. The blue and red solid lines denote average relative
error (%) and median relative error (%), respectively. The upper and lower boundaries represent the upper and lower quartiles of relative
error (%), respectively. The fences are 1.5 times the inter-quartile ranges of the upper/lower quartiles. The red circles denote data that are
not included between the fences. (a) Oct. 3; (b) Oct. 4.
**3.4.2 Sensitivity to the size of the sliding window**
Figure 11 shows the source location errors obtained with different sliding-window sizes. The mean/median error is less than
8% for Oct. 3 and less than 11% for Oct. 4, both of which are smaller than for the various search ranges. This indicates that
the proposed method is more robust to this parameter than to the search range. For both days, the lowest mean/median and
error range occur with relatively large window sizes, i.e. window size of 9 for Oct. 3 and window size of 10 for Oct. 4. This is
because a large window size increases the strength of the filtering and removes the temporal variations in the release rates
more completely. However, a large window size leads to increased computational cost. Because the errors vary in a limited
range, a medium window size provides a better balance between accuracy and computational cost. Feature selection improves
the results for medium and small window sizes, but may have less effect with large window sizes. This tendency implies that
it is more appropriate to apply feature selection with medium window sizes than with large window sizes, as in this study.

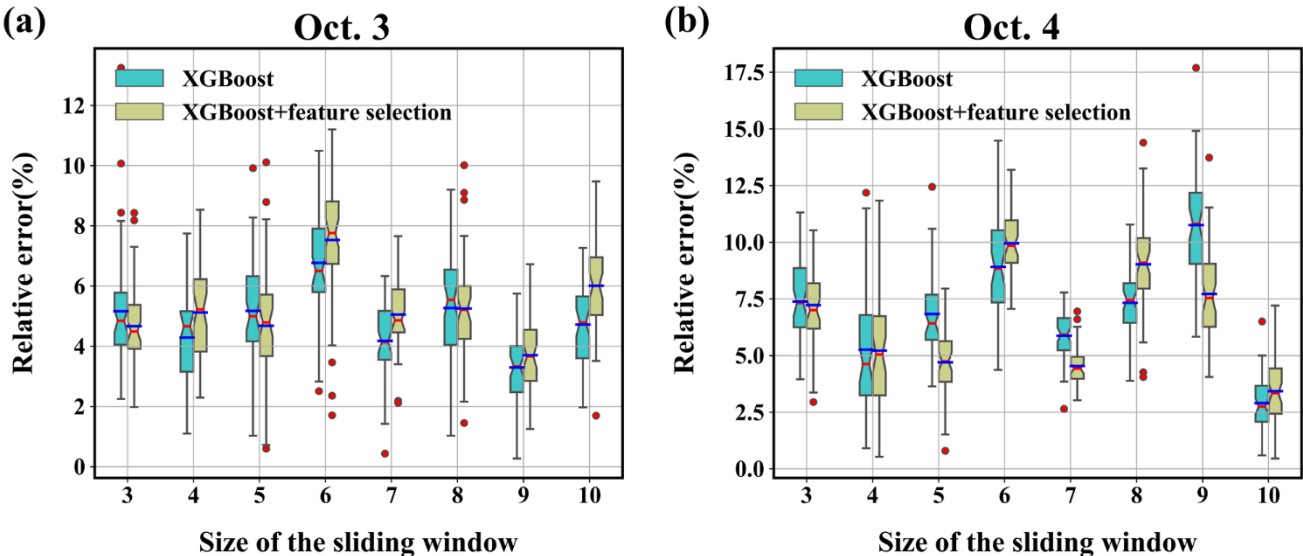


**Figure 11.** Sensitivity to the size of the sliding window. (a) Oct. 3; (b) Oct. 4.
**3.4.3 Sensitivity to the feature type**
Figure 12 compares the results obtained with different feature types. For Oct. 3, the source location errors are quite low when
using only the time-domain features for the reconstruction; indeed, the errors are only slightly larger than when using all the
features. In contrast, the results obtained using only the frequency-domain features exhibit larger errors, indicating that the
time-domain features make a greater contribution to the results for Oct. 3. For Oct. 4, the mean source location errors are
similar when using either the time- or frequency-domain features, but the error range is higher when the frequency-domain
features are used. In addition, the errors of both single-domain-feature results are higher than those of the all-feature results,
indicating that both feature types should be included to ensure accurate and robust source location estimation.

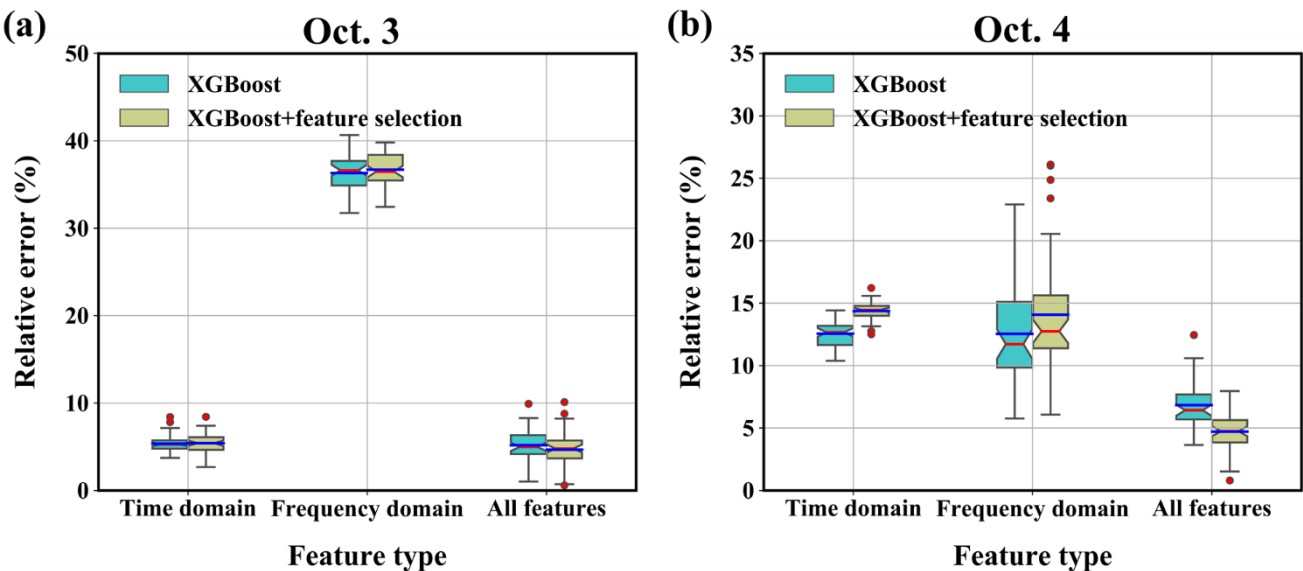

**Figure 12.** Sensitivity to the feature type. (a) Oct. 3; (b) Oct. 4.

### 3.4.4 Sensitivity to the number and combination of observation sites

Figure 13 compares the results obtained with different numbers and combinations of observation sites. The results indicate that the source location error may be more sensitive to the position of the observation site than to the number of sites included. The error level of all-site estimations is relatively low for both days, indicating that increasing the number of observation sites better constrains the solution and help improve the robustness of the model. However, the lowest error levels are achieved by a subset of sites, i.e. Site ABD on Oct. 3 and Site BD on Oct. 4. This is possibly because including all observation sites may cause overfitting and reduce the prediction accuracy. This overfitting can be alleviated by using only representative sites at appropriate position, which capture the environmental variability and provide clear information for locating the source. For Oct.3, multi-site estimations with Site B always produce low error levels, and single-site estimation using Site B also achieves high accuracy. For Oct.4, multi-site estimations with Site BD always achieve relatively low error levels. These results demonstrate the importance of using representative sites for source location estimation. The representative sites (Site B for Oct. 3 and Site BD for Oct. 4) are consistent with the importance calculated in the feature selection step (Fig. 4), preliminarily indicating the potential for feature selection to identify representative sites. In addition, feature selection reduces the mean error level in most cases.

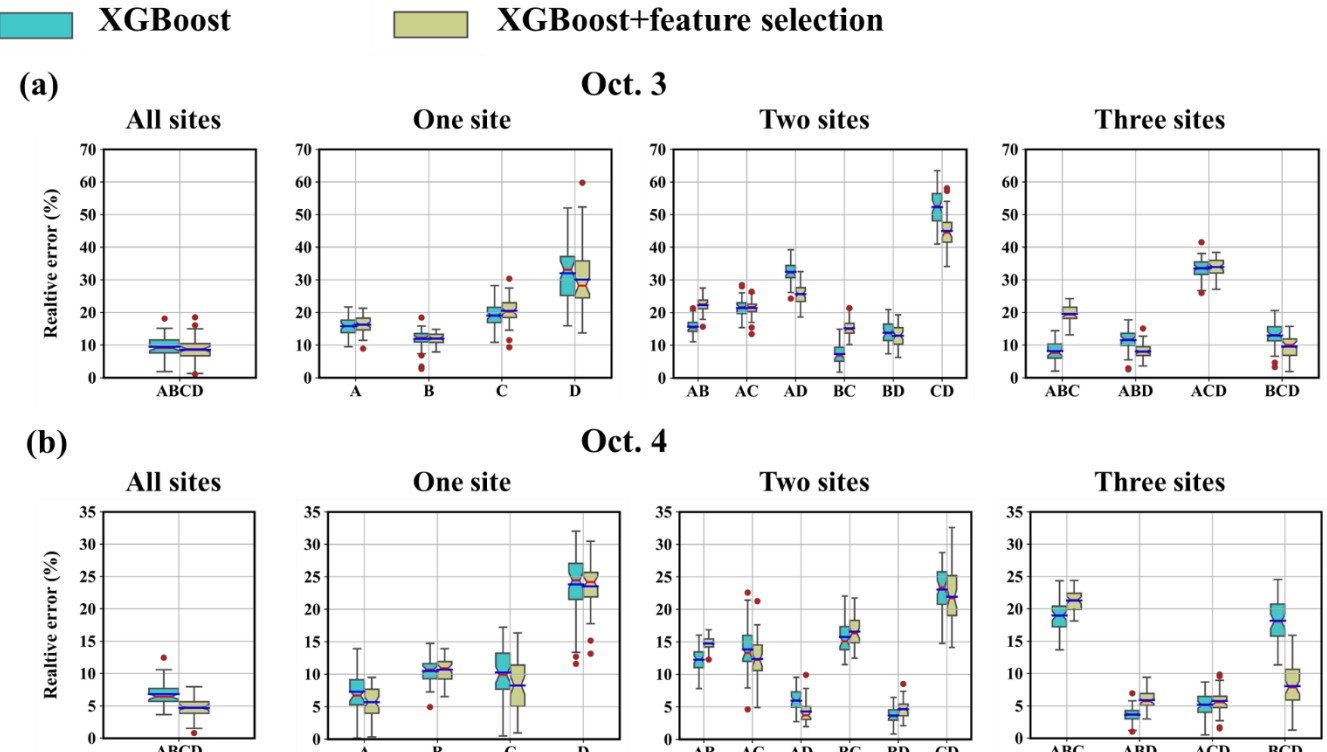

**Figure 13.** Sensitivity to the number and combination of observation sites. (a) Oct. 3; (b) Oct. 4.

### 3.4.5 Sensitivity to the meteorological errors

Figure 14 illustrates the distribution of mean relative source location errors (averaged across 50 groups of hyperparameters) retrieved with 50 perturbed meteorological inputs. For Oct. 3, the estimates generally present a low error level (generally below 10%), and the 50th percentile error level is lower than the error of the unperturbed results (4.68%). In comparison, for Oct. 4, most perturbed results exhibit larger errors (primarily 10%–20%) than the unperturbed result (4.71%), indicating that models for Oct. 4 are more sensitive to the meteorological errors. This sensitivity difference results from the layout of the observation sites (Fig. 2a). The sites on Oct. 3 were almost perpendicular to the prevailing wind direction, capturing the plume under a large range of wind directions. In contrast, the sites on Oct. 4 were basically parallel to the wind direction, capturing the plume only for a very limited range of wind directions. This result indicates the importance of site layout for robust reconstruction in the presence of meteorological errors. Feature selection slightly changes the mean relative error distribution and its percentiles for both days, indicating that meteorological errors may alter the importance of each feature and reduce the effectiveness of feature selection. In addition to meteorological errors, dispersion errors such as wet deposition parameterization (Zhuang et al., 2023) may influence the result, but these errors are not dominant in the two field experiments. The handling of such dispersion errors will be investigated in future work.

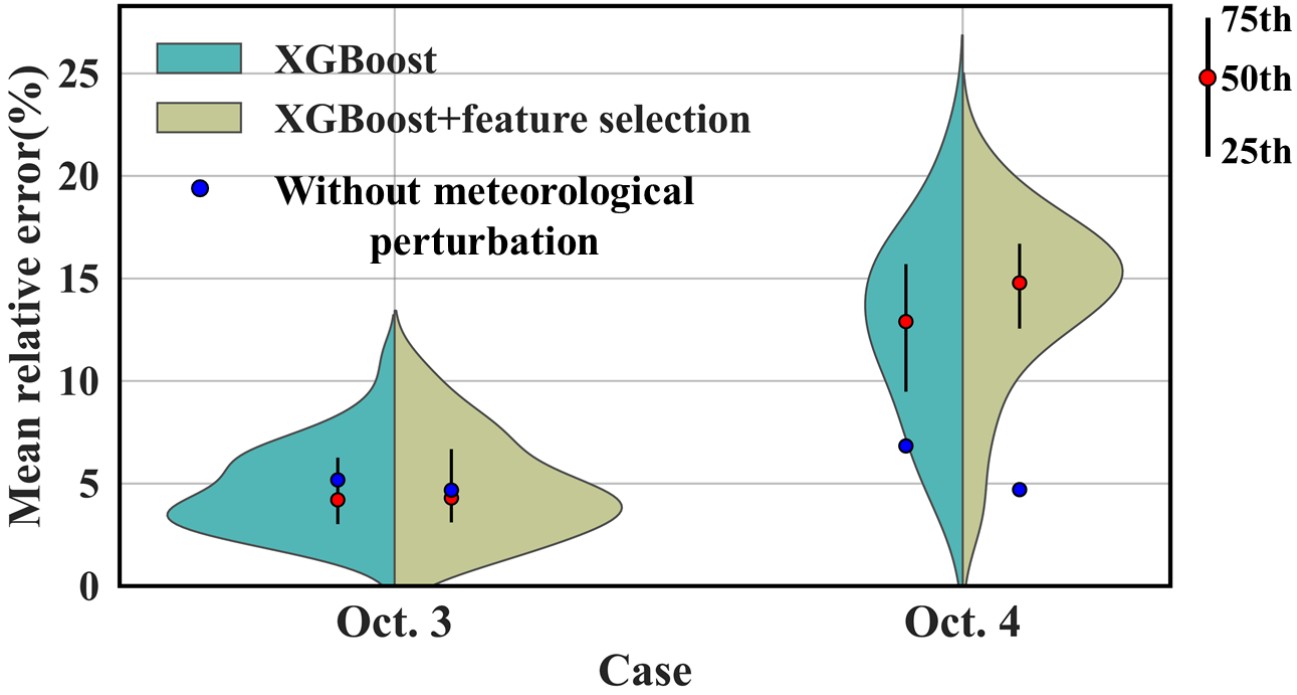

563

**Figure 14.** Sensitivity to the meteorological errors. The violin plots illustrate the kernel density estimation of errors under different meteorological groups for XGBoost models before and after feature selection. The vertical black lines inside the violins depict the interquartile range, capturing the 25th, 50th (red dots), and 75th percentiles of mean relative errors. The blue dots denote the mean relative source location errors for models without meteorological perturbation, as listed in Table 4.

## 4. Conclusions

In this study, we relaxed the unrealistic constant-release assumption of source reconstruction. Instead, we took advantage of the fact that most atmospheric radionuclide releases have a spatially fixed source, and thus the release rate mainly influences the peak values in the temporal observations. Based on this, a more general spatiotemporally separated source reconstruction method was developed to estimate non-constant releases. The separation process was achieved by applying a temporal sliding-window average filter to the observations. This filter reduces the influence of temporal variations in the release rates on the observations, so that the relative spatiotemporal distribution of the filtered observations is dominated by the source location and known meteorology. A response feature vector was extracted to quantify the long-term temporal response trends at each observation site, involving tailored indicators of both the time and frequency domains. The XGBoost algorithm was used to train a machine learning model that links the source location to the feature vector, enabling independent source location estimation without knowing the release rate. With the retrieved source location, the detailed temporal variations of the release rate were determined using the PAMILT algorithm. Validation was performed against the two-day SCK-CEN [41]Ar field experimental data and two groups of ETEX-1 data. The results demonstrate that the proposed method successfully removes

the influence of temporal variations in release rates across observations and accurately reconstructs both the spatial location and temporal variations of the source.

For the local-scale SCK-CEN [41]Ar experiment, source location was reconstructed with lowest errors of only 0.60% (Oct. 3) and 0.80% (Oct. 4), significantly lower than for the correlation-based method and Bayesian method. In terms of the release rate, the PAMILT method reconstructed the temporal variations, peak, and total release with high accuracy, thus avoiding the unrealistic oscillations given by the Bayesian estimate. The proposed method produced smaller uncertainty ranges than the Bayesian method and avoided the multimodal distribution of the Bayesian method. The feature selection process removed the redundant features and reduced the reconstruction errors. For the continental-scale ETEX-1 experiment, the lowest relative source location errors were 0.20% and 0.70% for Group 1 and Group 2, respectively, which were again lower than for the correlation-based and Bayesian methods. The proposed method provides highly accurate mean estimates of the release rate for both groups, although with a large uncertainty range.

Sensitivity analyses on the SCK-CEN [41]Ar experiment revealed that the proposed method exhibits stable source location estimation performance with different parameters and remains effective with only a single observation site, as long as the selected site is appropriately located. Moreover, the proposed method shows robust source location estimation in the presence of meteorological errors, with mean source location error levels below 10%, on condition that the site layout is appropriate.

These results demonstrate that spatiotemporally separated source reconstruction is feasible and achieves satisfactory accuracy in multi-scale release scenarios, thereby providing a promising framework for reconstructing atmospheric radionuclide releases. However, the proposed method does not consider the influence of temporal variations in the release rate on the plume shape. Our future efforts will be directed towards integrating spatial features to further enhance the method.

*Code and data availability*. The code and data for the proposed method can be downloaded from Zenodo (https://doi.org/10.5281/zenodo.10200141). More recent versions of the code and data will be published on GitHub.com (https://github.com/rocket1ab/Source-reconstruction-of-non-constant-atmospheric-radionuclide-releases, last access: 23 November 2023). The implementation is provided in Python, and the instruction file is also available in the provided link.

*Author contributions*. YX conducted the source reconstruction tests and wrote the manuscript draft; SF provided guidance on the RIMPUFF modeling and suggestions on source reconstruction tests; XD and SZ reviewed and edited the manuscript.

*Competing interests*. The authors have declared that they have no conflict of interest.

*Acknowledgements*. This work is supported by the National Natural Science Foundation of China (grant numbers 12275152 and 11875037), LingChuang Research Project of China National Nuclear Corporation, and International Atomic Energy Agency (TC project number CRP9053).

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
