# Peer review of "A spatiotemporally separated framework for reconstructing the"

_Geoscientific Model Development, 2023_

## Author Comment (AC1)

Dear Referee,

We thank the referee for taking the time to provide such a constructive and thorough review of our manuscript (GMD-2023-173). According to the suggestive comments, we have made some modifications to the manuscript, and the responses are listed below. To guide the review process, comments from the referee and original texts in the manuscript are presented in black, our responses are in blue, and any text modifications made to the manuscript are highlighted in red italics. Links are provided below for easy navigation in the document.

**General comments**
**Major points**
**Minor points**
**References**

We are looking forward to your reply.

Best regards,
Yours sincerely
Sheng Fang

**General comments**

The paper presented a source reconstructing procedure by first locating the source location before estimating the emission rates. A machine learning method has been used in the first step to locate the source location. The overall results are quite interesting and encouraging.

However, there are several shortcomings in this manuscript. Some of the statements are not accurate and some terminology uses are also questionable. The presentation of the machine learning method is not easy to follow for those who are not quite familiar with the same method and software. In addition, the dispersion model errors affect the results but are not sufficiently considered or discussed. It is also a concern that the method is only tested with a single set of experimental data. More test cases are probably needed.

**Response to general comments:**

Thank you for your valuable feedback and suggestive comments on our manuscript. We are particularly grateful for your remarks about our results being "quite interesting and encouraging". We have addressed the issues you have raised and revised the manuscript accordingly, which are detailed as follows:

*(1) Inaccurate statements and terminology use:*

Following your comments, we have thoroughly reviewed our manuscript and corrected all statements that were inaccurate or unclear. Particularly, we have elaborated and rephrased the "Spatiotemporally-decoupled" and "constant-release assumption" in the introduction section as "*Spatiotemporally-separated*" and "*Assumptions on the release characteristics*", respectively. Moreover, the title of the revised manuscript has been changed to "*A spatiotemporally-separated framework for reconstructing the source of atmospheric radionuclide releases*" to better describe the current two-step method. More detailed revisions are provided in the response to your specific comments. Regarding the use of terminology, we have provided clear definitions and detailed explanations, such as:

**(1.1) Line 15 of section "Abstract": "source localization"**

The term "source localization" has been replaced with "source location estimation" in the revised manuscript. For example,

► Line 15 of section "Abstract" has been replaced with:

[revised manuscript text omitted]

**(1.3) Line 108 of section "2.1 Source reconstruction models": "$A(r) = [A_1(r), A_2(r), \cdots, A_N(r)]^T \in \mathbb{R}^{N \times N}$"**

The matrix $\mathbf{A}(\mathbf{r})$ is not a square matrix in general. We have modified the dimension of the matrix: $\mathbf{A}(\mathbf{r}) =$

$[A_1(\mathbf{r}),\ A_2(\mathbf{r}),\cdots,A_N(\mathbf{r})]^T \in \mathbb{R}^{N\times S}$, where $N$ is the number of sequential time steps and $S$ is the length of release rate vector $\mathbf{q}$.

►     Line 100-109 of section "2.1 Source reconstruction models" have been replaced with:

"For an atmospheric radionuclide release, Eq. (1) relates the observations at each observation site to the source parameters:

$$\boldsymbol{\mu} = \mathbf{F}(\mathbf{r},\mathbf{q}) + \boldsymbol{\varepsilon}\,, \tag{1}$$

where $\boldsymbol{\mu} = [\mu_1,\mu_2,\cdots,\mu_N] \in \mathbb{R}^N$ is an observation vector composed of observations at $N$ sequential time steps, the function $\mathbf{F}$ maps the source parameters to the observations, i.e. an atmospheric dispersion model, $\mathbf{r}$ refers to the source location, $q \in \mathbb{R}^S$ is the temporally varying release rate, and $\boldsymbol{\varepsilon} \in \mathbb{R}^N$ is a vector containing both model and measurement errors.

In most source reconstruction models, $\mathbf{F}$ is simplified to the product of $\mathbf{q}$ and a source–receptor matrix $\mathbf{A}$ that depends on the source location:

$$\boldsymbol{\mu} = \mathbf{A}(\mathbf{r})\mathbf{q} + \boldsymbol{\varepsilon}\,, \tag{2}$$

where $A(r) = [A_1(r),\ A_2(r),\cdots,A_N(r)]^T \in \mathbb{R}^{N\times S}$ and each row describes the sensitivity of an observation to the release rate $\mathbf{q}$ given the source location $\mathbf{r}$."

**(1.4) Line 131 of section "2.2 Spatiotemporal decoupling": "sliding window"**

As outlined in Eq. (4), a one-sided window is employed. This one-sided temporal sliding-window average filter involves the current and previous observations in the window, acknowledging that future observations are not available for filtering in practice. Compared to the centered window, the one-sided window excels in real-time data processing and rapid response to changes in the observations, making it more suitable for real applications.

►     Line 128-132 of section "2.2 Spatiotemporal decoupling" have been replaced with:

"In this study, the following operator matrix is constructed to impose a *one-sided* temporal sliding-window average filter (Eamonn Keogh, Selina Chu, 2004):

$$\mathbf{P} = \frac{1}{T}\begin{bmatrix} 1 & & & & & & \\ 1 & 1 & & & & & \\ & \vdots & & & & & \\ 1 & 1 & \cdots & 1 & & & \\ 1 & 1 & \cdots & 1 & 1 & & \\ & 1 & 1 & \cdots & 1 & 1 & \\ & & 1 & 1 & \cdots & 1 & 1 \\ & & & \ddots & \ddots & \ddots & \ddots & \ddots \\ & & & & 1 & 1 & \cdots & 1 & 1 \\ & & & & 1 & 1 & 1 & 1 & 1 \end{bmatrix}, \tag{4}$$

where $T$ is the size of the sliding window. *This one-sided filter involves the current and previous observations in the window, acknowledging that future observations are not available for filtering in practice.* Although a sliding-window average filter is used in this study, Eq. (3) is compatible with more advanced processing methods."

**(1.5) Line 228 of section "2.6.2 Simulation settings of atmospheric dispersion model": "sample"**

This term refers to an individual simulation using one of the candidate source locations. Therefore, each "sample" represents a simulated dispersion scenario with a different candidate source location. To clarify this point, we have replaced the term "sample" with "simulation" in the revised manuscript and explained the meaning of "sample".

►     Line 228-230 of section "2.6.2 Simulation settings of atmospheric dispersion model" have been replaced with:

"To establish the datasets for the XGBoost model, *2000 simulations and 1000 simulations with different source locations*

*were performed* by RIMPUFF for Oct. 3 and Oct. 4, respectively. The *candidate* source locations were *randomly* sampled from the shaded zones in Fig. 2a, which were determined according to the positions of the observation sites and the upwind direction. *Each simulation, along with its corresponding source location, forms one sample.*"

*(2) Presentation of the machine learning method:*

We have added descriptions to the figure caption of the Fig. 1 and relabeled the root nodes using yellow squares in Fig. 1, providing a more accurate and detailed introduction to the decision tree model. To avoid confusion, we have replaced the symbol "$T$" with "$M$" in Eq. (7) and have provided additional descriptions for all the parameters.

► Line 147-157 of section "2.3 Source localization without knowing the exact release rates" have been replaced with: "where $K$ is the number of trees, $\mathcal{F} = \{f(x) = \omega_{Q(x)}\}(Q: \mathbb{R}^p \rightarrow M, \omega \in \mathbb{R}^M)$ is the space of decision trees, and $Q$ represents the structure of each tree, mapping the feature vector to $M$ leaf nodes. Each $f_k$ corresponds to an independent tree structure $Q$ with leaf node weight $\omega = (\omega_1, \omega_2, \cdots, \omega_M)$. Equation (5) is then used to predict $\hat{\mathbf{r}}_i = (\hat{x}_i, \hat{y}_i)$ for the $i$-th sample.

[Figure]

**Figure 1.** Flowchart of XGBoost for predicting $\hat{\mathbf{r}}_i$ based on decision tree model. *The yellow squares are the root nodes within each tree, representing the input features in this paper. The purple ellipses denote the child nodes where the model evaluates input features and make decisions to split the data. The green rectangles depict the leaf nodes and refer to the prediction results. The vertical rectangles abstract the internal splitting processes of the trees, indicating decision-making not explicitly detailed in the diagram.*

XGBoost trains $G(\mathbf{X})$ in Eq. (5) by continuously fitting the residual error until the following objective function is minimized:

$$Obj^{(t)} = \sum_{i=1}^{n} \left( \mathbf{r}_i - \left( \hat{\mathbf{r}}_i^{(t-1)} + f_t(\mathbf{X}_i) \right) \right)^2 + \sum_{i=1}^{t} \Omega(f_i), \tag{6}$$

where $t$ represents the training of the $t$-th tree and $\Omega(f_i)$ is the regularization term, given by:

$$\Omega(f) = \Upsilon M + \frac{1}{2}\lambda \sum_{j=1}^{M} \omega_j^2, \tag{7}$$

*where $M$ is the number of leaf nodes, $\omega_j$ is the leaf node weight for the j-th leaf node, $\Upsilon$ and $\lambda$ are penalty coefficients.* The minimization of Eq. (6) provides the parametric model $G(\mathbf{X})$ that maps the feature ensemble $\mathbf{X}$ extracted from $\mathbf{\mu}_p$ to the source location $\mathbf{r}$."

*(3)  Dispersion model error considerations:*

We agree with your point regarding the impact of dispersion model errors. To address this issue, we will add an in-depth analysis of the dispersion model errors and their influences on our source reconstruction. Perturbation will be added to the input parameters (such as the wind speed and atmospheric stability) of atmospheric dispersion simulations, to simulate the potential dispersion model error. The proposed method will use these perturbed simulations to train the XGBoost model and to reconstruct the source. The error statistics of the reconstructed source will be discussed both qualitatively and quantitatively. Because these simulations are time-consuming, we will provide the results in the revised manuscript.

*(4)  The testing of the method:*

To address your concern, we have added another validation based on the first release of the European Tracer Experiment (ETEX-1) (Nodop et al., 1998), which is continental scale. We have provided an overview of the ETEX-1 experiment and source reconstruction results below. Detailed results and discussions have been included in the revised manuscript.

**2.6 Multi-scale Validations**

**2.6.1 Field experiments**

The ETEX-1 experiment took place at Monterfil in Brittany, France, on 23 October 1994 (Nodop et al., 1998). During ETEX-1, a total release of 340 kg of PMCH was released into the atmosphere during 23 Oct 1994 16:00:00 UTC and 24 Oct 1994 03:50:00 UTC. As illustrated in Fig. 2b, the source coordinates were (2.0083°W, 48.058°N). A total of 3104 available observations (3h-averaged concentrations) were collected by 168 sites. ETEX-1 has been commonly used as a validation scenario for reconstructing atmospheric radionuclide releases (Ulimoen and Klein, 2023; Tomas et al., 2021). The candidate source locations are uniformly sampled from the green shaded zone. We choose two groups of observation sites: the first comprises four sites (i.e. B05, D10, D16, F02) randomly selected from the sites within the sample zone (Group1, with a total of 92 available observations), and the second involves four sites (i.e. CR02, D15, DK08, S09) randomly selected from the sites beyond the sample zone boundaries (Group2, with a total of 90 available observations). Compared to the SCK-CEN [41]Ar experiment, the observations of the ETEX-1 exhibit temporal sparsity, lower temporal resolution, and increased complexity in meteorological conditions.

[Figure]

**Figure 2.** Release location and observation sites of two filed experiments. (a) SCK-CEN [41]Ar experiment. The map was created based on the relative positions of the release source and observation sites, as detailed in (Drews et al., 2002). The coordinate of the sample border is (500m, -200m) and (1180m, 580m) on Oct. 3, and (450m, 10m) and (850m, 450m) on Oct. 4. It was plotted using MATLAB 2016b, instead of created by a map provider; (b) ETEX-1 experiment. The map was created based on the real longitudes and latitudes of the release source and observation sites, as detailed in (Nodop et al., 1998). The coordinate of the sample border is (10°W, 40°N) and (10°E, 60°N). It was plotted using the function *cartopy* of Python, instead of created by a map provider.

**2.6.2 Simulation settings of atmospheric dispersion model**

For the ETEX-1 experiment, the FLEXible PARTicle (FLEXPART) model (version 10.4) was applied to simulate the dispersion of PMCH (Pisso et al., 2019). The meteorological data were from the United States National Centers of Environmental Prediction Climate Forecast System Reanalysis, which has a spatial resolution of 0.5°×0.5° and time resolution of 6h. To rapidly establish the relationship between the varying source locations and the observations, 182 backward simulations were performed using FLEXPART. The candidate source locations were uniformly sampled from the shaded zone in Fig. 2b, resulting a total of 6561 source locations. As described in Sect. 2.5.1, 2624 candidate source locations are preserved by pre-screening step. The constant factors mentioned in Sect. 2.5.2 are $5.60\times10^{12}$ and $2.86\times10^{13}$.

**3.2 Optimization of XGBoost model**
**3.2.2 Feature selection**

Figure 4 compares the importance of the selected features at each site. For the ETEX-1 experiment, Fig. 4c and d

showed that the features of Group1 and Group2 are almost reserved after the feature selection process (only one feature is removed for each case), indicating fewer redundant features than that in the SCK-CEN $^{41}$Ar experiment. The time-domain features are also dominant, but the frequency-domain features at some sites also play important roles such as D16 and S09. The MCVs of the ETEX-1 experiment have similar variation trend as that of the SCK-CEN $^{41}$Ar experiment (Fig. S4c and d). However, the Group1 has obviously lower MCV than other experiments.

(a)

[Figure]

(b)

[Figure]

[Figure]

**Figure 4.** Feature importance. (a) Oct. 3; (b) Oct. 4; (c) Group1; (d) Group2.

[Figure]

**Figure S4.** Results of feature selection in x (longitude) and y (latitude) directions. (a) Oct. 3; (b) Oct. 4; (c) Group1; (d) Group2. The black stars denote the optimal number of features. The table inserted in each subgraph lists the selected features for each observation site.

**3.3 Source reconstruction**
**3.3.1 Source location estimation**
Figure 5 compares the best-estimated source locations of the correlation-based method, the Bayesian method, and the proposed method with the ground truth. For the ETEX-1 experiment, the pre-screening zone also covers the true source location for Group1 and Group2. But source locations estimated by the correlation-based method are 411.85 km and 486.41 km away from the ground truth for Group1 and Group2, respectively. The location error of the Bayesian method estimates is only 30.50 km for Group1, but it increases to 520.77 km for Group2, indicating its sensitivity to the observations. In contrary, the proposed method achieves the lowest source location errors, which are below 10 km and 20 km for Group1 and Group2, respectively. Because the feature selection hardly removed the features (Fig. 4c and d), the estimated source locations with and without feature selection basically overlap for both groups.

[Figure]

**Figure 5.** Source location estimation results. The yellow dots denote the maximum correlation points, which are the localization results of the correlation-based method. The green and red stars represent the localization results based on

XGBoost before and after feature selection, respectively. The cyan diamonds represent the localization results based on the Bayesian method. (a) Oct. 3; (b) Oct. 4; (c) Group1; (d) Group2. A detailed enlargement of the region around (2.5°W, 47.5°N) to (1.5°W, 48.5°N) is shown in the bottom right corner in (c) and (d) to highlight the source location estimation results of the proposed methods.

**3.3.2 Release rates**

Figure 6 displays the release rates estimated by the Bayesian and PAMILT methods based on the source location estimates in Fig. 5. For the two ETEX-1 groups (Fig. 6c and d), Bayesian estimates exhibit noticeable fluctuations, leading to underestimations–58.11% for Group1 and 51.44% for Group2. Furthermore, the temporal profile of the Bayesian estimates for Group2 completely falls out of the true release window. In contrast, most releases of the PAMILT estimates are within the true release time window, especially for Group2, despite overestimations reach 52.38% for Group1 and 57.65% for Group2, after feature selection process. Compared to the SCK-CEN [41]Ar experiment, the increased deviation in the ETEX-1 experiment may be attributed to the limited observations at the four sites as shown in Fig. S3.

[Figure]

**Figure 6.** Release rate estimation results with different location estimates. (a) Oct. 3; (b) Oct. 4; (c) Group1; (d) Group2. The release rates labelled by XGBoost or XGBoost+feature selection are estimated using PAMILT method.

[Figure]

**Figure S3.** Observations before and after filtering at observation sites. (a) Oct. 3; (b) Oct. 4; (c) Group1; (d) Group2.

**3.3.3 Uncertainty range**

Figure 7 compares the spatial distribution of 50 estimates produced by the different source location estimation methods. For the ETEX-1 experiment, the estimates of the correlation-based method are quite spread, whereas those of the Bayesian method are more concentrated. The Bayesian estimates are close to the truth in Group1, but are noticeably deviated in Group2. The phenomenon indicates that the Bayesian method is sensitive to the observations, especially when observations are limited. Fig. S5c and d also reveals that the Bayesian-estimated posterior distribution is multimodal for both ETEX-1 groups, which can be avoided by using additional observations (Fig. S5e). In contrast, the

proposed method provides estimates quite concentrated around the truth for both Group1 and Group2, indicating its efficiency in the case of limited observations. Compared to Group2, Group1 exhibits lower source location error. This is attributed to the fact that, as shown in Fig. 2, Group1's four observation sites are situated closer to the sampled source locations than those of Group2, thereby providing more accurate plume characteristics and resulting in lower dispersion error.

[Figure]

**Figure 7.** Distribution of source location estimation results over 50 calculations. (a) Oct. 3; (b) Oct. 4; (c) Group1; (d) Group2.

[Figure]

**Figure S5.** Posterior distributions of source location parameters. (a) Oct. 3; (b) Oct. 4; (c) Group1; (d) Group2; (e) ETEX-1 (all observations in ETEX-1 are used). The black solid lines denote the true location parameters and the dashed lines denote the mean estimates of all posterior samples.

Figure 9 compares the uncertainty ranges of the release rate estimates in the two ETEX-1 groups. For both groups, the Bayesian estimates show underestimations (including the mean estimate), but it presents small uncertainty range (Fig. 9a and c). In addition, the Bayesian estimates all fall out of the time window of the true release in Group2 (Fig.9c). The mean PAMILT estimates are more accurate mean Bayesian estimates, which most releases are within the time window of the true release (Fig. 9b and d). However, PAMILT estimates for the ETEX-1 experiment show a large uncertainty range in both groups, in contrast to the SCK-CEN [41]Ar experiment. The reason for this discrepancy is that the source-receptor matrices of the ETEX-1 experiment are more sensitive to errors in source location than those of the SCK-CEN [41]Ar experiment. The greater sensitivity originates from the ETEX-1 experiment's more complex meteorological conditions, both spatially and temporally. As for the mean total release, the Bayesian method shows underestimations of 70.93% for Group1 and 74.15% for Group2. In comparison, the proposed method shows deviation of only 0.71% for Group1 and 0.09% for Group2, after feature selection.

[Figure]

**Figure 9.** Release rate estimates over 50 calculations in ETEX-1 experiment. (a) Group1-Bayesian method; (b) Group1-PAMILT method; (c) Group2-Bayesian method; (d) Group2-PAMILT method.

Table 3 lists the mean and standard deviation of the relative errors for the 50 estimates given by different methods. For the ETEX-1 experiment, the Bayesian method shows case-sensitive performances with respect to the mean relative error of source location estimation, whereas the proposed method gives the most accurate source locations with small uncertainties for both groups. As for the total release, the proposed methods show smaller mean relative errors than the Bayesian methods, but the Bayesian method shows a smaller standard deviation.

**Table 3.** Relative errors of source reconstruction. $\delta_{\mathbf{r}}$ represents the relative error of source location, which is positive and $\delta_Q$ denotes the relative error of total release, where a positive value indicates overestimation and a negative value denotes underestimation.

| Experiments | Statistical parameters (Relative error) | | Correlation-based method | Bayesian method | The proposed method | |
|---|---|---|---|---|---|---|
| | | | | | XGBoost | XGBoost+ feature selection |
| Oct. 3 | $\delta_{\mathbf{r}}$ | Mean | 14.10% | 11.88% | 5.18% | 4.68% |
| | | Std | 11.37% | 7.53% | 1.79% | 2.05% |
| | $\delta_Q$ | Mean | - | 153.61% | -16.93% | -18.30% |
| | | Std | - | 189.76% | 9.45% | 8.01% |
| Oct. 4 | $\delta_{\mathbf{r}}$ | Mean | 14.30% | 12.83% | 6.83% | 4.71% |
| | | Std | 9.60% | 1.68% | 1.76% | 1.53% |
| | $\delta_Q$ | Mean | - | 42.29% | -54.12% | -47.42% |
| | | Std | - | 15.05% | 6.47% | 5.85% |
| Group1 | $\delta_{\mathbf{r}}$ | Mean | 16.95% | 3.22% | 2.32% | 2.42% |
| | | Std | 7.46% | 2.75% | 1.43% | 1.43% |
| | $\delta_Q$ | Mean | - | -70.93% | 18.12% | -0.71% |
| | | Std | - | 17.87% | 99.85% | 102.01% |
| Group2 | $\delta_{\mathbf{r}}$ | Mean | 21.9% | 23.97% | 5.21% | 4.97% |
| | | Std | 5.05% | 1.97% | 2.42% | 2.35% |
| | $\delta_Q$ | Mean | - | -74.15% | 16.67% | 0.09% |
| | | Std | - | 11.68% | 93.50% | 109.56% |

**Major points**

**Comment#1:**

Title: Both "Generalized" and "Spatiotemporally-decoupled" are not accurately reflecting the current two-step method. The word "non-constant" in the title does not sound appropriate either. In reality, there are rarely constant releases. The author should reconsider the title.

**Response to comment#1:**

Thank you very much for the comment on the title. We have deleted the "non-constant" and "Generalized" in the title and have replaced the term "Spatiotemporally-decoupled" with "*Spatiotemporally-separated*", to more accurately reflect the essence of the two-step method described in the manuscript. The revised title now reads: "*A spatiotemporally-separated framework for reconstructing the source of atmospheric radionuclide releases.*"

**Comment#2:**

Abstract, lines 12-14: This statement is not accurate. The temporal variation of the release rates may be reflected on the plume shape, not only on the temporal variations of the observations. In theory, some problems cannot be decoupled. So the proposed method cannot be a real general framework. The limitation of the method has to be pointed out in the paper.

**Response to comment#2:**

*(1) Regarding the influence of the temporal variation of the release rates:*

We agree that the temporal variation of release rates influence the plume shape. This influence may be difficult to capture using only a limited number of observation sites, which is the case of SCK-CEN [41]Ar experiment. For this reason, we focus on reducing the influence of temporal variations in the release rate on the observations, whereas the influence on the plume shape is not directly considered. However, our future efforts will be directed towards integrating spatial features to further enhance the method. The limitation and the future efforts have been pointed out in the section "4. Conclusions" after Line 475:

"*However, the proposed method does not consider the influence of temporal variations of the release rate on the plume shape. Our future efforts will be directed towards integrating spatial features to further enhance the method.*"

*(2) Concerning whether the proposed method is truly a general framework:*

You raised an essential point about theoretical constraints where some problems cannot be decoupled. For these problems, our goal is to minimize the influence of temporal variations in the release rate on the observations, so that we can achieve spatiotemporally-separated reconstruction. To eliminate ambiguity, we have restated the characteristics of the proposed method, using the term "*spatiotemporally-separated*" rather than "spatiotemporally-decoupled" in the revised manuscript. Furthermore, to verify the applicability of the proposed method, we have also validated it using another field experiment at a different spatial scale, which have been presented in the responses to the **General comments**. This will help readers better understand the limitations and superiority of the proposed method and encourage further researches to overcome the constraints. Followed by comment#1 and comment#2, we have revised the abstract to ensure that it accurately reflects the updated focus of our study. Additionally, several relevant titles have been updated to ensure consistency with these modifications.

► Line 9-14 of section "Abstract" have been replaced with:

"Determining the source location and release rate are critical in assessing the environmental consequences of atmospheric radionuclide releases, but remain challenging because of the huge multi-dimensional solution space. We

propose a *spatiotemporally-separated* two-step framework to reduce the dimension of the solution space in each step and improves the *source* reconstruction accuracy of *radionuclide* releases. The *separating* process is conducted by applying a temporal sliding-window average filter to the observations, thereby reducing the influence of temporal variations in the release rate *on the observations* and ensuring that the features of the filtered data are dominated by the source location."

► Title "2.2 Spatiotemporal decoupling" in line 110 has been replaced with:

"2.2 *Observation filtering for spatiotemporally-separated reconstruction*"

► Title "2.8.1 Decoupling" in line 255 has been replaced with:

"2.8.1 *Observation filtering*"

► Title "3.1 Decoupling performance" in line 291 has been replaced with:

"3.1 *Filtering performance*"

**Comment#3:**

Abstract, line 15: Locating a source location is not "localization". This needs to be corrected throughout the paper.

**Response to comment#3:**

We appreciate your comment on the use of the term "localization". We have replaced the relevant descriptions with "source location estimation", which may precisely describe the process in our research. Accordingly, we have diligently revised the term throughout the paper to ensure accuracy and consistency. For example,

► Line 15 of section "Abstract" has been replaced with:

"A machine learning model is trained to link these features to the source location, enabling independent *source location estimation*."

► Line 94-97 of section "1. Introduction" has been replaced with:

"The performance of the proposed method is compared with the correlation-based method for *source location estimation* and the Bayesian method for spatiotemporal accuracy. The sensitivity of the *source location estimation* to the spatial search range, size of the sliding window, feature type, and number and combination of sites is also investigated for SCK-CEN $^{41}$Ar experiment."

**Comment#4:**

Abstract, line 18: A relative error of about 50% for the Oct. 4 total release is probably not deemed "accurate". It is better to present the results more objectively with the actual number listed in Table 3.

**Response to comment#4:**

Thank you for your valuable feedback. In light of your suggestion, we have modified the abstract to more objectively reflect the results.

► Line 16-19 of section "Abstract" has been changed from:

"Validation using SCK-CEN $^{41}$Ar experimental data demonstrates that the localization error is less than 1%, and the temporal variations, peak release rate, and total release are reconstructed accurately. The proposed method exhibits higher accuracy and a smaller uncertainty range than the correlation-based and Bayesian methods."

to:

"*In applying the method to the SCK-CEN $^{41}$Ar experiment, the lowest relative source location errors are only 0.60% and 0.80% for Oct. 3 and Oct. 4, respectively. The proposed method demonstrates higher accuracy and a smaller uncertainty range than the correlation-based and Bayesian methods in estimating source location. Regarding release rate estimation,*

*the temporal variations are accurately reconstructed. Compared to the Bayesian method, the mean total release estimate is improved, showing an average 65.09% reduction in relative error.*"

**Comment#5:**

Line 94: The authors seem to suggest that the correlation-based method only applies when constant-release assumption is made. This is not accurate. Constant release is only one assumption that reduces the complexity of the problem. If the release starting time or duration is not known. Such assumption may not be enough to guarantee a unique solution of the source location. On the other hand, if a source is not constant, but the release time period and temporal profile are known, it is probably easy to get the source location even without the constant-release assumption.

**Response to comment#5:**

We agree with you that constant release is only one assumption that reduces the complexity of the problem and our descriptions need to be improved. To avoid confusion, we have deleted the term "(constant-release assumption)" in Line 94 in the revised manuscript. In the introduction section, we have emphasized that: the constant-release assumption may lead to inaccurate source location estimation, such as the case of the correlation-based method, because the constant-release assumption ignores the interaction between the time-varying release characteristics and non-stationary meteorological fields. We also agree that the source location can easily be estimated, if the release time period and temporal profile are known, even without the constant-release assumption. We did not mention this scenario, because this is not the focus of our study. Instead, we mainly consider atmospheric radionuclide releases where both the release time period and the temporal profile are unknown, which presents a more complex challenge for source location estimation.

► Line 60-78 of section "1. Introduction" has been replaced with:

"*Assumptions on the release characteristics aim to reduce the dimension of the solution space to 4 or 5, i.e. the two source location coordinates, the total release, and the release time (or the release start and end time), by assuming that the substances are released instantaneously at a release time (or constantly during a release time period) (Kovalets et al., 2020, 2018; Efthimiou et al., 2018, 2017; Tomas et al., 2021; Andronopoulos and Kovalets, 2021; Ma et al., 2018). Under these assumptions, the correlation-based method has exhibited high accuracy for ideal cases under stationary meteorological conditions, such as synthetic simulation experiments (Ma et al., 2018) and wind tunnel experiments (Kovalets et al., 2018; Efthimiou et al., 2017). However, previous studies have also demonstrated that the application in real-world cases may be much more challenging, (Kovalets et al., 2020; Tomas et al., 2021; Andronopoulos and Kovalets, 2021; Becker et al., 2007), because the release usually exhibits temporal variations and may experience non-stationary meteorological fields. The interaction between the time-varying release characteristics and non-stationary meteorological fields is ignored in the instantaneous-release and constant-release assumption, leading to inaccurate reconstruction.*"

► Line 94-95 of section "1. Introduction" has been replaced with:

"The performance of the proposed method is compared with the correlation-based method  for source location estimation and the Bayesian method  for spatiotemporal accuracy."

**Comment#6:**

Line 108: It is wrong to assume a square matrix. The dimensions of the observation and source vectors are independent and rarely the same.

**Response to comment#6:**

Thank you for pointing the error of the matrix. As you mentioned, the matrix $\mathbf{A}(\mathbf{r})$ is not a square matrix in general. We have modified the dimension of the matrix: $\mathbf{A}(\mathbf{r}) = [A_1(\mathbf{r}), A_2(\mathbf{r}), \cdots, A_N(\mathbf{r})]^T \in \mathbb{R}^{N \times S}$, where $N$ is the number of sequential time steps and $S$ is the length of release rate vector $\mathbf{q}$.

► Line 100-109 of section "2.1 Source reconstruction models" have been replaced with:

"For an atmospheric radionuclide release, Eq. (1) relates the observations at each observation site to the source parameters:

$$\boldsymbol{\mu} = \mathbf{F}(\mathbf{r}, \mathbf{q}) + \boldsymbol{\varepsilon} , \qquad (1)$$

where $\boldsymbol{\mu} = [\mu_1, \mu_2, \cdots, \mu_N] \in \mathbb{R}^N$ is an observation vector composed of observations at $N$ sequential time steps, the function $\mathbf{F}$ maps the source parameters to the observations, i.e. an atmospheric dispersion model, $\mathbf{r}$ refers to the source location, $q \in \mathbb{R}^S$ is the temporally varying release rate, and $\boldsymbol{\varepsilon} \in \mathbb{R}^N$ is a vector containing both model and measurement errors.

In most source reconstruction models, $\mathbf{F}$ is simplified to the product of $\mathbf{q}$ and a source–receptor matrix $\mathbf{A}$ that depends on the source location:

$$\boldsymbol{\mu} = \mathbf{A}(\mathbf{r})\mathbf{q} + \boldsymbol{\varepsilon} , \qquad (2)$$

where $A(r) = [A_1(r), A_2(r), \cdots, A_N(r)]^T \in \mathbb{R}^{N \times S}$ and each row describes the sensitivity of an observation to the release rate $\mathbf{q}$ given the source location $\mathbf{r}$."

**Comment#7:**

Lines 122-124: The statement is not correct. The emissions combined with the meteorological conditions together determine the concentrations at any given measurement site, including the peak values and its timing.

**Response to comment#7:**

We agree that the emission and the meteorological jointly influence the peak values and its timing. Our work aims to smooth out the peak observations that is mainly shaped by the temporal release profile. We have restated our method based on this effect.

► Line 122-124 of section "2.2 Spatiotemporal decoupling" have been replaced with:

"With a fixed source location, *the release rate and meteorology jointly determine the temporal variation of the observations (Li et al., 2019). The influence of meteorology can be pre-calculated as source-receptor sensitivities and subsequently stored in matrix $A(r)$*."

► Line 134-136 of section "2.3 Source localization without knowing the exact release rates" have been replaced with:

"After applying the filter in Eq. (4), *those peak observations that are mainly shaped by the temporal release profile, are smoothed out,* but the influences of the source position and meteorology remain relatively unchanged, as they determine the long-term temporal trends of observations and are less affected by the filter."

**Comment#8:**

Line 228: It is very confusing to use "sample" for the different candidate source locations.

**Response to comment#8:**

We apologize for any confusion caused by the term "sample". This term refers to an individual simulation using one of the candidate source locations. Therefore, each "sample" represents a simulated dispersion scenario with a different candidate source location. To clarify this point, we have replaced the word "sample" with "simulation" in the revised manuscript.

► Line 228-230 of section "2.6.2 Simulation settings of atmospheric dispersion model" have been replaced with:

"To establish the datasets for the XGBoost model, *2000 simulations and 1000 simulations with different source locations*

*were performed* by RIMPUFF for Oct. 3 and Oct. 4, respectively. The *candidate* source locations were *randomly* sampled from the shaded zones in Fig. 2a, which were determined according to the positions of the observation sites and the upwind direction. *Each simulation, along with its corresponding source location, forms one sample.*"

**Comment#9**

Lines 287-288: The hyper-parameters used in the 50 runs should be given in the supplementary document.

**Response to comment#9:**

We appreciate your suggestion regarding the inclusion of hyper-parameters. We have provided all hyper-parameters of the 50 runs in the revised version of the supplementary document.

**Comment#10:**

Figure S3: What is the sliding window applied here? It does seem to be a sided window rather than centered one. Please explain this in the paper.

**Response to comment#10:**

We apologize for the lack of clarity regarding the sliding window used in Figure S3. As outlined in Eq. (4), a one-sided window is employed. This one-sided temporal sliding-window average filter involves the current and previous observations in the window, acknowledging that future observations are not available for filtering in practice. Compared to the centered window, the one-sided window excels in real-time data processing and rapid response to changes in the observations, making it more suitable for real applications. Relevant descriptions of the sliding window have been modified in the revised manuscript for clarity.

► Line 128-132 of section "2.2 Spatiotemporal decoupling" have been replaced with:

"In this study, the following operator matrix is constructed to impose a *one-sided* temporal sliding-window average filter (Eamonn Keogh, Selina Chu, 2004):

$$
\mathbf{P} = \frac{1}{T}
\begin{bmatrix}
1 & & & & & & & & & \\
1 & 1 & & & & & & & & \\
& \vdots & & & & & & & & \\
1 & 1 & \cdots & 1 & & & & & & \\
1 & 1 & \cdots & 1 & 1 & & & & & \\
& 1 & 1 & \cdots & 1 & 1 & & & & \\
& & 1 & 1 & \cdots & 1 & 1 & & & \\
& & & \ddots & \ddots & \ddots & \ddots & \ddots & & \\
& & & & 1 & 1 & \cdots & 1 & 1 & \\
& & & & 1 & 1 & 1 & 1 & 1 &
\end{bmatrix}, \tag{4}
$$

where $T$ is the size of the sliding window. *This one-sided filter involves the current and previous observations in the window, acknowledging that future observations are not available for filtering in practice.* Although a sliding-window average filter is used in this study, Eq. (3) is compatible with more advanced processing methods."

**Minor points**

**Comment#1:**

Line 46, T3-10: Please explain what T3-10 distributions are.

**Response to comment#1:**

The notation T3-10 denotes a Student's *t*-distribution with degrees of freedom ranging from 3 to 10, as referenced in

(Wang et al., 2017). The *t*-distribution (also known as $t_v$) is applicable for estimating the mean of a normally distributed population, when the sample size is small and the population standard deviation is unknown. In this distribution, the parameter $v$ represents the degrees of freedom and determines the distribution's shape. As $v$ increases, the *t*-distribution approaches the normal distribution. To eliminate the ambiguity, we have replaced "T3-10" with "*t-distribution (with degrees of freedom ranging from 3 to 10)*".

► Line 45-47 of section "1. Introduction" have been replaced with:

"Other candidates include *t-distribution (with degrees of freedom ranging from 3 to 10), Cauchy distribution, and log-Cauchy distribution*, which have been compared with normal and log-normal distributions in reconstructing the source parameters of the Prairie Grass field experiment (Wang et al., 2017)."

**Comment#2:**

Line 60: What does "deterministic assumption" mean? It is quite confusing.

**Response to comment#2:**

We apologize for any confusion caused by the term "deterministic assumption". Deterministic assumptions aim to define the physical feature of source parameters. A typical one is the constant-release assumption, which assumes that the substances are released at a constant rate during the release period (Kovalets et al., 2020, 2018; Efthimiou et al., 2018, 2017; Tomas et al., 2021; Andronopoulos and Kovalets, 2021; Ma et al., 2018). To avoid confusion, we have replaced the terms "Statistical assumption" and "Deterministic assumption" with "*Assumptions on model-observation discrepancies*" and "*Assumptions on the release characteristics*", respectively, in the introduction section in the revised manuscript.

► Line 38-78 of section "1. Introduction" have been replaced with:

[revised manuscript text omitted]

**Comment#3:**

Figure 1: What do the different shapes and colors in the diagram mean?

**Response to comment#3:**

We apologize for not providing detailed explanations for the shapes and colors in Figure 1. We have added descriptions to the figure caption of the Fig. 1 and relabeled the root nodes using yellow squares in Fig. 1**,** providing a more accurate and detailed introduction to the decision tree model.

►     Line 150-151 of section "2.3 Source localization without knowing the exact release rates" have been replaced with:

[Figure]

"**Figure 1.** Flowchart of XGBoost for predicting $\hat{r}_i$ based on decision tree model. *The yellow squares are the root nodes within each tree, representing the input features in this paper. The purple ellipses denote the child nodes where the model evaluates input features and make decisions to split the data. The green rectangles depict the leaf nodes and refer to the prediction results. The vertical rectangles abstract the internal splitting processes of the trees, indicating decision-*

*making not explicitly detailed in the diagram.*"

**Comment#4:**

Equation (7): Please explain all the parameters here.

**Response to comment#4:**

We appreciate your attention to Equation (7). To avoid confusion, we have replaced the symbol "$T$" with "$M$" in Eq. (7) and have provided additional descriptions for all the parameters.

► Line 147-157 of section "2.3 Source localization without knowing the exact release rates" have been replaced with: "where $K$ is the number of trees, $\mathcal{F} = \{f(x) = \omega_{Q(x)}\}(Q: \mathbb{R}^p \to M, \omega \in \mathbb{R}^M)$ is the space of decision trees, and $Q$ represents the structure of each tree, mapping the feature vector to $M$ leaf nodes. Each $f_k$ corresponds to an independent tree structure $Q$ with leaf node weight $\omega = (\omega_1, \omega_2, \cdots, \omega_M)$. Equation (5) is then used to predict $\hat{\mathbf{r}}_i = (\hat{x}_i, \hat{y}_i)$ for the $i$-th sample.

XGBoost trains $G(\mathbf{X})$ in Eq. (5) by continuously fitting the residual error until the following objective function is minimized:

$$Obj^{(t)} = \sum_{i=1}^{n}\left(\mathbf{r}_i - \left(\hat{\mathbf{r}}_i^{(t-1)} + f_t(\mathbf{X}_i)\right)\right)^2 + \sum_{i=1}^{t}\Omega(f_i), \tag{6}$$

where $t$ represents the training of the $t$-th tree and $\Omega(f_i)$ is the regularization term, given by:

$$\Omega(f) = YM + \frac{1}{2}\lambda\sum_{j=1}^{M}\omega_j^2, \tag{7}$$

*where M is the number of leaf nodes, $\omega_j$ is the leaf node weight for the j-th leaf node, Y and $\lambda$ are penalty coefficients.*
The minimization of Eq. (6) provides the parametric model $G(\mathbf{X})$ that maps the feature ensemble $\mathbf{X}$ extracted from $\mu_p$ to the source location $\mathbf{r}$."

**Comment#5:**

Line 160: Why is the amplitude quantity called "wave rate"?

**Response to comment#5:**

We apologize for the unclear definition. We aim to define the "wave_rate" as a statistical measure that quantifies the fluctuations of $\mu_p$ over time. To reduce the impact of extreme values, the "wave rate" is calculated as the difference between the 90th and 10th quantiles of the normalized observation series. To avoid any ambiguity, we have removed the term "amplitude" from the revised manuscript and clarified the definition of "wave_rate" to ensure it accurately reflects the intended concept.

► Line 159-160 of section "2.3 Source localization without knowing the exact release rates" have been replaced with: "Among the time-domain features, the wave rate quantifies *the fluctuations of $\mu_p$ over time without being overly influenced by extreme values*,"

**Comment#6:**

Lines 160-161: The median value is not a central moment.

**Response to comment#6:**

We appreciate your observation regarding the classification of median value. Upon reviewing relevant literature (Witte and Witte, 2017), the temporal mean and median values are indeed recognized measures of central tendency in statistical analysis. We have made revisions in the manuscript to clarify this point.

► Line 160-161 of section "2.3 Source localization without knowing the exact release rates" have been replaced with:

"while the temporal mean and median values *are measures of central tendency of $\mu_p$ (Witte and Witte, 2017).*"

**Comment#7:**

Line 232: If it is 40[th] percentile, the number of samples for Oct.3 and Oct. 4 should be 1200 and 600.

**Response to comment#7:**

We appreciate your careful review and for identifying this discrepancy. Indeed, we intended to reference the 60[th] percentile, not the 40[th]. To clarify this point, we have revised our descriptions and instead specified that source locations corresponding to the highest 40% of correlation coefficients are selected for further analysis.

► Line 230-232 of section "2.6.2 Simulation settings of atmospheric dispersion model" have been replaced with:

"As described in Sect. 2.5.1, we calculated the correlation coefficient for each sample and *preserved 40% samples with the highest 40% of correlation coefficients* (i.e. 800 samples for Oct. 3 and 400 samples for Oct. 4)."

**Comment#8:**

Line 237: The authors probably mean 80[th], 60[th], 50[th], 40[th], 20[th], and 0[th].

**Response to comment#8:**

We appreciate your attention to this detail. You are correct. In line with the response to comment#7, we have revised the relevant descriptions to accurately reflect the search range.

► Line 236-238 of section "2.7 Sensitivity study" have been replaced with:

"The search range is controlled by the pre-screening threshold*, which is the top proportion of the correlation coefficients in the pre-screening step. Specifically, we use source locations corresponding to the highest 20%, 40%, 50%, 60%, 80%, and 100% of correlation coefficients to define search ranges, with a lower proportion indicating a narrower and more focused search area.*"

**Comment#9:**

Line 238: "A lower percentile" should be "a higher percentile".

**Response to comment#9:**

We appreciate your attention in identifying this discrepancy. We have corrected this error in the revised manuscript, which is consistent with the revisions discussed in comment#7 and comment#8.

**Comment#10:**

Figure 8: No shade appears for the Bayesian inversion results in the lower left panel.

**Response to comment#10:**

Thank you for pointing out the visualization issue regarding Fig. 8. To resolve this problem, we have enlarged the shading in this area for better visualization. Additionally, we have adjusted the shading range from [minimum, maximum] to [lower quartile, upper quartile] to better represent the results.

► Line 385-387 of section "3.3.3 Uncertainty range" have been replaced with:

[Figure]

"**Figure 8.** Release rate estimates over 50 calculations *in SCK-CEN [41]Ar experiment*. *(a) Oct. 3-Bayesian method; (b) Oct. 3-The proposed methods; (c) Oct. 4-Bayesian method; (d) Oct. 4-The proposed methods. The shadow represents the uncertainty range between the lower quartile and the upper quartile. The shadow of each figure is amplified by an enlarged subgraph. The legends in each figure provide the mean estimates of total release.*"

**Comment#11:**

Line 416: What do the various pre-screening ranges refer to?

**Response to comment#11:**

We apologize for any confusion caused by the term "pre-screening ranges". These pre-screening ranges, as detailed in Sect. 2.5.1, refer to the specific subsets of source locations selected based on their correlation coefficients (i.e. search range). The pre-screening process is designed to reduce computational costs and eliminate low-quality samples by focusing on the most promising source locations for further analysis. To eliminate the ambiguity, we have replaced the "pre-screening ranges" with "*search ranges*".

► Line 403-404 of section "3.4.1 Sensitivity to the search range" have been replaced with:

"The error is smaller with a lower threshold, implying that a small *search range* helps reduce the mean and median errors."

► Line 415-417 of section "3.4.1 Sensitivity to the search range" have been replaced with:

"The mean/median error is less than 8% for Oct. 3 and less than 11% for Oct. 4, both of which are smaller than for the various *search ranges* in Fig. 9. This indicates that the proposed method is more robust to this parameter than to the *search range*."

**Comment#12:**

Figure S1: Should it be 20% instead of 10% for the five-fold cross-validation?

**Response to comment#12:**

We appreciate your attention to detail in Figure S1. You are correct about the discrepancy in the percentage for the five-fold cross-validation; it should be 20% instead of 10%. We have made this correction in the revised figure.

► Line 30-31 of Supplementary Material have been replaced with:

[Figure]

**"Figure S1.** Flowchart of the proposed *spatiotemporally-separated* source reconstruction method."

**Comment#13:**

Table S1: Brief descriptions of the hyperparameters should be provided.

**Response to comment#13:**

We appreciate your suggestion regarding Table S1. Brief descriptions of the hyperparameters have been included in the caption of Table S1 for clarity:

► Line 51 of Supplementary Material have been replaced with:

[revised manuscript text omitted]

---

## Author Comment (AC2)

Dear Referee,

We thank the referee for taking the time to provide such a constructive and thorough review of our manuscript (GMD-2023-173). According to the suggestive comments, we have made some modifications to the manuscript, and the responses are listed below. To guide the review process, comments from the referee and original texts in the manuscript are presented in black, our responses are in blue, and any text modifications made to the manuscript are highlighted in red italics. Links are provided below for easy navigation in the document.

**General comments**
**Specific comments**
**References**

We are looking forward to your reply.

Best regards,
Yours sincerely
Sheng Fang

**General comments**

This study proposes a novel approach to the source reconstruction of atmospheric radionuclide emissions in non-stationary emission scenarios. By moving away from the unrealistic assumption of constant emissions and developing a method for spatiotemporally decoupled source reconstruction, it effectively leverages the fact that variations in emission rates significantly impact observations. The methodology involves training machine learning models with the XGBoost algorithm and determining detailed temporal variations in emission rates using the PAMILT algorithm. The paper makes a significant contribution to the field of atmospheric radionuclide emission source reconstruction. The proposed methodology offers an effective means for accurately localizing sources and estimating emission rates in non-stationary scenarios, presenting a promising framework for future research and applications.

[1] The utilization of a temporal sliding-window average filter is commendable. However, elucidating the criteria for feature selection and the impact of varying combinations of observation sites on source estimation would enhance the paper.

[2] Validating the proposed method against the SCK-CEN $^{41}$Ar field experiment data underscores its efficacy and applicability. Nonetheless, conducting further validation studies under diverse scenarios and conditions would enrich our understanding of the method's applicability and limitations. It is recommended to include additional case studies involving different types of releases and weather conditions to assess the method's efficiency and adaptability more comprehensively.

**Response to general comments:**

Thank you for your valuable feedback and suggestive comments on our manuscript, which not only recognizes the innovation and contribution of our approach but also highlights areas for further enhancement of our work. Below are our responses to your main points:

***(1) The criteria for feature selection and the impact of varying combinations of observation sites***
***(1.1) The criteria for feature selection***

The mean cross-validation score (MCV) is used as the criterion for feature selection, and the optimal feature subset is selected as the one that achieves the highest MCV. This selection is implemented by recursively removing the feature with the least importance, and assessing the MCV based on cross validation (Akhtar et al., 2019). It starts with training a XGBoost model with all features, and assessing the importance of each feature based on its contribution to the accuracy of the XGBoost model. Then, the feature with the least importance is removed and the XGBoost model is retrained using the remaining features. The feature importance and MCV are updated accordingly for the removal of another feature. This iteration continues until the optimal number of features is identified, corresponding to the highest MCV achieved during the process.

Using this criterion, unimportant features can be removed to improve the XGBoost model's prediction accuracy, while simultaneously reducing the risk of overfitting and computational costs.

To clearly reflect the criteria of feature selection, we have added some descriptions in relevant section.

► Lines 202-204 of section "2.5.3 Automatic optimization of XGBoost model" have been replaced with:

*"The initial input features (Table 1) are optimized through a feature selection step, which finds the optimal feature subset that achieves the best MCV. The selection is implemented by recursively removing the feature with the least importance,*

*and reassessing the MCV based on cross validation (Akhtar et al., 2019). It starts with training a XGBoost model with all features, and assessing the importance of each feature based on its contribution to the model accuracy. The feature with the least importance is removed and the XGBoost model is retrained using the remaining features. The feature importance and MCV are updated accordingly for the removal of another feature. This iteration continues until the optimal number of features is identified, corresponding to the highest MCV achieved during the process.* The overall flowchart of the proposed spatiotemporally decoupled source reconstruction model is shown in Fig. S1."

*(1.2) Impact of varying combinations of observation sites*

We agree that it is important to discuss the impact of varying combinations of observation sites. Briefly speaking, the selection of representative sites is more important for model performance than increasing the number of sites. In this study, we have demonstrated this impact through sensitivity studies with respect to both the number and combination of observation sites.

1) The number of observation sites

Additional observation sites can better capture environmental variability and impose stronger constraints on the estimation, leading to more robust results. However, the usage of all observation sites may cause overfitting of the XGBoost model and reduce the prediction accuracy of the trained model. Our sensitivity study (Fig. 12) also reveals that locating the source using all observation sites does not achieve the lowest error level, though the error level remains low.

2) The position of observation sites

Observation sites located at appropriate position can capture environmental variability and provide adequate information for locating the source. Utilizing only these representative sites can alleviate overfitting and enhance the prediction accuracy of the XGBoost model. The sensitivity study demonstrates that the lowest error levels are achieved by a subset of sites, i.e. Site ABD on Oct. 3 and Site BD on Oct. 4. For Oct.3, multi-site estimations with Site B always produce lower error levels, and single-site estimation using Site B also achieves high accuracy. For Oct.4, multi-site estimations with Site BD always achieve relatively low error levels. These results prove the importance of representative sites in source location estimation. In addition, the representative sites (Site B for Oct. 3 and Site BD for Oct. 4) are consistent with the feature selection results in Fig. 4, preliminarily indicating the potential of feature selection to identify representative sites.

To highlight the impact of varying combinations of observation sites, we have added some descriptions in relevant section.

► Lines 438-449 of section "3.4.4 Sensitivity to the number and combination of observation sites" have been replaced with:

"Figure 12 compares the results obtained with different numbers and combinations of observation sites. The results indicate that the localization error may be more sensitive to the position of the observation site than to the number of sites included. *The error level of all-site estimations is relatively low for both days, indicating that increasing the number of observation sites can better constrain the solution and help improve the robustness of the model. However, the lowest error levels are achieved by a subset of sites, i.e. Site ABD on Oct. 3 and Site BD on Oct. 4. This is possibly because the usage of all observation sites may cause overfitting and reduce the prediction accuracy of the XGBoost model. On the other hand, using only representative sites at appropriate position is effective to alleviate overfitting and enhance the model accuracy, because these representative sites can capture environmental variability and provide adequate information for locating the source. For Oct.3, multi-site estimations with Site B always produce lower error levels, and*

*single-site estimation using Site B also achieves high accuracy. For Oct.4, multi-site estimations with Site BD always achieve relatively low error levels. These results prove the importance of representative sites in source location estimation. The representative sites (Site B for Oct. 3 and Site BD for Oct. 4) are consistent with the importance calculated in the feature selection (Fig. 4), preliminarily indicating the potential of feature selection to identify representative sites. In addition, feature selection also reduces the mean error level in most cases.*"

***(2) More validation of the method:***

We acknowledge the importance of validating our method against diverse scenarios and weather conditions to assess its robustness and practical applicability.

To address your concern, we have incorporated an additional validation case based on the first release of the European Tracer Experiment (ETEX-1) (Nodop et al., 1998), involving a different type of releases (continental-scale) and more complex meteorological conditions (temporally and spatially varying), to thoroughly assess the method's efficiency and adaptability. During ETEX-1, a total of 340 kg of perfluoromethylcyclohexane (PMCH) was released continuously into the atmosphere from 23 Oct 1994 16:00:00 UTC and 24 Oct 1994 03:50:00 UTC. Assuming the release could have occurred between 23 Oct 1994 00:00:00 UTC and 28 Oct 1994 00:00:00 UTC, it is viewed as a temporally-varying release, with a release rate of zero outside the actual release window. A total of 3104 available observations (3h-averaged concentrations) were collected by 168 sites. As shown in Fig. 2b, we choose two groups of observation sites: the first comprises four sites (i.e. B05, D10, D16, F02) randomly selected from the sites within the sample zone (Group1, with a total of 92 available observations), and the second involves four sites (i.e. CR02, D15, DK08, S09) randomly selected from the sites beyond the sample zone boundaries (Group2, with a total of 90 available observations).

For the continental-scale ETEX-1 experiment, the proposed method still achieves the lowest source location errors among all methods, which are below 10 km and 20 km (less than the grid size of 0.25°×0.25°) for Group1 and Group2, respectively. Regarding the results of the uncertainty analysis, the average relative source location error was 2.42% and 4.97% for Group1 and Group2, respectively, lower than the correlation-based and Bayesian methods. The proposed method provides highly accurate mean estimates of release rate for both groups, although with a large uncertainty range. These results demonstrate that spatiotemporally decoupled source reconstruction is feasible and achieves satisfactory accuracy in multi-scale release scenario, thereby providing a promising framework for reconstructing atmospheric radionuclide releases.

An overview of the ETEX-1 experiment and corresponding source reconstruction results are provided below. Detailed results and discussions have been included in the revised manuscript.

**2.6 Multi-scale Validations**
**2.6.1 Field experiments**

The ETEX-1 experiment took place at Monterfil in Brittany, France, on 23 October 1994 (Nodop et al., 1998). During ETEX-1, a total release of 340 kg of PMCH was released into the atmosphere during 23 Oct 1994 16:00:00 UTC and 24 Oct 1994 03:50:00 UTC. As illustrated in Fig. 2b, the source coordinates were (2.0083°W, 48.058°N). A total of 3104 available observations (3h-averaged concentrations) were collected by 168 sites. ETEX-1 has been commonly used as a validation scenario for reconstructing atmospheric radionuclide releases (Ulimoen and Klein, 2023; Tomas et al., 2021). The candidate source locations are uniformly sampled from the green shaded zone. We choose two groups of observation sites: the first comprises four sites (i.e. B05, D10, D16, F02) randomly selected from the sites within the sample zone (Group1, with a total of 92 available observations), and the second involves four sites (i.e. CR02, D15, DK08, S09) randomly selected from the sites beyond the sample zone boundaries (Group2, with a total of 90 available observations).

Compared to the SCK-CEN $^{41}$Ar experiment, the observations of the ETEX-1 exhibit temporal sparsity, lower temporal resolution, and increased complexity in meteorological conditions.

[Figure]

**Figure 2.** Release location and observation sites of two filed experiments. (a) SCK-CEN $^{41}$Ar experiment. The map was created based on the relative positions of the release source and observation sites, as detailed in (Drews et al., 2002). The

coordinate of the sample border is (500m, -200m) and (1180m, 580m) on Oct. 3, and (450m, 10m) and (850m, 450m) on Oct. 4. It was plotted using MATLAB 2016b, instead of created by a map provider; (b) ETEX-1 experiment. The map was created based on the real longitudes and latitudes of the release source and observation sites, as detailed in (Nodop et al., 1998). The coordinate of the sample border is (10°W, 40°N) and (10°E, 60°N). It was plotted using the function *cartopy* of Python, instead of created by a map provider.

**2.6.2 Simulation settings of atmospheric dispersion model**

For the ETEX-1 experiment, the FLEXible PARTicle (FLEXPART) model (version 10.4) was applied to simulate the dispersion of PMCH (Pisso et al., 2019). The meteorological data were from the United States National Centers of Environmental Prediction Climate Forecast System Reanalysis, which has a spatial resolution of 0.5°×0.5° and time resolution of 6h. To rapidly establish the relationship between the varying source locations and the observations, 182 backward simulations were performed using FLEXPART. The candidate source locations were uniformly sampled from the shaded zone in Fig. 2b, resulting a total of 6561 source locations. As described in Sect. 2.5.1, 2624 candidate source locations are preserved by pre-screening step. The constant factors mentioned in Sect. 2.5.2 are $5.60×10^{12}$ and $2.86×10^{13}$.

**3.2 Optimization of XGBoost model**
**3.2.2 Feature selection**

Figure 4 compares the importance of the selected features at each site. For the ETEX-1 experiment, Fig. 4c and d showed that the features of Group1 and Group2 are almost reserved after the feature selection process (only one feature is removed for each case), indicating fewer redundant features than that in the SCK-CEN [41]Ar experiment. The time-domain features are also dominant, but the frequency-domain features at some sites also play important roles such as D16 and S09. The MCVs of the ETEX-1 experiment have similar variation trend as that of the SCK-CEN [41]Ar experiment (Fig. S4c and d). However, the Group1 has obviously lower MCV than other experiments.

**(a)**

[Figure]

**(b)**

[Figure]

[Figure]

**Figure 4.** Feature importance. (a) Oct. 3; (b) Oct. 4; (c) Group1; (d) Group2.

[Figure]

**Figure S4.** Results of feature selection in x (longitude) and y (latitude) directions. (a) Oct. 3; (b) Oct. 4; (c) Group1; (d) Group2. The black stars denote the optimal number of features. The table inserted in each subgraph lists the selected features for each observation site.

**3.3 Source reconstruction**
**3.3.1 Source location estimation**

Figure 5 compares the best-estimated source locations of the correlation-based method, the Bayesian method, and the proposed method with the ground truth. For the ETEX-1 experiment, the pre-screening zone also covers the true source location for Group1 and Group2. But source locations estimated by the correlation-based method are 411.85 km and 486.41 km away from the ground truth for Group1 and Group2, respectively. The location error of the Bayesian method estimates is only 30.50 km for Group1, but it increases to 520.77 km for Group2, indicating its sensitivity to the observations. In contrary, the proposed method achieves the lowest source location errors, which are below 10 km and 20 km for Group1 and Group2, respectively. Because the feature selection hardly removed the features (Fig. 4c and d), the estimated source locations with and without feature selection basically overlap for both groups.

[Figure]

**Figure 5.** Source location estimation results. The yellow dots denote the maximum correlation points, which are the localization results of the correlation-based method. The green and red stars represent the localization results based on XGBoost before and after feature selection, respectively. The cyan diamonds represent the localization results based on the Bayesian method. (a) Oct. 3; (b) Oct. 4; (c) Group1; (d) Group2. A detailed enlargement of the region around (2.5°W, 47.5°N) to (1.5°W, 48.5°N) is shown in the bottom right corner in (c) and (d) to highlight the source location estimation results of the proposed methods.

**3.3.2 Release rates**

Figure 6 displays the release rates estimated by the Bayesian and PAMILT methods based on the source location estimates in Fig. 5. For the two ETEX-1 groups (Fig. 6c and d), Bayesian estimates exhibit noticeable fluctuations, leading to underestimations–58.11% for Group1 and 51.44% for Group2. Furthermore, the temporal profile of the Bayesian estimates for Group2 completely falls out of the true release window. In contrast, most releases of the PAMILT estimates are within the true release time window, especially for Group2, despite overestimations reach 52.38% for Group1 and 57.65% for Group2, after feature selection process. Compared to the SCK-CEN [41]Ar experiment, the increased deviation in the ETEX-1 experiment may be attributed to the limited observations at the four sites as shown in Fig. S3.

[Figure]

**Figure 6.** Release rate estimation results with different location estimates. (a) Oct. 3; (b) Oct. 4; (c) Group1; (d) Group2. The release rates labelled by XGBoost or XGBoost+feature selection are estimated using PAMILT method.

[Figure]

**Figure S3.** Observations before and after filtering at observation sites. (a) Oct. 3; (b) Oct. 4; (c) Group1; (d) Group2.

**3.3.3 Uncertainty range**

Figure 7 compares the spatial distribution of 50 estimates produced by the different source location estimation methods. For the ETEX-1 experiment, the estimates of the correlation-based method are quite spread, whereas those of the Bayesian method are more concentrated. The Bayesian estimates are close to the truth in Group1, but are noticeably deviated in Group2. The phenomenon indicates that the Bayesian method is sensitive to the observations, especially when observations are limited. Fig. S5c and d also reveals that the Bayesian-estimated posterior distribution is multimodal for both ETEX-1 groups, which can be avoided by using additional observations (Fig. S5e). In contrast, the

proposed method provides estimates quite concentrated around the truth for both Group1 and Group2, indicating its efficiency in the case of limited observations. Compared to Group2, Group1 exhibits lower source location error. This is attributed to the fact that, as shown in Fig. 2, Group1's four observation sites are situated closer to the sampled source locations than those of Group2, thereby providing more accurate plume characteristics and resulting in lower dispersion error.

[Figure]

**Figure 7.** Spatial distribution of 50 source location estimates. (a) Oct. 3; (b) Oct. 4; (c) Group1; (d) Group2. Each circle denotes an individual estimate as detailed in Sect. 2.8.5, with color variations indicating the respective method employed. Histograms along the axes represent the frequency distribution of the estimates along the respective axis.

[Figure]

**Figure S5.** Posterior distributions of source location parameters. (a) Oct. 3; (b) Oct. 4; (c) Group1; (d) Group2; (e) ETEX-1 (all observations in ETEX-1 are used). The black solid lines denote the true location parameters and the dashed lines denote the mean estimates of all posterior samples.

Figure 9 compares the uncertainty ranges of the release rate estimates in the two ETEX-1 groups. For both groups, the Bayesian estimates show underestimations (including the mean estimate), but it presents small uncertainty range (Fig. 9a and c). In addition, the Bayesian estimates all fall out of the time window of the true release in Group2 (Fig.9c). The mean PAMILT estimates are more accurate mean Bayesian estimates, which most releases are within the time window of the true release (Fig. 9b and d). However, PAMILT estimates for the ETEX-1 experiment show a large uncertainty range in both groups, in contrast to the SCK-CEN [41]Ar experiment. The reason for this discrepancy is that the source-receptor matrices of the ETEX-1 experiment are more sensitive to errors in source location than those of the SCK-CEN [41]Ar experiment. The greater sensitivity originates from the ETEX-1 experiment's more complex meteorological conditions, both spatially and temporally. As for the mean total release, the Bayesian method shows underestimations of 70.93% for Group1 and 74.15% for Group2. In comparison, the proposed method shows deviation of only 0.71% for Group1 and 0.09% for Group2, after feature selection.

[Figure]

**Figure 9.** Release rate estimates over 50 calculations in ETEX-1 experiment. (a) Group1-Bayesian method; (b) Group1-PAMILT method; (c) Group2-Bayesian method; (d) Group2-PAMILT method.

Table 3 lists the mean and standard deviation of the relative errors for the 50 estimates given by different methods. For the ETEX-1 experiment, the Bayesian method shows case-sensitive performances with respect to the mean relative error of source location estimation, whereas the proposed method gives the most accurate source locations with small uncertainties for both groups. As for the total release, the proposed methods show smaller mean relative errors than the Bayesian methods, but the Bayesian method shows a smaller standard deviation.

**Table 3.** Relative errors of source reconstruction. $\delta_{\mathbf{r}}$ represents the relative error of source location, which is positive and $\delta_Q$ denotes the relative error of total release, where a positive value indicates overestimation and a negative value denotes underestimation.

| Experiments | Statistical parameters (Relative error) | | Correlation-based method | Bayesian method | The proposed method | |
| --- | --- | --- | --- | --- | --- | --- |
| | | | | | XGBoost | XGBoost+ feature selection |
| Oct. 3 | $\delta_{\mathbf{r}}$ | Mean | 14.10% | 11.88% | 5.18% | 4.68% |
| | | Std | 11.37% | 7.53% | 1.79% | 2.05% |
| | $\delta_Q$ | Mean | - | 153.61% | -16.93% | -18.30% |
| | | Std | - | 189.76% | 9.45% | 8.01% |
| Oct. 4 | $\delta_{\mathbf{r}}$ | Mean | 14.30% | 12.83% | 6.83% | 4.71% |
| | | Std | 9.60% | 1.68% | 1.76% | 1.53% |
| | $\delta_Q$ | Mean | - | 42.29% | -54.12% | -47.42% |
| | | Std | - | 15.05% | 6.47% | 5.85% |
| Group1 | $\delta_{\mathbf{r}}$ | Mean | 16.95% | 3.22% | 2.32% | 2.42% |
| | | Std | 7.46% | 2.75% | 1.43% | 1.43% |
| | $\delta_Q$ | Mean | - | -70.93% | 18.12% | -0.71% |
| | | Std | - | 17.87% | 99.85% | 102.01% |
| Group2 | $\delta_{\mathbf{r}}$ | Mean | 21.9% | 23.97% | 5.21% | 4.97% |
| | | Std | 5.05% | 1.97% | 2.42% | 2.35% |
| | $\delta_Q$ | Mean | - | -74.15% | 16.67% | 0.09% |
| | | Std | - | 11.68% | 93.50% | 109.56% |

**Specific comments**

**Comment#1:**

L215 - Figure 2: The axes represent distances and should therefore have identical scales for clarity and accuracy.

**Response to comment#1:**

We appreciate your attention on the different scales of axes on Figure 2. We have revised the figure to ensure that both axes represent distances with identical scales.

► Figure 2 has been replaced with:

[Figure]

"**Figure 2**. Release location and observation sites of SCK-CEN [41]Ar experiment. The map was created based on the relative positions of the release source and observation sites, as detailed in (Drews et al., 2002). It was plotted using MATLAB 2016b, instead of created by a map provider."

**Comment#2:**

L222 - Consideration of vertical information could provide a more comprehensive understanding of the dispersion patterns. How does the model account for vertical dispersion?

**Response to comment#2:**

Thank you for your constructive comment. We agree that incorporating vertical information can aid in understanding the dispersion patterns. In the SCK-CEN [41]Ar experiment, the [41]Ar was emitted from a 60-m stack, while the ground-level fluence rates were collected by NaI (Tl) gamma detectors. Due to the lack of vertical observations in

this experiment, the vertical dispersion has not been discussed in the manuscript.

The RIMPUFF model is a gaussian puff model that uses the diffusion coefficient in vertical direction to describe the vertical dispersion of each puff. In this study, the Karlsruhe-Jülich diffusion coefficients were used to calculate the vertical dispersion, which has been validated for the SCK-CEN [41]Ar experiment and has shown good accuracy (Li et al., 2019).

In the future, we will try to incorporate the vertical dispersion information into the source parameters.

To ensure clarity, we have added some descriptions of vertical information in the revised manuscript.

► Lines 218 of section "2.6.1 SCK-CEN [41]Ar field experiment" have been replaced with:

"The 60-s-average *ground-level* fluence rates were continuously collected by an array of NaI (Tl) gamma detectors,"

► Lines 225-226 of section "2.6.2 Simulation settings of atmospheric dispersion model" have been replaced with:

"The calculation domain measured 1800 m×1800 and the grid resolution was 10 m×10 m. *The release height of [41]Ar was presumed to be 60 m.*"

**Comment#3:**

L302 - Figure 3: To ensure clarity and accuracy in data representation, the scales on the vertical and horizontal axes must be consistent.

**Response to comment#3:**

We appreciate your attention on the different scales on axes on Figure 3. We have carefully adjusted Figure 3 so that the scales on both the vertical and horizontal axes are now consistent.

► Figure 3 has been replaced with:

[Figure]

"**Figure 3**. Scatter plots of the original (yellow squares) and decoupled (green squares) observations versus the constant-release simulation results. (a) Oct. 3-Synthetic observations; (b) Oct. 4-Synthetic observations; (c) Oct. 3-Real observations; (d) Oct. 4-Real observations."

**Comment#4:**

L340 - The capability to estimate with greater accuracy than the grid size warrants a discussion. What implications does this have for the model's precision and its practical significance?

**Response to comment#4:**

Thank you for your insightful query. The ability to estimate source locations with accuracy surpassing the grid size can be attributed to the strong fitting capability of the optimized XGBoost model (Chen and Guestrin, 2016; Grinsztajn et al., 2022), which excels in interpolating within the grid size and extrapolating beyond the source location samples. As discussed in our response to the General comments, ETEX-1 experiment also achieved source location accuracy beyond the grid size (Fig. 5c and d, the grid size is 0.25°×0.25°), suggesting that the phenomenon is not merely coincidental. In addition, previous studies also achieved similar source location accuracy using traditional methods (Lucas et al., 2017; Tichý et al., 2017). However, this ability, although inherent, does not uniformly manifest across all optimized XGBoost models, as external factors like observation noises and meteorological data inaccuracies can also impact the accuracy of source location estimation. The uncertainty analysis in Sect. 3.3.3 has demonstrated that the source location estimates tend to cluster within several grids surrounding the true source, which is more reasonable and practical in real-world scenarios. Detailed discussions are as follows:

(1) *Enhanced accuracy through XGBoost*: The high accuracy in locating the source is directly achieved by the XGBoost model, since it establishes the complex nonlinear relationships between the input features and the source location. Utilizing automatic optimization techniques (detailed in Sect. 2.5.3), 8 main hyperparameters of XGBoost and 24 observation series features are finely tuned to achieve an optimized model. This optimization not only mitigates the risk of overfitting but also enhances the model's ability for interpolation within the grid size and extrapolation beyond the source location samples.

(2) *Validation on ETEX-1 experiment*: The proposed method has been validated through ETEX-1 experiment, as discussed in our response to the General comments. Compared to SCK-CEN [41]Ar experiment, ETEX-1 involves a different type of releases (continental-scale) and more complex meteorological conditions (temporally and spatially varying). As shown in Fig. 5c and d, this experiment also achieved source location accuracy beyond the grid size (0.25°×0.25°), suggesting that the phenomenon is not merely coincidental. Providing that the XGBoost model is effectively optimized and the observations are reliable, the model has ability to achieve high accuracy.

[Figure]

**Figure 5.** Source location estimation results. The yellow dots denote the maximum correlation points, which are the localization results of the correlation-based method. The green and red stars represent the localization results based on XGBoost before and after feature selection, respectively. The cyan diamonds represent the localization results based on the Bayesian method. (a) Oct. 3; (b) Oct. 4; (c) Group1; (d) Group2. A detailed enlargement of the region around (2.5°W, 47.5°N) to (1.5°W, 48.5°N) is shown in the bottom right corner in (c) and (d) to highlight the source location estimation results of the proposed methods.

(3) **Uncertainty analysis of XGBoost hyperparameters**: An uncertainty analysis of the XGBoost hyperparameters (Fig. 7) has revealed that not all source location estimates achieve greater accuracy than the grid size. Instead, source location estimates tend to cluster within several grids surrounding the true source. This phenomenon highlights the practical significance of the proposed method.

[Figure]

**Figure 7.** Spatial distribution of 50 source location estimates. (a) Oct. 3; (b) Oct. 4; (c) Group1; (d) Group2. Each circle denotes an individual estimate as detailed in Sect. 2.8.5, with color variations indicating the respective method employed. Histograms along the axes represent the frequency distribution of the estimates along the respective axis.

To avoid confusion, we have added some discussions in the revised manuscript to explain the greater accuracy than the grid size. The revised manuscript has also included the reconstruction results of ETEX-1 experiment (see our response to the General comments), which will further prove the model's ability.

► Lines 339-344 of section "3.3.1 Localization" have been replaced with:

"The estimates without feature selection are only 10.65 m (Oct. 3) and 20.62 m (Oct. 4) away from the true locations. Feature selection further reduces these errors to 6.19 m (Oct. 3) and 4.52 m (Oct. 4), which are below the grid size (10 m × 10 m) of the ATDM simulation. *The ability to estimate source locations with accuracy surpassing the grid size can be attributed to the strong fitting capability of the optimized XGBoost model* (Chen and Guestrin, 2016; Grinsztajn et al.,

*2022**), which excels in interpolating within the grid size and extrapolating beyond the source location samples. However, this capability, although inherent, does not uniformly manifest across all optimized XGBoost models, as external factors like observation noises and meteorological data inaccuracies can also impact the accuracy of source location estimation. Therefore, uncertainty analysis will be performed in Sect. 3.3.3 to demonstrate the model's uncertainty range in estimating the source location.* The proposed method gives a relative error of less than 0.9% for both days, whereas the Bayesian method produces a relative error of above 11% and that of the correlation-based method can be as high as 26%. The best estimates for Oct. 3 are more accurate than those for Oct. 4, possibly because of the better layout of observation sites (Fig. 2) and the better decoupling results (Fig. 3)."

**Comment#5:**

L346 - Figure 5: As these axes represent distances, maintaining identical scales on both axes is crucial for accurate data interpretation.

**Response to comment#5:**

Thank you for pointing out this issue. We have revised Figure 5 to ensure that both the horizontal and vertical axes have the same scale.

► Figure 5 has been replaced with:

[Figure]

"**Figure 5.** Source localization results. The yellow dots denote the maximum correlation points, which are the localization results of the correlation-based method. The green and red stars represent the localization results based on XGBoost before and after feature selection, respectively. The cyan diamonds represent the localization results based on the Bayesian method. (a) Oct. 3; (b) Oct. 4."

**Comment#6:**

L374 - Figure 7: Given that both axes represent distances, their scales should be identical. The complexity of the graphs

necessitates a detailed explanation within the figure caption to aid in interpretation.

**Response to comment#6:**

Thank you for your constructive comment. We have revised Figure 7 to ensure that both axes are now on identical scales and expanded the figure caption to include a detailed explanation of the graph's components.

► Figure 7 and its figure caption have been replaced with:

[Figure]

"**Figure 7.** *Spatial distribution of 50 source location estimates. (a) Oct. 3; (b) Oct. 4. Each circle denotes an individual estimate as detailed in Sect. 2.8.5, with color variations indicating the respective method employed. Histograms along the axes represent the frequency distribution of the estimates along the respective axis.*"

**Thanks again for such a thorough review!**

**References**

Akhtar, F., Li, J., Pei, Y., Xu, Y., Rajput, A., and Wang, Q.: Optimal Features Subset Selection for Large for Gestational Age Classification Using GridSearch Based Recursive Feature Elimination with Cross-Validation Scheme, in: International Conference on Frontier Computing, 63–71, https://doi.org/10.1007/978-981-15-3250-4_8, 2019.

Chen, T. and Guestrin, C.: XGBoost: A scalable tree boosting system, Proc. ACM SIGKDD Int. Conf. Knowl. Discov. Data Min., 13-17-Augu, 785–794, https://doi.org/10.1145/2939672.2939785, 2016.

Drews, M., Aage, H. K., Bargholz, K., Ejsing Jørgensen, H., Korsbech, U., Lauritzen, B., Mikkelsen, T., Rojas-Palma, C., and Ammel, R. Van: Measurements of plume geometry and argon-41 radiation field at the BR1 reactor in Mol, Belgium, 1–43 pp., 2002.

Grinsztajn, L., Oyallon, E., and Varoquaux, G.: Why do tree-based models still outperform deep learning on tabular data?, 2022.

Li, X., Xiong, W., Hu, X., Sun, S., Li, H., Yang, X., Zhang, Q., Nibart, M., Albergel, A., and Fang, S.: An accurate and ultrafast method for estimating three-dimensional radiological dose rate fields from arbitrary atmospheric radionuclide distributions, Atmos. Environ., 199, 143–154, https://doi.org/10.1016/j.atmosenv.2018.11.001, 2019.

Lucas, D. D., Simpson, M., Cameron-Smith, P., and Baskett, R. L.: Bayesian inverse modeling of the atmospheric transport and emissions of a controlled tracer release from a nuclear power plant, Atmos. Chem. Phys., 17, 13521–13543, https://doi.org/10.5194/acp-17-13521-2017, 2017.

Nodop, K., Connolly, R., and Girardi, F.: The field campaigns of the European tracer experiment (ETEX): Overview and results, Atmos. Environ., 32, 4095–4108, https://doi.org/10.1016/S1352-2310(98)00190-3, 1998.

Pisso, I., Sollum, E., Grythe, H., Kristiansen, N. I., Cassiani, M., Eckhardt, S., Arnold, D., Morton, D., Thompson, R. L., Groot Zwaaftink, C. D., Evangeliou, N., Sodemann, H., Haimberger, L., Henne, S., Brunner, D., Burkhart, J. F., Fouilloux, A., Brioude, J., Philipp, A., Seibert, P., and Stohl, A.: The Lagrangian particle dispersion model FLEXPART version 10.4, Geosci. Model Dev., 12, 4955–4997, https://doi.org/10.5194/gmd-12-4955-2019, 2019.

Tichý, O., Ŝmídl, V., Hofman, R., Ŝindelárová, K., Hýza, M., and Stohl, A.: Bayesian inverse modeling and source location of an unintended 131I release in Europe in the fall of 2011, Atmos. Chem. Phys., 17, 12677–12696, https://doi.org/10.5194/acp-17-12677-2017, 2017.

Tomas, J. M., Peereboom, V., Kloosterman, A., and van Dijk, A.: Detection of radioactivity of unknown origin: Protective actions based on inverse modelling, J. Environ. Radioact., 235–236, 106643, https://doi.org/10.1016/j.jenvrad.2021.106643, 2021.

Ulimoen, M. and Klein, H.: Localisation of atmospheric release of radioisotopes using inverse methods and footprints of receptors as sources, J. Hazard. Mater., 451, https://doi.org/10.1016/j.jhazmat.2023.131156, 2023.

---

## Author Response (AR1)

Dear editors and referees,

Thank you for taking the time to provide such constructive and thorough reviews of our manuscript (GMD-2023-173). We greatly appreciate the interests that the editors and the referees have taken in our manuscript.

All the comments and suggestions are very helpful for improving our paper. In the revised manuscript, we have addressed all the comments from the referees. Specifically, we have made the following main changes in this revision:

1. **We have reformulated the texts, figures and tables in a more compact and clear way. To avoid confusion, inaccuracies in statements and terminology have been corrected.**

2. **We have provided additional details on the dispersion model, reconstruction method and hyperparameters. Discussions around the results have been enhanced to highlight the method's strengths and limitations.**

3. **We have added another validation based on the first release of the European Tracer Experiment, to more comprehensively assess the method's efficiency and applicability. Furthermore, we have included an analysis of the reconstruction's sensitivity to the meteorological errors to demonstrate the effect of dispersion model errors.**

The responses are listed below. To guide the review process, comments from the referee and original texts in the manuscript are presented in black, our responses are in blue, and any text modifications made to the manuscript are highlighted in red italics. The line numbers mentioned in this response correspond to those in the revised manuscript (the clear version, not the marked-up version). Links are provided below for easy navigation in the document.

**Referee #1**
    **General comments**
    **Major points**
    **Minor points**
    **References**

**Referee #2**
    **General comments**
    **Specific comments**
    **References**

We are looking forward to your reply.

Best regards,
Yours sincerely
Sheng Fang

**Referee #1**

**General comments**

The paper presented a source reconstructing procedure by first locating the source location before estimating the emission rates. A machine learning method has been used in the first step to locate the source location. The overall results are quite interesting and encouraging.

However, there are several shortcomings in this manuscript. Some of the statements are not accurate and some terminology uses are also questionable. The presentation of the machine learning method is not easy to follow for those who are not quite familiar with the same method and software. In addition, the dispersion model errors affect the results but are not sufficiently considered or discussed. It is also a concern that the method is only tested with a single set of experimental data. More test cases are probably needed.

**Response to general comments:**

Thank you for your valuable feedback and suggestive comments on our manuscript. We are particularly grateful for your remarks about our results being "quite interesting and encouraging". We have addressed the issues you have raised and revised the manuscript accordingly, which are detailed as follows:

*(1) Inaccurate statements and terminology use:*

Following your comments, we have thoroughly reviewed our manuscript and corrected all statements that were inaccurate or unclear. Particularly, we have elaborated and rephrased the "Spatiotemporally-decoupled" and "constant-release assumption" in the introduction section as "*Spatiotemporally separated*" and "*Assumptions on the release characteristics*", respectively. Moreover, the title of the revised manuscript has been changed to "*A spatiotemporally separated framework for reconstructing the source of atmospheric radionuclide releases*" to better describe the current two-step method. More detailed revisions are provided in the response to your specific comments. Regarding the use of terminology, we have provided clear definitions and detailed explanations, such as:

**(1.1) Line 15 of section "Abstract": "source localization"**

The term "source localization" has been replaced with "source location estimation" in the revised manuscript. For example,

► Lines 13-14 of section "Abstract":

"A machine learning model is trained to link these features to the source location, enabling independent *source location estimations*."

► Lines 92-96 of section "1. Introduction":

"The performance of the proposed method is compared with the correlation-based method *in terms of source location estimation* and the Bayesian method *in terms of* spatiotemporal accuracy. The sensitivity of the *source location estimation* to the spatial search range, size of the sliding window, feature type, number and combination of sites *, and meteorological errors* is also investigated *for the SCK-CEN $^{41}$Ar experiment*."

**(1.2) Line 60 of section "1. Introduction": "Deterministic assumption"**

Deterministic assumptions aim to define the physical feature of source parameters. A typical one is the constant-release assumption, which assumes that the substances are released at a constant rate during the release period (Kovalets

et al., 2020, 2018; Efthimiou et al., 2018, 2017; Tomas et al., 2021; Andronopoulos and Kovalets, 2021; Ma et al., 2018). To avoid confusion, we have replaced the terms "Statistical assumption" and "Deterministic assumption" with "*Assumptions on model–observation discrepancies*" and "*Assumptions on the release characteristics*", respectively, in the introduction section in the revised manuscript.

► Lines 41-44 of section "1. Introduction":

"To reduce the problem of ill-posedness, most previous studies have attempted to constrain the reconstruction by imposing assumptions on *the model–observation discrepancies or release characteristics. Assumptions on model–observation discrepancies* are widely used in Bayesian methods to simultaneously reconstruct the posterior distributions of spatiotemporal source parameters (De Meutter et al., 2021; Meutter and Hoffman, 2020; Xue et al., 2017)."

► Lines 64-74 of section "1. Introduction":

"*Assumptions on the release characteristics aim to reduce the dimension of the solution space to 4 or 5, namely the two source location coordinates, the total release, and the release time (or the release start and end time), i.e. an instantaneous release at one time or constant release over a period (Kovalets et al., 2020, 2018; Efthimiou et al., 2018, 2017; Tomas et al., 2021; Andronopoulos and Kovalets, 2021; Ma et al., 2018). Under these assumptions, the correlation-based method exhibits high accuracy for ideal cases under stationary meteorological conditions, such as synthetic simulation experiments (Ma et al., 2018) and wind tunnel experiments (Kovalets et al., 2018; Efthimiou et al., 2017). However, previous studies have also demonstrated that real-world applications may be much more challenging, (Kovalets et al., 2020; Tomas et al., 2021; Andronopoulos and Kovalets, 2021; Becker et al., 2007) because the release usually exhibits temporal variations and may experience non-stationary meteorological fields. The interaction between the time-varying release characteristics and non-stationary meteorological fields is neglected in the instantaneous-release and constant-release assumptions, leading to inaccurate reconstruction.*"

**(1.3) Line 108 of section "2.1 Source reconstruction models": "$A(r) = [A_1(r), A_2(r), \cdots, A_N(r)]^T \in \mathbb{R}^{N \times N}$"**

The matrix $\mathbf{A(r)}$ is not a square matrix in general. We have modified the dimension of the matrix: $\mathbf{A(r)} = [A_1(\mathbf{r}), A_2(\mathbf{r}), \cdots, A_N(\mathbf{r})]^T \in \mathbb{R}^{N \times S}$, where $N$ is the number of sequential time steps and $S$ is the length of release rate vector $\mathbf{q}$.

► Lines 99-108 of section "2.1 Source reconstruction models":

"For an atmospheric radionuclide release, Eq. (1) relates the observations at each observation site to the source parameters:

$$\boldsymbol{\mu} = \mathbf{F(r, q)} + \boldsymbol{\varepsilon} , \tag{1}$$

where $\boldsymbol{\mu} = [\mu_1, \mu_2, \cdots, \mu_N]^T \in \mathbb{R}^N$ is an observation vector composed of observations at $N$ sequential time steps, the function $\mathbf{F}$ maps the source parameters to the observations, i.e. an atmospheric dispersion model, $\mathbf{r}$ refers to the source location, $q \in \mathbb{R}^S$ is the temporally varying release rate, and $\boldsymbol{\varepsilon} \in \mathbb{R}^N$ is a vector containing both model and measurement errors.

In most source reconstruction models, $\mathbf{F}$ is simplified to the product of $\mathbf{q}$ and a source–receptor matrix $\mathbf{A}$ that depends on the source location:

$$\boldsymbol{\mu} = \mathbf{A(r)q} + \boldsymbol{\varepsilon} , \tag{2}$$

where $A(r) = [A_1(r), A_2(r), \cdots, A_N(r)]^T \in \mathbb{R}^{N \times S}$ and each row describes the sensitivity of an observation to the release rate $\mathbf{q}$ given the source location $\mathbf{r}$."

**(1.4) Line 131 of section "2.2 Spatiotemporal decoupling": "sliding window"**

As outlined in Eq. (4), a one-sided window is employed. This one-sided temporal sliding-window average filter involves the current and previous observations in the window, acknowledging that future observations are not available for filtering in practice. Compared to the centered window, the one-sided window excels in real-time data processing and rapid response to changes in the observations, making it more suitable for real applications.

►     Lines 128-133 of section "2.2 Observation filtering for spatiotemporally separated reconstruction":

"In this study, the following operator matrix is constructed to impose a *one-sided* temporal sliding-window average filter (Eamonn Keogh, Selina Chu, 2004):

$$\mathbf{P} = \frac{1}{T} \begin{bmatrix} 1 & & & & & & & \\ 1 & 1 & & & & & & \\ & \vdots & & & & & & \\ 1 & 1 & \cdots & 1 & & & & \\ 1 & 1 & \cdots & 1 & 1 & & & \\ & 1 & 1 & \cdots & 1 & 1 & & \\ & & 1 & 1 & \cdots & 1 & 1 & \\ & & & \ddots & \ddots & \ddots & \ddots & \ddots \\ & & & & 1 & 1 & \cdots & 1 & 1 \\ & & & & 1 & 1 & 1 & 1 & 1 \end{bmatrix}, \tag{4}$$

where $T$ is the size of the sliding window. *This one-sided filter involves the current and previous observations in the window, acknowledging that future observations are not available for filtering in practice.* Although a sliding-window average filter is used in this study, Eq. (3) is compatible with more advanced processing methods."

**(1.5) Line 228 of section "2.6.2 Simulation settings of atmospheric dispersion model": "sample"**

This term refers to an individual simulation using one of the candidate source locations. Therefore, each "sample" represents a simulated dispersion scenario with a different candidate source location. To clarify this point, we have replaced the term "sample" with "simulation" in the revised manuscript and explained the meaning of "sample".

►     Line 253-257 of section "2.6.2 Simulation settings of atmospheric dispersion model":

"To establish the datasets for the XGBoost model, *2000 simulations and 1000 simulations with different source locations were performed* by RIMPUFF for Oct. 3 and Oct. 4, respectively. *Candidate* source locations were *randomly* sampled from the shaded zones in Fig. 2a, which were determined according to the positions of the observation sites and the upwind direction. *Each simulation, along with its corresponding source location, forms one sample.*"

*(2)  Presentation of the machine learning method:*

We have added descriptions to the figure caption of the Fig. 1 and relabeled the root nodes using yellow squares in Fig. 1, providing a more accurate and detailed introduction to the decision tree model. To avoid confusion, we have replaced the symbol "$T$" with "$M$" in Eq. (7) and have provided additional descriptions for all the parameters.

►     Lines 149-163 of section "2.3 Source location estimation without knowing the exact release rates":

"where $K$ is the number of trees, $\mathcal{F} = \{f(x) = \boldsymbol{\omega}_{Q(x)}\}(Q: \mathbb{R}^p \to M, \boldsymbol{\omega} \in \mathbb{R}^M)$ is the space of *the* decision trees, and $Q$ represents the structure of each tree, mapping the feature vector to $M$ leaf nodes. Each $f_k$ corresponds to an independent tree structure $Q$ with leaf node weight*s* $\boldsymbol{\omega} = (\omega_1, \omega_2, \cdots, \omega_M)$. Equation (5) is then used to predict $\hat{\mathbf{r}}_i = (\hat{x}_i, \hat{y}_i)$ for the $i$-th sample.

[Figure]

**Figure 1.** Flowchart of XGBoost for predicting $\hat{\mathbf{r}}_i$ based on decision tree model. *The yellow squares are the root nodes within each tree, representing the input features in this paper. The purple ellipses denote the child nodes where the model evaluates input features and make decisions to split the data. The green rectangles depict the leaf nodes and refer to the prediction results. The vertical rectangles abstract the internal splitting processes of the trees, indicating decision-making not explicitly detailed in the diagram.*

XGBoost trains $G(\mathbf{X})$ in Eq. (5) by continuously fitting the residual error until the following objective function is minimized:

$$Obj^{(t)} = \sum_{i=1}^{n}\left(\mathbf{r}_i - \left(\hat{\mathbf{r}}_i^{(t-1)} + f_t(\mathbf{X}_i)\right)\right)^2 + \sum_{i=1}^{t}\Omega(f_i) , \tag{6}$$

where $t$ represents the training of the $t$-th tree and $\Omega(f_i)$ is the regularization term, given by:

$$\Omega(f) = \Upsilon M + \frac{1}{2}\lambda\sum_{j=1}^{M}\omega_j^2 , \tag{7}$$

*where M is the number of leaf nodes, $\omega_j$ is the leaf node weight for the j-th leaf node, and $\Upsilon$, $\lambda$ are penalty coefficients.* The minimization of Eq. (6) provides *a* parametric model $G(\mathbf{X})$ that maps the feature ensemble $\mathbf{X}$ extracted from $\boldsymbol{\mu}_p$ to the source location $\mathbf{r}$."

*(3) Dispersion model error considerations:*

We agree with your point regarding the impact of dispersion model errors. To address this issue, we have explored the sensitivity of the reconstruction to meteorological errors, which are an important source of uncertainties in atmospheric dispersion modeling. To simulate such uncertainties, a stochastic perturbation of ±10% is introduced to the observed wind speeds in the x and y components, and a ±1 stability class perturbation is applied to the stability parameters (e.g., from C to B or D). For both days, 50 meteorological groups are generated based on these random perturbations. For each group, the source location was estimated 50 times with randomly initialized hyperparameters. The results demonstrate that the source location estimates of Oct. 3 exhibit a low error level (generally below 10%), and the 50th error level is even lower than the error of the unperturbed results. In comparison, source location estimates of Oct. 4 are slightly more sensitive to the meteorological errors, which exhibit errors of around 10%–20%. The difference between the two days may results from the layout of the observation sites, suggesting that low error levels can be achieved with an appropriate site layout, even under certain meteorological errors. Previous studies have indicated that dispersion errors

such as wet deposition parameterization (Zhuang et al., 2023) may influence the dispersion simulation result; However, these errors are not dominant in the SCK-CEN [41]Ar and ETEX-1 field experiments. The handling of such dispersion model errors will be investigated in our future work, based on radionuclide leakage scenarios such as the 2017 [106]Ru event (Saunier et al., 2019). Detailed descriptions have been added in the revised manuscript.

► Lines 21-22 of section "Abstract":

"*With an appropriate site layout, low error levels can be achieved from only a single observation site or under meteorological errors.*"

► Lines 94-96 of section "1. Introduction":

"The sensitivity of the *source location estimation* to the spatial search range, size of the sliding window, feature type,  number and combination of sites, *and meteorological errors* is also investigated *for the SCK-CEN [41]Ar experiment.*"

► Lines 287-291 of section "2.7 Sensitivity study":

"*(5) Meteorological errors*

*Meteorological errors are important uncertainties in source reconstruction, especially the random errors in the wind field (Mekhaimr and Abdel Wahab, 2019). To simulate such uncertainties, a stochastic perturbation of ±10% is introduced to the observed wind speeds in the x and y components, and a ±1 stability class perturbation is applied to the stability parameters (e.g., from C to B or D). For both days, 50 meteorological groups are generated based on these random perturbations.*"

► Lines 550-567 of section "3.4 Sensitivity analysis results":

"*3.4.5 Sensitivity to the meteorological errors*

*Figure 14 illustrates the distribution of mean relative source location errors (averaged across 50 groups of hyperparameters) retrieved with 50 perturbed meteorological inputs. For Oct. 3, the estimates generally present a low error level (generally below 10%), and the 50th error level is lower than the error of the unperturbed results (4.68%). In comparison, for Oct. 4, most perturbed results exhibit larger errors (primarily 10%–20%) than the unperturbed result (4.71%), indicating that models for Oct. 4 are more sensitive to the meteorological errors. This sensitivity difference results from the layout of the observation sites (Fig. 2a). The sites on Oct. 3 were almost perpendicular to the prevailing wind direction, capturing the plume under a large range of wind directions. In contrast, the sites on Oct. 4 were basically parallel to the wind direction, capturing the plume only for a very limited range of wind directions. This result indicates the importance of site layout for robust reconstruction in the presence of meteorological errors. Feature selection slightly changes the mean relative error distribution and its percentiles for both days, indicating that meteorological errors may alter the importance of each feature and reduce the effectiveness of feature selection. In addition to meteorological errors, dispersion errors such as wet deposition parameterization (Zhuang et al., 2023) may influence the result, but these errors are not dominant in the two field experiments. The handling of such dispersion errors will be investigated in future work.*

[Figure]

**Figure 14.** *Sensitivity to the meteorological errors. The violin plots illustrate the kernel density estimation of errors under different meteorological groups for XGBoost models before and after feature selection. The vertical black lines inside the violins depict the interquartile range, capturing the 25th, 50th (red dots), and 75th percentiles of mean relative errors. The blue dots denote the mean relative source location errors for models without meteorological perturbation, as listed in Table 4.*"

*(4)  The testing of the method:*

To address your concern, we have added another validation based on the first release of the European Tracer Experiment (ETEX-1) (Nodop et al., 1998), which is continental scale. During ETEX-1, a total of 340 kg of perfluoromethylcyclohexane (PMCH) was released continuously into the atmosphere from 23 October 1994 at 16:00:00 UTC and 24 October at 1994 03:50:00 UTC. Assuming the release could have occurred between 23 October at 1994 00:00:00 UTC and 28 October 1994 at 00:00:00 UTC, it is viewed as a temporally-varying release, with a release rate of zero outside the actual release window. A total of 3104 available observations (3-h-averaged concentrations) were collected at 168 ground sites. As shown in Fig. 2b, we choose two groups of observation sites: the first comprises four sites (i.e. B05, D10, D16, F02) randomly selected from the sites within the sample zone (Group1, with a total of 92 available observations), and the second involves four sites (i.e. CR02, D15, DK08, S09) randomly selected from the sites beyond the sample zone boundaries (Group2, with a total of 90 available observations).

For the continental-scale ETEX-1 experiment, the proposed method still achieves the lowest source location errors among all methods, which are 5.19 km for Group 1 (a relative error of 0.20%) and 17.65 km for Group 2 (a relative error of 0.70%). Regarding the results of the uncertainty analysis, the mean relative source location errors are 2.42% and 4.97% for Group 1 and Group 2, respectively, lower than the correlation-based and Bayesian methods. The proposed method provides highly accurate mean estimates of release rate for both groups after feature selection, although with a large uncertainty range. These results demonstrate that spatiotemporally separated source reconstruction is feasible and achieves satisfactory accuracy in multi-scale release scenarios, thereby providing a promising framework for reconstructing atmospheric radionuclide releases.

We have provided an overview of the ETEX-1 experiment and source reconstruction results below. Detailed results

and discussions have been included in the revised manuscript.

►     Lines 15-20 of section "Abstract":

[revised manuscript text omitted]

”

"**Table S4.** *Hyperparameter optimization results of all 50 runs in Group 1 of ETEX-1 experiment.*

| Run | Hyperparameters | | | | | | | |
|-----|-----------|---------------|--------------|------------------|-----------|------------------|------------|---------|
| | max_depth | learning_rate | n_estimators | min_child_weight | subsample | colsample_bytree | reg_lambda | gamma |
| 1 | 3 | 0.07742 | 63 | 9 | 0.55953 | 0.44389 | 0.35824 | 0.81392 |
| 2 | 4 | 0.05011 | 142 | 3 | 0.70683 | 0.23027 | 4.89201 | 0.98193 |
| 3 | 3 | 0.06110 | 99 | 2 | 0.93364 | 0.37012 | 2.20274 | 0.76370 |
| 4 | 7 | 0.06117 | 64 | 10 | 0.55117 | 0.24878 | 2.34822 | 0.07471 |
| 5 | 3 | 0.07906 | 76 | 9 | 0.71631 | 0.38280 | 1.84573 | 0.22755 |
| 6 | 3 | 0.05067 | 116 | 4 | 0.70755 | 0.37613 | 1.50917 | 0.85795 |
| 7 | 3 | 0.06296 | 86 | 9 | 0.76834 | 0.34797 | 1.04674 | 0.44097 |
| 8 | 3 | 0.05342 | 106 | 5 | 0.53520 | 0.18199 | 4.18213 | 0.89476 |
| 9 | 3 | 0.07584 | 76 | 4 | 0.95540 | 0.53685 | 1.30937 | 0.49309 |
| 10 | 3 | 0.07213 | 84 | 5 | 0.98527 | 0.41048 | 2.61014 | 0.06896 |
| 11 | 3 | 0.05907 | 115 | 6 | 0.74495 | 0.37051 | 1.96059 | 0.27702 |
| 12 | 3 | 0.09471 | 76 | 3 | 0.91290 | 0.51041 | 0.38558 | 0.05232 |
| 13 | 3 | 0.07018 | 103 | 3 | 0.87242 | 0.35732 | 2.77176 | 0.16073 |
| 14 | 3 | 0.07072 | 95 | 10 | 0.98317 | 0.34966 | 4.67025 | 0.95006 |
| 15 | 3 | 0.08357 | 66 | 2 | 0.80913 | 0.37858 | 2.74202 | 0.05494 |
| 16 | 3 | 0.05001 | 121 | 3 | 0.52214 | 0.27101 | 4.30584 | 0.30632 |
| 17 | 3 | 0.09354 | 50 | 6 | 0.64736 | 0.62473 | 2.55863 | 0.35745 |
| 18 | 3 | 0.05486 | 134 | 8 | 0.66206 | 0.71278 | 0.80280 | 0.97413 |
| 19 | 3 | 0.07556 | 102 | 5 | 0.70927 | 0.34789 | 3.12167 | 0.99997 |
| 20 | 3 | 0.07479 | 52 | 9 | 0.79240 | 0.56720 | 1.02323 | 0.32951 |
| 21 | 3 | 0.05518 | 78 | 10 | 0.66309 | 0.99871 | 3.14571 | 0.84078 |
| 22 | 3 | 0.05139 | 111 | 6 | 0.73839 | 0.42603 | 2.49218 | 0.87318 |
| 23 | 3 | 0.10952 | 50 | 5 | 0.97390 | 0.22350 | 0.88047 | 0.35097 |

| | | | | | | | | |
|---|---|---|---|---|---|---|---|---|
| 24 | 3 | 0.06860 | 93 | 8 | 0.66447 | 0.33031 | 3.95098 | 0.67626 |
| 25 | 3 | 0.08670 | 54 | 5 | 0.79857 | 0.39303 | 3.19098 | 0.54197 |
| 26 | 3 | 0.08125 | 67 | 8 | 0.94963 | 0.33151 | 4.40350 | 0.06507 |
| 27 | 3 | 0.08396 | 50 | 2 | 0.93428 | 0.37792 | 3.77359 | 0.13881 |
| 28 | 3 | 0.05418 | 95 | 9 | 0.99598 | 0.25227 | 1.60204 | 0.38791 |
| 29 | 3 | 0.06935 | 89 | 8 | 0.90482 | 0.35876 | 4.15848 | 0.85423 |
| 30 | 3 | 0.06319 | 76 | 2 | 0.91583 | 0.43665 | 3.35600 | 0.88327 |
| 31 | 3 | 0.09009 | 50 | 4 | 0.94183 | 0.48645 | 4.03998 | 0.17582 |
| 32 | 3 | 0.06213 | 148 | 8 | 0.69235 | 0.34063 | 1.38470 | 0.71691 |
| 33 | 3 | 0.05735 | 74 | 10 | 0.83704 | 0.36175 | 2.28311 | 0.93893 |
| 34 | 3 | 0.06437 | 72 | 3 | 0.98748 | 0.31363 | 1.63480 | 0.22685 |
| 35 | 3 | 0.06022 | 60 | 6 | 0.66204 | 0.69317 | 1.21692 | 0.30900 |
| 36 | 3 | 0.09555 | 53 | 5 | 0.80980 | 0.46487 | 1.90000 | 0.60232 |
| 37 | 3 | 0.08434 | 50 | 8 | 0.52774 | 0.26641 | 0.48391 | 0.31574 |
| 38 | 3 | 0.07105 | 51 | 5 | 0.96131 | 0.63725 | 2.01205 | 0.60509 |
| 39 | 4 | 0.05005 | 62 | 10 | 0.77964 | 0.22026 | 3.55884 | 0.74839 |
| 40 | 4 | 0.07553 | 74 | 2 | 0.99323 | 0.36292 | 2.61782 | 0.17595 |
| 41 | 3 | 0.05239 | 77 | 7 | 0.50028 | 0.95751 | 2.41469 | 0.72211 |
| 42 | 3 | 0.08421 | 51 | 5 | 0.99977 | 0.45864 | 2.15063 | 0.54258 |
| 43 | 3 | 0.05414 | 82 | 8 | 0.62226 | 0.76122 | 2.83002 | 0.53414 |
| 44 | 3 | 0.05259 | 111 | 4 | 0.93432 | 0.32197 | 2.04760 | 0.44156 |
| 45 | 3 | 0.09454 | 60 | 4 | 0.72047 | 0.23879 | 4.65624 | 0.75740 |
| 46 | 3 | 0.05013 | 93 | 4 | 0.58250 | 0.40493 | 2.11383 | 0.47864 |
| 47 | 3 | 0.11330 | 50 | 2 | 0.71413 | 0.69524 | 3.50503 | 0.16269 |
| 48 | 3 | 0.08314 | 50 | 10 | 0.96691 | 0.50529 | 2.97909 | 0.95771 |
| 49 | 6 | 0.05978 | 107 | 7 | 0.63666 | 0.20488 | 0.61715 | 0.79254 |
| 50 | 3 | 0.07770 | 71 | 3 | 0.98943 | 0.58108 | 1.17867 | 0.22360 |

**Table S5.** *Hyperparameter optimization results of all 50 runs in Group 2 of ETEX-1 experiment.*

| Run | Hyperparameters | | | | | | | |
| --- | --- | --- | --- | --- | --- | --- | --- | --- |
| | *max_depth* | *learning_rate* | *n_estimators* | *min_child_weight* | *subsample* | *colsample_bytree* | *reg_lambda* | *gamma* |
| 1 | 8 | 0.17252 | 127 | 7 | 0.58651 | 0.60647 | 4.02984 | 0.60724 |
| 2 | 8 | 0.22988 | 68 | 9 | 0.54811 | 0.91321 | 2.29180 | 0.12852 |
| 3 | 8 | 0.10002 | 213 | 10 | 0.75238 | 0.64053 | 4.99653 | 0.10778 |
| 4 | 4 | 0.12366 | 147 | 5 | 0.87463 | 0.83119 | 0.63085 | 0.26995 |
| 5 | 5 | 0.23441 | 297 | 7 | 0.97266 | 0.36675 | 0.34489 | 0.99034 |
| 6 | 4 | 0.20533 | 155 | 4 | 0.63974 | 0.56526 | 2.63317 | 0.38177 |
| 7 | 5 | 0.10641 | 151 | 3 | 0.76094 | 0.79390 | 0.98025 | 0.96160 |
| 8 | 4 | 0.17290 | 222 | 4 | 0.55589 | 0.76284 | 1.62191 | 0.47379 |
| 9 | 4 | 0.05855 | 160 | 3 | 0.88818 | 0.79781 | 4.98019 | 0.84983 |
| 10 | 4 | 0.11741 | 184 | 5 | 0.79714 | 0.83203 | 4.43324 | 0.62777 |
| 11 | 4 | 0.16266 | 136 | 2 | 0.75690 | 0.93818 | 3.30854 | 0.55481 |
| 12 | 5 | 0.23134 | 144 | 2 | 0.58373 | 0.60423 | 2.76711 | 0.16986 |
| 13 | 8 | 0.27891 | 193 | 7 | 0.64977 | 0.89059 | 3.88204 | 0.23152 |
| 14 | 7 | 0.18603 | 245 | 8 | 0.77290 | 0.78709 | 3.45149 | 0.01806 |
| 15 | 5 | 0.16915 | 268 | 10 | 0.62385 | 0.49651 | 2.30355 | 0.27120 |
| 16 | 4 | 0.20217 | 64 | 2 | 0.92784 | 0.78470 | 0.94699 | 0.93657 |
| 17 | 5 | 0.10871 | 181 | 9 | 0.70489 | 0.84917 | 4.43678 | 0.07228 |
| 18 | 5 | 0.07394 | 297 | 8 | 0.87839 | 0.62200 | 3.24008 | 0.11160 |
| 19 | 6 | 0.20293 | 216 | 9 | 0.66381 | 0.89210 | 4.08151 | 0.60613 |
| 20 | 7 | 0.20570 | 158 | 4 | 0.50653 | 0.86393 | 3.36667 | 0.79227 |
| 21 | 5 | 0.20083 | 88 | 7 | 0.57460 | 0.62410 | 1.26707 | 0.17321 |
| 22 | 4 | 0.27072 | 50 | 4 | 0.85604 | 0.86560 | 0.16264 | 0.44052 |
| 23 | 8 | 0.15380 | 86 | 5 | 0.67811 | 0.74505 | 4.54334 | 0.93377 |
| 24 | 4 | 0.16205 | 183 | 6 | 0.59364 | 0.93969 | 1.05664 | 0.40669 |
| 25 | 6 | 0.14171 | 288 | 6 | 0.75389 | 0.85527 | 4.65363 | 0.50557 |
| 26 | 7 | 0.21287 | 253 | 9 | 0.59311 | 0.65113 | 2.79234 | 0.83703 |

| | | | | | | | | |
|---|---|---|---|---|---|---|---|---|
| 27 | 4 | 0.15371 | 247 | 5 | 0.77890 | 0.52357 | 4.81584 | 0.67752 |
| 28 | 5 | 0.11665 | 135 | 5 | 0.79729 | 0.86017 | 4.26743 | 0.12912 |
| 29 | 4 | 0.08378 | 192 | 4 | 0.52749 | 0.79980 | 2.64816 | 0.57092 |
| 30 | 6 | 0.13030 | 210 | 3 | 0.50209 | 0.61548 | 3.80894 | 0.64347 |
| 31 | 7 | 0.24148 | 173 | 10 | 0.64711 | 0.79358 | 2.66441 | 0.23023 |
| 32 | 6 | 0.09301 | 204 | 8 | 0.69879 | 0.97301 | 4.67770 | 0.36945 |
| 33 | 5 | 0.12318 | 283 | 6 | 0.93580 | 0.70267 | 2.23369 | 0.17565 |
| 34 | 4 | 0.23289 | 227 | 7 | 0.60924 | 0.76662 | 2.97809 | 0.22066 |
| 35 | 6 | 0.21219 | 162 | 3 | 0.54969 | 0.50796 | 4.01790 | 0.10632 |
| 36 | 5 | 0.10657 | 148 | 3 | 0.77407 | 0.84022 | 4.19435 | 0.53237 |
| 37 | 4 | 0.14220 | 169 | 4 | 0.69411 | 0.90516 | 2.46148 | 0.83182 |
| 38 | 6 | 0.11224 | 239 | 4 | 0.64335 | 0.91879 | 1.53421 | 0.43750 |
| 39 | 5 | 0.08990 | 98 | 5 | 0.83843 | 0.99546 | 3.80815 | 0.86071 |
| 40 | 6 | 0.19006 | 130 | 4 | 0.95749 | 0.88483 | 3.68950 | 0.17261 |
| 41 | 5 | 0.21434 | 93 | 6 | 0.80593 | 0.97025 | 2.23769 | 0.40479 |
| 42 | 7 | 0.07619 | 234 | 6 | 0.62146 | 0.77954 | 4.62217 | 0.85628 |
| 43 | 4 | 0.17377 | 273 | 6 | 0.85218 | 0.79578 | 3.43808 | 0.62076 |
| 44 | 4 | 0.18522 | 135 | 4 | 0.82615 | 0.63563 | 4.24215 | 0.56409 |
| 45 | 4 | 0.14993 | 152 | 8 | 0.60441 | 0.80580 | 2.50467 | 0.09351 |
| 46 | 5 | 0.15229 | 164 | 7 | 0.94667 | 0.83661 | 3.59476 | 0.15891 |
| 47 | 5 | 0.15393 | 116 | 9 | 0.90651 | 0.85377 | 4.60433 | 0.89894 |
| 48 | 5 | 0.22272 | 290 | 9 | 0.86799 | 0.85502 | 4.52637 | 0.79836 |
| 49 | 5 | 0.11275 | 91 | 4 | 0.72730 | 0.75528 | 3.72672 | 0.17298 |
| 50 | 4 | 0.13702 | 299 | 7 | 0.95702 | 0.91622 | 2.93120 | 0.22371 |

"

**Major points**

**Comment#1:**

Title: Both "Generalized" and "Spatiotemporally-decoupled" are not accurately reflecting the current two-step method.

The word "non-constant" in the title does not sound appropriate either. In reality, there are rarely constant releases. The author should reconsider the title.

**Response to comment#1:**

Thank you very much for the comment on the title. We have deleted the "non-constant" and "Generalized" in the title and have replaced the term "Spatiotemporally-decoupled" with "*Spatiotemporally separated*", to more accurately reflect the essence of the two-step method described in the manuscript. The revised title now reads: "*A spatiotemporally separated framework for reconstructing the source of atmospheric radionuclide releases.*"

**Comment#2:**

Abstract, lines 12-14: This statement is not accurate. The temporal variation of the release rates may be reflected on the plume shape, not only on the temporal variations of the observations. In theory, some problems cannot be decoupled. So the proposed method cannot be a real general framework. The limitation of the method has to be pointed out in the paper.

**Response to comment#2:**

*(1) Regarding the influence of the temporal variation of the release rates:*

We agree that the temporal variation of release rates influence the plume shape. This influence may be difficult to capture using only a limited number of observation sites, which is the case of SCK-CEN $^{41}$Ar experiment. For this reason, we focus on reducing the influence of temporal variations in the release rate on the observations, whereas the influence on the plume shape is not directly considered. However, our future efforts will be directed towards integrating spatial features to further enhance the method. The limitation and the future efforts have been pointed out in the section "4. Conclusions".

► Lines 598-599 of section "4. Conclusions":

"*However, the proposed method does not consider the influence of temporal variations in the release rate on the plume shape. Our future efforts will be directed towards integrating spatial features to further enhance the method.*"

*(2) Concerning whether the proposed method is truly a general framework:*

You raised an essential point about theoretical constraints where some problems cannot be decoupled. For these problems, our goal is to minimize the influence of temporal variations in the release rate on the observations, so that we can achieve spatiotemporally separated reconstruction. To eliminate ambiguity, we have restated the characteristics of the proposed method, using the term "*spatiotemporally separated*" rather than "spatiotemporally-decoupled" in the revised manuscript. The terms "decoupled" and "decoupling" in some sentences have been replaced by "*filtered*" and "*filtering*", respectively. Furthermore, to verify the applicability of the proposed method, we have also validated it using another field experiment at a different spatial scale, which have been presented in the responses to the **General comments**. This will help readers better understand the limitations and superiority of the proposed method and encourage further researches to overcome the constraints. Followed by comment#1 and comment#2, we have revised the abstract to ensure that it accurately reflects the updated focus of our study. Additionally, several relevant titles and sentences have been updated to ensure consistency with these modifications.

► Lines 8-13 of section "Abstract":

"*Determining the source location and release rate are critical *tasks* in assessing the environmental consequences of atmospheric radionuclide releases, but remain challenging because of the huge multi-dimensional solution space. We propose a *spatiotemporally separated* two-step framework *that reduces* the dimension of the solution space in each step and improves the *source* reconstruction accuracy. The *separation* process *applies* a temporal sliding-window average

filter to the observations, thereby reducing the influence of temporal variations in the release rate *on the observations* and ensuring that the features of the filtered data are dominated by the source location."

► Title of section 2.2 in line 109:

"2.2 *Observation filtering for spatiotemporally separated reconstruction*"

► Line 128 of section "2.2 Observation filtering for spatiotemporally separated reconstruction":

"where $\mathbf{\mu}_p$ refers to the *filtered* observations."

► Lines 132-133 of section "2.2 Observation filtering for spatiotemporally separated reconstruction":

"Although a sliding-window average filter is used in this study, Eq. (3) is compatible with more advanced processing methods."

► Lines 214-215 of section "2.5.3 Automatic optimization of XGBoost model":

"The overall flowchart of the proposed spatiotemporally *separated* source reconstruction model is shown in Fig. S1."

► Lines 277-278 of section "2.7 Sensitivity study":

"With these *filtered* data, the XGBoost model is trained using the same pattern for the *source location estimation*."

► Title of section 2.8.1 in line 296:

"2.8.1 *Observation filtering*"

► Lines 297-298 of section "2.8.1 Observation filtering":

"The feasibility of *filtering is* demonstrated using both the synthetic and real observations of the SCK-CEN [41]Ar experiment *and the real observations of the ETEX-1 experiment.*"

► Lines 303-304 of section "2.8.1 Observation filtering":

"The *filtering* performance is evaluated by comparing the simulation–observation differences before and after the *filtering* step."

► Title of section 3.1 in line 335:

"3.1 *Filtering performance*"

► Lines 337-343 of section "3.1 Filtering performance":

"Figure 3 compares the *filtering* performance for both the synthetic and real observations, *where* the constant-release simulations *are plotted* against the observations before and after *filtering*. For the synthetic observations, the *filtered* data are more concentrated along the 1:1 line for both days, and all *filtered* data fall within the 2-fold lines for Oct. 3. For the real observations, the dots before *filtering* in Fig. 3 have a dispersed distribution for both Oct. 3 and Oct. 4, indicating limited correlations with the simulations. After *filtering*, the dots are more concentrated *towards* the 1:1 line *for both the SCK-CEN [41]Ar and ETEX-1 experiments.* These phenomena indicate a noticeably increased agreement between the *filtered* observations and the constant-release simulations."

► Lines 348-353 of section "3.1 Filtering performance":

"*Table 3* quantitatively compares the *results* presented in Fig. 3. *For each case,* all metrics are greatly improved after *filtering*, confirming the better agreement between the *filtered* observations and the constant-release simulations. The improved agreement indicates that the *filtering* step significantly reduces the influence of temporal variations in release rates across the observations. The *filtering* performs better with the synthetic observations than with the real observations, because the synthetic observations are free of measurement errors. *The filtering process produces a better effect with the SCK-CEN [41]Ar experiment than with the ETEX-1 experiment, owing to the sparser observations in the ETEX-1 experiment (Fig. S3).*"

► Lines 515-517 of section "3.4.2 Sensitivity to the size of the sliding window":

"This is because a large window size increases the strength of the *filtering* and removes the temporal variations in the

release rates more completely."

► Lines 571-573 of section "4. Conclusions":

"Based on this, a more general spatiotemporally *separated* source reconstruction method was developed to estimate non-constant releases. *The separation process* was achieved by applying a temporal sliding-window average filter to the observations."

**Comment#3:**

Abstract, line 15: Locating a source location is not "localization". This needs to be corrected throughout the paper.

**Response to comment#3:**

We appreciate your comment on the use of the term "localization". We have replaced the relevant descriptions with "*source location estimation*" or "*locating the source*", which may precisely describe the process in our research. Accordingly, we have diligently revised the term throughout the paper to ensure accuracy and consistency.

► Lines 13-14 of section "Abstract":

"A machine learning model is trained to link these features to the source location, enabling independent *source location estimations*."

► Lines 85-86 of section "1. Introduction":

"Using this optimized model, the source *location is estimated* based on the filtered observations."

► Lines 92-96 of section "1. Introduction":

"The performance of the proposed method is compared with the correlation-based method *in terms of source location estimation* and the Bayesian method *in terms of* spatiotemporal accuracy. The sensitivity of the *source location estimation* to the spatial search range, size of the sliding window, feature type, and number and combination of sites is also investigated for SCK-CEN $^{41}$Ar experiment."

► Lines 123-125 of section "2.2 Observation filtering for spatiotemporally separated reconstruction":

"*By reducing the influence of the release rate,* the constant-release case can be approximated *and the sensitivity of the observations to the source location can be improved*, enabling separate *source location* and release rate estimation*s* and reducing the solution space at each step."

► Title of section 2.3 in line 134:

"2.3 *Source location estimation* without knowing the exact release rates"

► Lines 137-141 of section "2.3 Source location estimation without knowing the exact release rates":

"The meteorology is known, so it becomes possible to *locate* the source using the filtered observations. Nevertheless, the specificity of *source location estimation* methods that rely on direct observation–simulation comparisons may be substantially compromised because the peak amplitude is reduced. A better choice for *locating the source* would be to use the response features of the filtered observations, which preserve most of the location information."

► Lines 185-186 of section "2.5.1 Pre-screening of potential source locations":

"*Source locations corresponding to the highest 40% of correlation coefficients* are selected as the search range of the subsequent refined *source location estimation* using XGBoost."

► Lines 196-197 of section "2.5.3 Automatic optimization of XGBoost model":

"The XGBoost model for *source location estimation* is automatically optimized with respect to the hyperparameters and feature selection."

► Lines 276-278 of section "2.7 Sensitivity study":

"Temporal filtering with different sliding-window sizes is applied to *separate* the *source location estimation* from the

release rate estimation. In this study, the size of the sliding window ranges from 3–10. With these *filtered* data, the XGBoost model is trained using the same pattern for the *source location estimation*."

►     Lines 280-281 of section "2.7 Sensitivity study":

"The XGBoost model is trained using only time-domain features and only frequency-domain features to investigate the influence of these features on the *source location estimation*."

►     Lines 284-285 of section "2.7 Sensitivity study":

"The XGBoost model is trained and applied to the *source location estimation* with different numbers of observation sites, namely a single site, two sites, and three sites."

►     Lines 293-294 of section "2.7 Sensitivity study":

"The performance of *source location estimation* is compared quantitatively using the metrics specified in Sect. 2.8.3."

►     Line 314 of section "2.8.3 source reconstruction":

"The relative errors *in the* source *location* ($\delta_{\mathbf{r}}$) and total release ($\delta_Q$) are calculated to evaluate the source reconstruction accuracy:"

►     Lines 323-324 of section "2.8.4 Comparison with the Bayesian method":

"The proposed method is compared with the popular Bayesian method based on the SCK-CEN $^{41}$Ar *and ETEX-1 experiments*, with the same search range used for *locating the source* in both methods (Fig. 2)."

►     Line 357 of section "3.2.1 Hyperparameters":

"Table S1 summarizes the optimal hyperparameters and corresponding GCs used for *source location estimation* in this study;"

►     Title of section 3.3.1 in line 384:

"3.3.1 *Source locations*"

►     Lines 408-410 of section "3.3.1 Source locations":

"**Figure 5.** *Source location estimation* results *of SCK-CEN $^{41}$Ar experiment:* (a) Oct. 3; (b) Oct. 4*; and ETEX-1 experiment: (c) Group 1; (d) Group 2. A detailed enlargement of the region around (2.5°W, 47.5°N) to (1.5°W, 48.5°N) is shown in the bottom right corner in (c) and (d) to highlight the source location estimation results of the proposed method.*"

►     Lines 414-415 of section "3.3.2 Release rates":

"Figure 6 displays the release rates estimated by the Bayesian and PAMILT methods based on the *source location estimates* in Fig. 5."

►     Lines 458-460 of section "3.3.3 Uncertainty range":

"Feature selection improves the mean estimate and reduces the uncertainty range of PAMILT because it improves the *source location estimation*, thus reducing the deviation in the inverse model of the release rate."

►     Line 500 of section "3.4.1 Sensitivity to the search range":

"*Figure 10* displays the *source location* error*s* obtained using different pre-screening thresholds to determine the search range."

►     Line 512 of section "3.4.2 Sensitivity to the size of the sliding window":

"*Figure 11* shows the *source location* errors obtained with different sliding-window sizes."

►     Line 524 of section "3.4.3 Sensitivity to the feature type":

"For Oct. 3, the *source location* errors are quite low when using only the time-domain features for the reconstruction;"

►     Lines 527-530 of section "3.4.3 Sensitivity to the feature type":

"For Oct. 4, the mean *source location* errors are similar when using either the time- or frequency-domain features, but

the error range is higher when the frequency-domain features are used. In addition, the errors of both single-domain-feature results are higher than those of the all-feature results, indicating that both feature types should be *included* to ensure accurate and robust *source location estimation*."

► Lines 534-535 of section "3.4.4 Sensitivity to the number and combination of observation sites":

"The results indicate that the *source location* error may be more sensitive to the position of the observation site than to the number of sites included."

► Lines 576-578 of section "4. Conclusions":

"The XGBoost algorithm was used to train a machine learning model that links the source location to the feature vector, enabling independent *source location estimation* without knowing the release rate."

**Comment#4:**

Abstract, line 18: A relative error of about 50% for the Oct. 4 total release is probably not deemed "accurate". It is better to present the results more objectively with the actual number listed in Table 3.

**Response to comment#4:**

Thank you for your valuable feedback. In light of your suggestion, we have modified the abstract to more objectively reflect the results. In the revised abstract, the presented results are the averages of the two days for the SCK-CEN $^{41}$Ar experiment and the two groups for the ETEX-1 experiment, respectively.

► Lines 15-20 of section "Abstract":

"*This method is validated against the local-scale SCK-CEN $^{41}$Ar field experiment and the first release of the continental-scale European Tracer Experiment, for which the lowest relative source location errors are 0.60% and 0.20%, respectively. This presents higher accuracy and a smaller uncertainty range than the correlation-based and Bayesian methods in estimating the source location. The temporal variations in release rates are accurately reconstructed, and the mean relative errors of the total release are 65.09% and 72.14% lower than the Bayesian method for the SCK-CEN experiment and European Tracer Experiment, respectively.*"

**Comment#5:**

Line 94: The authors seem to suggest that the correlation-based method only applies when constant-release assumption is made. This is not accurate. Constant release is only one assumption that reduces the complexity of the problem. If the release starting time or duration is not known. Such assumption may not be enough to guarantee a unique solution of the source location. On the other hand, if a source is not constant, but the release time period and temporal profile are known, it is probably easy to get the source location even without the constant-release assumption.

**Response to comment#5:**

We agree with you that constant release is only one assumption that reduces the complexity of the problem and our descriptions need to be improved. To avoid confusion, we have deleted the term "(constant-release assumption)" in the revised manuscript. In the introduction section, we have emphasized that: the constant-release assumption may lead to inaccurate source location estimation, such as the case of the correlation-based method, because the constant-release assumption ignores the interaction between the time-varying release characteristics and non-stationary meteorological fields. We also agree that the source location can easily be estimated, if the release time period and temporal profile are known, even without the constant-release assumption. We did not mention this scenario, because this is not the focus of our study. Instead, we mainly consider atmospheric radionuclide releases where both the release time period and the temporal profile are unknown, which presents a more complex challenge for source location estimation.

► Lines 64-82 of section "1. Introduction":

"*Assumptions on the release characteristics aim to reduce the dimension of the solution space to 4 or 5, namely the two source location coordinates, the total release, and the release time (or the release start and end time), i.e. an instantaneous release at one time or constant release over a period (Kovalets et al., 2020, 2018; Efthimiou et al., 2018, 2017; Tomas et al., 2021; Andronopoulos and Kovalets, 2021; Ma et al., 2018). Under these assumptions, the correlation-based method exhibits high accuracy for ideal cases under stationary meteorological conditions, such as synthetic simulation experiments (Ma et al., 2018) and wind tunnel experiments (Kovalets et al., 2018; Efthimiou et al., 2017). However, previous studies have also demonstrated that real-world applications may be much more challenging, (Kovalets et al., 2020; Tomas et al., 2021; Andronopoulos and Kovalets, 2021; Becker et al., 2007) because the release usually exhibits temporal variations and may experience non-stationary meteorological fields. The interaction between the time-varying release characteristics and non-stationary meteorological fields is neglected in the instantaneous-release and constant-release assumptions, leading to inaccurate reconstruction.*

*Given the assumption-related reconstruction deviations in complex scenarios, we propose a spatiotemporally separated source reconstruction method that is less dependent on such assumptions. Our approach reduces the complexity of the source reconstruction using the simple fact that the source location is fixed during the atmospheric radionuclide release process. In this case, the spatiotemporal variations of observations are influenced by the time-varying release rate, source location, and meteorology, of which the last variable is generally known. The proposed method reduces the influence of the release rate through a temporal sliding-window average filter, making the filtered observations more sensitive to the source location than to the release rate. After filtering, existing methods based on direct observation–simulation comparisons may be unable to locate the source.*"

► Line 92-94 of section "1. Introduction":

"The performance of the proposed method is compared with the correlation-based method  *in terms of* source location estimation and the Bayesian method  *in terms of* spatiotemporal accuracy."

**Comment#6:**

Line 108: It is wrong to assume a square matrix. The dimensions of the observation and source vectors are independent and rarely the same.

**Response to comment#6:**

Thank you for pointing the error of the matrix. As you mentioned, the matrix $\mathbf{A(r)}$ is not a square matrix in general. We have modified the dimension of the matrix: $\mathbf{A(r)} = [A_1(\mathbf{r}), A_2(\mathbf{r}), \cdots, A_N(\mathbf{r})]^T \in \mathbb{R}^{N \times S}$, where $N$ is the number of sequential time steps and $S$ is the length of release rate vector $\mathbf{q}$.

► Lines 99-108 of section "2.1 Source reconstruction models":

"For an atmospheric radionuclide release, Eq. (1) relates the observations at each observation site to the source parameters:

$$\boldsymbol{\mu} = \mathbf{F}(\mathbf{r}, \mathbf{q}) + \boldsymbol{\varepsilon} , \tag{1}$$

where $\boldsymbol{\mu} = [\mu_1, \mu_2, \cdots, \mu_N]^T \in \mathbb{R}^N$ is an observation vector composed of observations at $N$ sequential time steps, the function $\mathbf{F}$ maps the source parameters to the observations, i.e. an atmospheric dispersion model, $\mathbf{r}$ refers to the source location, $\mathbf{q} \in \mathbb{R}^S$ is the temporally varying release rate, and $\boldsymbol{\varepsilon} \in \mathbb{R}^N$ is a vector containing both model and measurement errors.

In most source reconstruction models, $\mathbf{F}$ is simplified to the product of $\mathbf{q}$ and a source–receptor matrix $\mathbf{A}$ that depends

on the source location:

$$\boldsymbol{\mu} = \mathbf{A}(\mathbf{r})\mathbf{q} + \boldsymbol{\varepsilon} , \tag{2}$$

where $\boldsymbol{A}(\boldsymbol{r}) = [A_1(\boldsymbol{r}), A_2(\boldsymbol{r}), \cdots, A_N(\boldsymbol{r})]^T \in \mathbb{R}^{N \times S}$ and each row describes the sensitivity of an observation to the release rate $\mathbf{q}$ given the source location $\mathbf{r}$."

**Comment#7:**

Lines 122-124: The statement is not correct. The emissions combined with the meteorological conditions together determine the concentrations at any given measurement site, including the peak values and its timing.

**Response to comment#7:**

We agree that the emission and the meteorological jointly influence the peak values and its timing. Our work aims to smooth out the peak observations that is primarily shaped by the temporal release profile. We have restated our method based on this effect.

► Lines 121-125 of section "2.2 Observation filtering for spatiotemporally separated reconstruction":

"With a fixed source location, the release rate *and meteorology jointly determine the temporal variations of the observations (Li et al., 2019). The influence of meteorology can be pre-calculated as the source–receptor sensitivities and subsequently stored in matrix $\boldsymbol{A}(\boldsymbol{r})$. By reducing the influence of the release rate,* the constant-release case can be approximated *and the sensitivity of the observations to the source location can be improved*, enabling separate *source location* and release rate *estimations* and reducing the solution space at each step."

► Line 135 of section "2.3 Source location estimation without knowing the exact release rates":

"After applying the filter in Eq. (4), *the peak observations, primarily shaped by the temporal release profile, are smoothed out*."

**Comment#8:**

Line 228: It is very confusing to use "sample" for the different candidate source locations.

**Response to comment#8:**

We apologize for any confusion caused by the term "sample". This term refers to an individual simulation using one of the candidate source locations. Therefore, each "sample" represents a simulated dispersion scenario with a different candidate source location. To clarify this point, we have replaced the word "sample" with "simulation" in the revised manuscript.

► Lines 253-257 of section "2.6.2 Simulation settings of atmospheric dispersion model":

"To establish the datasets for the XGBoost model, *2000 simulations and 1000 simulations with different source locations were performed* by RIMPUFF for *the experiments on* Oct. 3 and Oct. 4, respectively. *Candidate* source locations were *randomly* sampled from the shaded zones in Fig. 2*(a)*, which were determined according to the positions of the observation sites and the upwind direction. *Each simulation, along with its corresponding source location, forms one sample*."

**Comment#9**

Lines 287-288: The hyper-parameters used in the 50 runs should be given in the supplementary document.

**Response to comment#9:**

We appreciate your suggestion regarding the inclusion of hyperparameters. We have provided all hyperparameters of the 50 runs in the revised version of the supplementary document.

► Tables S2-S3 of Supplementary Material (Lines 50-53):

| Run | Hyperparameters | | | | | | | |
| --- | --- | --- | --- | --- | --- | --- | --- | --- |
| | max_depth | learning_rate | n_estimators | min_child_weight | subsample | colsample_bytree | reg_lambda | gamma |
| 1 | 8 | 0.06963 | 257 | 2 | 0.58442 | 0.99833 | 0.56084 | 0.03347 |
| 2 | 8 | 0.05003 | 261 | 3 | 0.65774 | 0.76821 | 3.67031 | 0.40337 |
| 3 | 7 | 0.08651 | 246 | 4 | 0.76497 | 0.86844 | 1.82068 | 0.02107 |
| 4 | 3 | 0.10114 | 240 | 3 | 0.72483 | 0.99964 | 1.90473 | 0.40321 |
| 5 | 4 | 0.09505 | 299 | 4 | 0.86627 | 0.91229 | 2.72513 | 0.73273 |
| 6 | 6 | 0.12811 | 198 | 4 | 0.86151 | 0.91167 | 2.01739 | 0.64585 |
| 7 | 8 | 0.12143 | 160 | 5 | 0.76193 | 0.87631 | 1.32947 | 0.10111 |
| 8 | 4 | 0.10118 | 149 | 2 | 0.75307 | 0.47997 | 1.42352 | 0.32648 |
| 9 | 6 | 0.08344 | 203 | 2 | 0.73900 | 0.58051 | 4.27579 | 0.36316 |
| 10 | 6 | 0.08371 | 293 | 4 | 0.70524 | 0.50305 | 1.58921 | 0.90349 |
| 11 | 6 | 0.08044 | 203 | 7 | 0.74233 | 0.83712 | 1.87067 | 0.66921 |
| 12 | 8 | 0.08750 | 298 | 5 | 0.73452 | 0.95439 | 3.24463 | 0.23793 |
| 13 | 8 | 0.12917 | 218 | 5 | 0.64402 | 0.57828 | 0.79801 | 0.43434 |
| 14 | 5 | 0.09389 | 279 | 2 | 0.97316 | 0.80680 | 1.61133 | 0.05062 |
| 15 | 8 | 0.14586 | 255 | 9 | 0.54883 | 0.74530 | 3.62691 | 0.21478 |
| 16 | 8 | 0.09194 | 160 | 3 | 0.59974 | 0.83406 | 0.33249 | 0.18032 |
| 17 | 4 | 0.08920 | 257 | 2 | 0.67346 | 0.99730 | 0.98970 | 0.17230 |
| 18 | 4 | 0.09419 | 294 | 4 | 0.79714 | 0.87812 | 3.77772 | 0.88406 |
| 19 | 5 | 0.07604 | 299 | 3 | 0.79858 | 0.83297 | 0.36589 | 0.27070 |
| 20 | 6 | 0.08451 | 231 | 3 | 0.76571 | 0.89974 | 2.67871 | 0.31997 |
| 21 | 5 | 0.15257 | 203 | 3 | 0.83687 | 0.94582 | 1.67365 | 0.06759 |
| 22 | 4 | 0.14711 | 180 | 3 | 0.82554 | 0.79287 | 1.10286 | 0.31295 |
| 23 | 5 | 0.08729 | 285 | 2 | 0.67684 | 0.91908 | 0.81695 | 0.76090 |
| 24 | 8 | 0.09440 | 235 | 2 | 0.66775 | 0.88929 | 4.40930 | 0.04806 |
| 25 | 7 | 0.10085 | 216 | 3 | 0.58725 | 0.68488 | 1.70407 | 0.25164 |

| Run | max_depth | learning_rate | n_estimators | min_child_weight | subsample | colsample_bytree | reg_lambda | gamma |
|---|---|---|---|---|---|---|---|---|
| 26 | 8 | 0.09937 | 200 | 3 | 0.83402 | 0.78555 | 3.59830 | 0.55999 |
| 27 | 3 | 0.12772 | 189 | 6 | 0.75408 | 0.99256 | 1.67164 | 0.24484 |
| 28 | 8 | 0.10973 | 183 | 2 | 0.50393 | 0.53818 | 0.67395 | 0.18678 |
| 29 | 6 | 0.09468 | 185 | 2 | 0.59535 | 0.75381 | 2.10634 | 0.48731 |
| 30 | 6 | 0.09652 | 247 | 3 | 0.69860 | 0.95369 | 0.05146 | 0.48637 |
| 31 | 7 | 0.06846 | 185 | 5 | 0.52549 | 0.61305 | 0.97320 | 0.17339 |
| 32 | 8 | 0.09323 | 215 | 4 | 0.74269 | 0.98432 | 4.30255 | 0.28215 |
| 33 | 7 | 0.09339 | 299 | 6 | 0.61681 | 0.49190 | 2.27687 | 0.96352 |
| 34 | 7 | 0.11693 | 234 | 3 | 0.74464 | 0.54387 | 1.02597 | 0.63504 |
| 35 | 4 | 0.06858 | 277 | 2 | 0.74264 | 0.92278 | 1.30424 | 0.81347 |
| 36 | 5 | 0.08068 | 246 | 5 | 0.73714 | 0.99006 | 1.39783 | 0.27963 |
| 37 | 8 | 0.08645 | 274 | 4 | 0.82352 | 0.99618 | 3.59875 | 0.82528 |
| 38 | 7 | 0.18011 | 226 | 8 | 0.66425 | 0.81094 | 0.98036 | 0.11274 |
| 39 | 5 | 0.08397 | 212 | 3 | 0.62934 | 0.45110 | 1.94896 | 0.64913 |
| 40 | 5 | 0.07800 | 228 | 2 | 0.66806 | 0.91700 | 0.32409 | 0.53206 |
| 41 | 6 | 0.07905 | 231 | 4 | 0.63064 | 0.93657 | 0.01082 | 0.03863 |
| 42 | 3 | 0.09277 | 261 | 3 | 0.72093 | 0.96486 | 1.73917 | 0.39009 |
| 43 | 5 | 0.08732 | 217 | 2 | 0.81405 | 0.78575 | 1.71376 | 0.85775 |
| 44 | 5 | 0.08417 | 225 | 3 | 0.61443 | 0.61703 | 2.06192 | 0.93001 |
| 45 | 8 | 0.14916 | 188 | 4 | 0.71686 | 0.87552 | 0.21908 | 0.58120 |
| 46 | 6 | 0.10745 | 179 | 3 | 0.82311 | 0.92434 | 3.99176 | 0.29124 |
| 47 | 4 | 0.13632 | 252 | 4 | 0.83077 | 0.92543 | 3.17264 | 0.31258 |
| 48 | 8 | 0.12402 | 176 | 4 | 0.70048 | 0.75866 | 3.18949 | 0.92647 |
| 49 | 8 | 0.07057 | 283 | 4 | 0.62353 | 0.39145 | 0.71074 | 0.47779 |
| 50 | 5 | 0.11104 | 197 | 3 | 0.79114 | 0.86436 | 3.16004 | 0.19049 |

**Table S3.** *Hyperparameter optimization results of all 50 runs in Oct 4 of SCK-CEN $^{41}$Ar experiment.*

| Run | Hyperparameters | | | | | | | |
|---|---|---|---|---|---|---|---|---|
| | max_depth | learning_rate | n_estimators | min_child_weight | subsample | colsample_bytree | reg_lambda | gamma |

| | | | | | | | | |
|---|---|---|---|---|---|---|---|---|
| 1 | 5 | 0.07095 | 242 | 10 | 0.53823 | 0.98511 | 3.43106 | 0.47567 |
| 2 | 6 | 0.13148 | 121 | 10 | 0.52493 | 0.91239 | 4.40584 | 0.19543 |
| 3 | 7 | 0.15575 | 102 | 9 | 0.50009 | 0.80159 | 1.43484 | 0.98014 |
| 4 | 7 | 0.10178 | 245 | 9 | 0.50621 | 0.55022 | 2.84160 | 0.78872 |
| 5 | 6 | 0.15994 | 88 | 9 | 0.54667 | 0.93870 | 0.68681 | 0.82277 |
| 6 | 6 | 0.07483 | 221 | 7 | 0.50084 | 0.97352 | 3.24507 | 0.65469 |
| 7 | 7 | 0.08191 | 179 | 8 | 0.50132 | 0.99302 | 4.00168 | 0.63458 |
| 8 | 5 | 0.05154 | 256 | 10 | 0.50279 | 0.97827 | 1.93594 | 0.01850 |
| 9 | 8 | 0.05644 | 294 | 10 | 0.56394 | 0.84780 | 2.05943 | 0.27115 |
| 10 | 4 | 0.07670 | 281 | 10 | 0.53927 | 0.97848 | 4.82450 | 0.27244 |
| 11 | 6 | 0.13550 | 112 | 10 | 0.53406 | 0.53254 | 2.40913 | 0.53195 |
| 12 | 6 | 0.08587 | 203 | 10 | 0.63046 | 0.84871 | 4.82865 | 0.96621 |
| 13 | 7 | 0.09585 | 288 | 10 | 0.50229 | 0.99575 | 4.49999 | 0.55975 |
| 14 | 7 | 0.08191 | 245 | 10 | 0.54189 | 0.87572 | 3.95663 | 0.85872 |
| 15 | 7 | 0.07474 | 174 | 7 | 0.51174 | 0.96561 | 2.97146 | 0.92806 |
| 16 | 5 | 0.08019 | 212 | 9 | 0.50228 | 0.98434 | 4.45686 | 0.41716 |
| 17 | 8 | 0.07642 | 205 | 9 | 0.52041 | 0.84692 | 4.46048 | 0.41196 |
| 18 | 8 | 0.08315 | 218 | 10 | 0.56530 | 0.92783 | 4.49138 | 0.69385 |
| 19 | 7 | 0.09760 | 171 | 10 | 0.57999 | 0.74319 | 0.76715 | 0.72171 |
| 20 | 5 | 0.10593 | 142 | 9 | 0.52512 | 0.99411 | 0.89520 | 0.27131 |
| 21 | 8 | 0.07079 | 185 | 9 | 0.50469 | 0.87378 | 1.08559 | 0.25444 |
| 22 | 6 | 0.11366 | 183 | 9 | 0.57435 | 0.77739 | 3.16044 | 0.93374 |
| 23 | 6 | 0.11157 | 254 | 9 | 0.52017 | 0.97489 | 1.90816 | 0.79666 |
| 24 | 7 | 0.11255 | 188 | 8 | 0.50796 | 0.74881 | 1.66455 | 0.77696 |
| 25 | 7 | 0.18193 | 79 | 10 | 0.68663 | 0.99816 | 2.79139 | 0.92738 |
| 26 | 5 | 0.20795 | 137 | 10 | 0.60099 | 0.37442 | 4.72568 | 0.01013 |
| 27 | 5 | 0.08039 | 208 | 10 | 0.55245 | 0.85163 | 2.68594 | 0.57202 |
| 28 | 4 | 0.13824 | 232 | 9 | 0.53167 | 0.97794 | 4.99790 | 0.72989 |

| | | | | | | | | |
|---|---|---|---|---|---|---|---|---|
| 29 | 5 | 0.11709 | 264 | 10 | 0.50079 | 0.65333 | 4.99177 | 0.01211 |
| 30 | 7 | 0.08384 | 183 | 9 | 0.54315 | 0.85012 | 2.95216 | 0.68107 |
| 31 | 7 | 0.07624 | 243 | 10 | 0.51390 | 0.68864 | 2.39622 | 0.79548 |
| 32 | 8 | 0.12670 | 219 | 9 | 0.51735 | 0.51438 | 4.86510 | 0.39015 |
| 33 | 7 | 0.17030 | 133 | 8 | 0.50620 | 0.74374 | 4.58307 | 0.02592 |
| 34 | 6 | 0.17322 | 101 | 7 | 0.54173 | 0.74255 | 4.24794 | 0.97291 |
| 35 | 4 | 0.09476 | 222 | 10 | 0.50025 | 0.77689 | 4.46467 | 0.83712 |
| 36 | 7 | 0.10917 | 175 | 9 | 0.52491 | 0.91604 | 2.16957 | 0.73717 |
| 37 | 7 | 0.08104 | 168 | 10 | 0.50263 | 0.50745 | 3.81626 | 0.69286 |
| 38 | 5 | 0.13034 | 245 | 10 | 0.50188 | 0.76513 | 2.11312 | 0.03408 |
| 39 | 8 | 0.08419 | 196 | 10 | 0.55446 | 0.85748 | 4.85152 | 0.63630 |
| 40 | 8 | 0.08592 | 205 | 10 | 0.52833 | 0.88829 | 2.58534 | 0.47814 |
| 41 | 6 | 0.16963 | 80 | 8 | 0.56274 | 0.94076 | 3.32882 | 0.66880 |
| 42 | 7 | 0.14335 | 123 | 8 | 0.56362 | 0.91686 | 4.92034 | 0.02893 |
| 43 | 8 | 0.12178 | 183 | 10 | 0.58719 | 0.86078 | 3.42019 | 0.41184 |
| 44 | 7 | 0.14413 | 185 | 10 | 0.52721 | 0.57415 | 2.30624 | 0.51660 |
| 45 | 6 | 0.08748 | 221 | 10 | 0.50936 | 0.70774 | 2.06636 | 0.28648 |
| 46 | 7 | 0.08946 | 176 | 10 | 0.55001 | 0.92642 | 3.51959 | 0.19652 |
| 47 | 7 | 0.09190 | 157 | 8 | 0.63144 | 0.74802 | 0.11220 | 0.61326 |
| 48 | 5 | 0.13140 | 200 | 10 | 0.51145 | 0.69994 | 3.88528 | 0.53884 |
| 49 | 5 | 0.12585 | 187 | 10 | 0.50120 | 0.93257 | 2.44567 | 0.98524 |
| 50 | 6 | 0.07366 | 250 | 9 | 0.53270 | 0.95103 | 1.97326 | 0.23232 |

"

**Comment#10:**

Figure S3: What is the sliding window applied here? It does seem to be a sided window rather than centered one. Please explain this in the paper.

**Response to comment#10:**

We apologize for the lack of clarity regarding the sliding window used in Figure S3. As outlined in Eq. (4), a one-sided window is employed. This one-sided temporal sliding-window average filter involves the current and previous observations in the window, acknowledging that future observations are not available for filtering in practice. Compared

to the centered window, the one-sided window excels in real-time data processing and rapid response to changes in the observations, making it more suitable for real applications. Relevant descriptions of the sliding window have been modified in the revised manuscript for clarity.

► Line 128-133 of section "2.2 Observation filtering for spatiotemporally separated reconstruction":

"In this study, the following operator matrix is constructed to impose a *one-sided* temporal sliding-window average filter (Eamonn Keogh, Selina Chu, 2004):

$$
\mathbf{P} = \frac{1}{T}
\begin{bmatrix}
1 & & & & & & & & \\
1 & 1 & & & & & & & \\
 & \vdots & & & & & & & \\
1 & 1 & \cdots & 1 & & & & & \\
1 & 1 & \cdots & 1 & 1 & & & & \\
 & 1 & 1 & \cdots & 1 & 1 & & & \\
 & & 1 & 1 & \cdots & 1 & 1 & & \\
 & & & \ddots & \ddots & \ddots & \ddots & \ddots & \\
 & & & & 1 & 1 & \cdots & 1 & 1 \\
 & & & & 1 & 1 & 1 & 1 & 1
\end{bmatrix},
\tag{4}
$$

where $T$ is the size of the sliding window. *This one-sided filter involves the current and previous observations in the window, acknowledging that future observations are not available for filtering in practice.* Although a sliding-window average filter is used in this study, Eq. (3) is compatible with more advanced processing methods."

**Minor points**

**Comment#1:**

Line 46, T3-10: Please explain what T3-10 distributions are.

**Response to comment#1:**

The notation T3-10 denotes a Student's $t$-distribution with degrees of freedom ranging from 3 to 10, as referenced in (Wang et al., 2017). The $t$-distribution (also known as $t_\nu$) is applicable for estimating the mean of a normally distributed population, when the sample size is small and the population standard deviation is unknown. In this distribution, the parameter $\nu$ represents the degrees of freedom and determines the distribution's shape. As $\nu$ increases, the $t$-distribution approaches the normal distribution. To eliminate the ambiguity, we have replaced "T3-10" with "*t-distribution (with degrees of freedom ranging from 3–10)*".

► Lines 48-51 of section "1. Introduction":

"Other *candidates* include *the t-distribution (with degrees of freedom ranging from 3–10), Cauchy distribution, and log-Cauchy distribution*, *all of* which *were* compared *against the* normal and log-normal distributions in *terms of* reconstructing the source parameters of the Prairie Grass field experiment (Wang et al., 2017)."

**Comment#2:**

Line 60: What does "deterministic assumption" mean? It is quite confusing.

**Response to comment#2:**

We apologize for any confusion caused by the term "deterministic assumption". Deterministic assumptions aim to define the physical feature of source parameters. A typical one is the constant-release assumption, which assumes that the substances are released at a constant rate during the release period (Kovalets et al., 2020, 2018; Efthimiou et al., 2018, 2017; Tomas et al., 2021; Andronopoulos and Kovalets, 2021; Ma et al., 2018). To avoid confusion, we have

replaced the terms "Statistical assumption" and "Deterministic assumption" with "*Assumptions on the model–observation discrepancies*" and "*Assumptions on the release characteristics*", respectively, in the introduction section in the revised manuscript.

► Lines 41-44 of section "1. Introduction":

"To reduce the problem of ill-posedness, most previous studies have attempted to constrain the reconstruction by imposing assumptions on *the model–observation discrepancies or release characteristics. Assumptions on model–observation discrepancies* are widely used in Bayesian methods to simultaneously reconstruct the posterior distributions of spatiotemporal source parameters (De Meutter et al., 2021; Meutter and Hoffman, 2020; Xue et al., 2017)."

► Lines 64-74 of section "1. Introduction":

"*Assumptions on the release characteristics aim to reduce the dimension of the solution space to 4 or 5, namely the two source location coordinates, the total release, and the release time (or the release start and end time), i.e. an instantaneous release at one time or constant release over a period (Kovalets et al., 2020, 2018; Efthimiou et al., 2018, 2017; Tomas et al., 2021; Andronopoulos and Kovalets, 2021; Ma et al., 2018). Under these assumptions, the correlation-based method exhibits high accuracy for ideal cases under stationary meteorological conditions, such as synthetic simulation experiments (Ma et al., 2018) and wind tunnel experiments (Kovalets et al., 2018; Efthimiou et al., 2017). However, previous studies have also demonstrated that real-world applications may be much more challenging, (Kovalets et al., 2020; Tomas et al., 2021; Andronopoulos and Kovalets, 2021; Becker et al., 2007) because the release usually exhibits temporal variations and may experience non-stationary meteorological fields. The interaction between the time-varying release characteristics and non-stationary meteorological fields is neglected in the instantaneous-release and constant-release assumptions, leading to inaccurate reconstruction.*"

**Comment#3:**

Figure 1: What do the different shapes and colors in the diagram mean?

**Response to comment#3:**

We apologize for not providing detailed explanations for the shapes and colors in Figure 1. We have added descriptions to the figure caption of the Fig. 1 and relabeled the root nodes using yellow squares in Fig. 1, providing a more accurate and detailed introduction to the decision tree model.

► Line 152-156 of section "2.3 Source location estimation without knowing the exact release rates":

[Figure]

"

**Figure 1.** Flowchart of XGBoost for predicting $\hat{\mathbf{f}}_i$ based on decision tree model. *The yellow squares are the root nodes within each tree, representing the input features in this paper. The purple ellipses denote the child nodes where the model evaluates input features and make decisions to split the data. The green rectangles depict the leaf nodes and refer to the prediction results. The vertical rectangles abstract the internal splitting processes of the trees, indicating decision-making not explicitly detailed in the diagram.*"

**Comment#4:**

Equation (7): Please explain all the parameters here.

**Response to comment#4:**

We appreciate your attention to Equation (7). To avoid confusion, we have replaced the symbol "$T$" with "$M$" in Eq. (7) and have provided additional descriptions for all the parameters.

► Lines 149-151 of section "2.3 Source location estimation without knowing the exact release rates":

"where $K$ is the number of trees, $\mathcal{F} = \{f(x) = \boldsymbol{\omega}_{Q(x)}\}(Q: \mathbb{R}^p \rightarrow M, \boldsymbol{\omega} \in \mathbb{R}^M)$ is the space of *the* decision trees, and $Q$ represents the structure of each tree, mapping the feature vector to $M$ leaf nodes. Each $f_k$ corresponds to an independent tree structure $Q$ with leaf node weight*s* $\boldsymbol{\omega} = (\omega_1, \omega_2, \cdots, \omega_M)$."

► Lines 159-163 of section "2.3 Source location estimation without knowing the exact release rates":

"where $t$ represents the training of the $t$-th tree and $\Omega(f_i)$ is the regularization term, given by:

$$\Omega(f) = \Upsilon M + \frac{1}{2}\lambda \sum_{j=1}^{M} \omega_j^2 , \tag{7}$$

*where $M$ is the number of leaf nodes, $\omega_j$ is the leaf node weight for the j-th leaf node, and $\Upsilon$, $\lambda$ are penalty coefficients.* The minimization of Eq. (6) provides *a* parametric model $G(\mathbf{X})$ that maps the feature ensemble $\mathbf{X}$ extracted from $\boldsymbol{\mu}_p$ to the source location $\mathbf{r}$."

**Comment#5:**

Line 160: Why is the amplitude quantity called "wave rate"?

**Response to comment#5:**

We apologize for the unclear definition. We aim to define the "wave_rate" as a statistical measure that quantifies the fluctuations of $\boldsymbol{\mu}_p$ over time. To reduce the impact of extreme values, the "wave rate" is calculated as the difference between the 90th and 10th quantiles of the normalized observation series. To avoid any ambiguity, we have removed the term "amplitude" from the revised manuscript and clarified the definition of "wave_rate" to ensure it accurately reflects the intended concept.

► Lines 165-167 of section "2.3 Source location estimation without knowing the exact release rates":

"Among the time-domain features, the wave rate quantifies *the fluctuations of $\boldsymbol{\mu}_p$ over time*, while the temporal mean and median values *are measures of the central tendency of $\boldsymbol{\mu}_p$ (Witte and Witte, 2017)*."

**Comment#6:**

Lines 160-161: The median value is not a central moment.

**Response to comment#6:**

We appreciate your observation regarding the classification of median value. Upon reviewing relevant literature (Witte and Witte, 2017), the temporal mean and median values are indeed recognized measures of central tendency in statistical analysis. We have made revisions in the manuscript to clarify this point.

► Lines 165-167 of section "2.3 Source location estimation without knowing the exact release rates":

"Among the time-domain features, the wave rate quantifies *the fluctuations of $\mu_p$ over time*, while the temporal mean and median values *are measures of the central tendency of $\mu_p$ (Witte and Witte, 2017).*"

**Comment#7:**

Line 232: If it is 40[th] percentile, the number of samples for Oct.3 and Oct. 4 should be 1200 and 600.

**Response to comment#7:**

We appreciate your careful review and for identifying this discrepancy. Indeed, we intended to reference the 60[th] percentile, not the 40[th]. To clarify this point, we have revised our descriptions and instead specified that source locations corresponding to the highest 40% of correlation coefficients are selected for further analysis.

► Lines 185-186 of section "2.5.1 Pre-screening of potential source locations":

"*Source locations corresponding to the highest 40% of correlation coefficients* are selected as the search range of the subsequent refined *source location estimation* using XGBoost."

► Lines 257-258 of section "2.6.2 Simulation settings of atmospheric dispersion model":

"As described in Sect. 2.5.1, we calculated the correlation coefficient for each sample and preserved *the 40% of samples with the highest 40% of correlation coefficients* (i.e. 800 samples for Oct. 3 and 400 samples for Oct. 4)."

**Comment#8:**

Line 237: The authors probably mean 80[th], 60[th], 50[th], 40[th], 20[th], and 0[th].

**Response to comment#8:**

We appreciate your attention to this detail. You are correct. In line with the response to comment#7, we have revised the relevant descriptions to accurately reflect the search range.

► Lines 271-274 of section "2.7 Sensitivity study":

"The search range is controlled by the pre-screening threshold*, which is the top proportion of the correlation coefficients in the pre-screening step. Specifically, we use source locations corresponding to the highest 20%, 40%, 50%, 60%, 80%, and 100% of correlation coefficients to define the search ranges, with a lower proportion indicating a narrower and more focused search area.*"

**Comment#9:**

Line 238: "A lower percentile" should be "a higher percentile".

**Response to comment#9:**

We appreciate your attention in identifying this discrepancy. We have corrected this error in the revised manuscript, which is consistent with the revisions discussed in comment#7 and comment#8.

► Lines 272-274 of section "2.7 Sensitivity study":

"*Specifically, we use source locations corresponding to the highest 20%, 40%, 50%, 60%, 80%, and 100% of correlation coefficients to define the search ranges, with a lower proportion indicating a narrower and more focused search area.*"

**Comment#10:**

Figure 8: No shade appears for the Bayesian inversion results in the lower left panel.

**Response to comment#10:**

Thank you for pointing out the visualization issue regarding Fig. 8. To resolve this problem, we have enlarged the

shading in this area for better visualization. Additionally, we have adjusted the shading range from [minimum, maximum] to [lower quartile, upper quartile] to better represent the results.

► Figure 8 of section "3.3.3 Uncertainty range" (Lines 463-467):

[Figure]

"**Figure 8.** Release rate estimates over 50 calculations *of SCK-CEN* [41]*Ar experiment*. *(a) Oct. 3-Bayesian method; (b) Oct. 3-PAMILT method; (c) Oct. 4-Bayesian method; (d) Oct. 4-PAMILT method. The shadow represents the uncertainty range between the lower quartile and the upper quartile. The shadow of each figure is amplified by an enlarged subgraph. The legends in each figure provide the mean estimates for the total release.*"

**Comment#11:**

Line 416: What do the various pre-screening ranges refer to?

**Response to comment#11:**

We apologize for any confusion caused by the term "pre-screening ranges". These pre-screening ranges, as detailed in Sect. 2.5.1, refer to the specific subsets of source locations selected based on their correlation coefficients (i.e. search range). The pre-screening process is designed to reduce computational costs and eliminate low-quality samples by focusing on the most promising source locations for further analysis. To eliminate the ambiguity, we have replaced the "pre-screening ranges" with "*search ranges*".

► Line 501 of section "3.4.1 Sensitivity to the search range":

"The error is smaller with a lower threshold, implying that a small *search range* helps reduce the mean and median errors."

► Lines 512-514 of section "3.4.2 Sensitivity to the size of the sliding window":

"The mean/median error is less than 8% for Oct. 3 and less than 11% for Oct. 4, both of which are smaller than for the various *search ranges* in Fig. 9. This indicates that the proposed method is more robust to this parameter than to the *search range*."

**Comment#12:**

Figure S1: Should it be 20% instead of 10% for the five-fold cross-validation?

**Response to comment#12:**

  We appreciate your attention to detail in Figure S1. You are correct about the discrepancy in the percentage for the five-fold cross-validation; it should be 20% instead of 10%. We have made this correction in the revised figure.

►  Figure S1 of Supplementary Material (Lines 29-30):

[Figure]

**Figure S1.** Flowchart of the proposed *spatiotemporally separated* source reconstruction method."

**Comment#13:**

Table S1: Brief descriptions of the hyperparameters should be provided.

**Response to comment#13:**

  We appreciate your suggestion regarding Table S1. Brief descriptions of the hyperparameters have been included in the caption of Table S1 for clarity:

►  Lines 45-48 of Supplementary Material:

[revised manuscript text omitted]

**Referee #2**

**General comments**

This study proposes a novel approach to the source reconstruction of atmospheric radionuclide emissions in non-stationary emission scenarios. By moving away from the unrealistic assumption of constant emissions and developing a method for spatiotemporally decoupled source reconstruction, it effectively leverages the fact that variations in emission rates significantly impact observations. The methodology involves training machine learning models with the XGBoost algorithm and determining detailed temporal variations in emission rates using the PAMILT algorithm. The paper makes a significant contribution to the field of atmospheric radionuclide emission source reconstruction. The proposed methodology offers an effective means for accurately localizing sources and estimating emission rates in non-stationary scenarios, presenting a promising framework for future research and applications.

[1] The utilization of a temporal sliding-window average filter is commendable. However, elucidating the criteria for feature selection and the impact of varying combinations of observation sites on source estimation would enhance the paper.

[2] Validating the proposed method against the SCK-CEN $^{41}$Ar field experiment data underscores its efficacy and applicability. Nonetheless, conducting further validation studies under diverse scenarios and conditions would enrich our understanding of the method's applicability and limitations. It is recommended to include additional case studies involving different types of releases and weather conditions to assess the method's efficiency and adaptability more comprehensively.

**Response to general comments:**

Thank you for your valuable feedback and suggestive comments on our manuscript, which not only recognizes the innovation and contribution of our approach but also highlights areas for further enhancement of our work. Below are our responses to your main points:

*(1) The criteria for feature selection and the impact of varying combinations of observation sites*
*(1.1) The criteria for feature selection*

The mean cross-validation score (MCV) is used as the criterion for feature selection, and the optimal feature subset is selected as the one that achieves the highest MCV. This selection is implemented by recursively removing the feature with the least importance, and assessing the MCV based on cross validation (Akhtar et al., 2019). It starts with training a XGBoost model with all features, and assessing the importance of each feature based on its contribution to the accuracy of the XGBoost model. Then, the feature with the least importance is removed and the XGBoost model is retrained using the remaining features. The feature importance and MCV are updated accordingly for the removal of another feature. This iteration continues until the optimal number of features is identified, corresponding to the highest MCV achieved during the process.

Using this criterion, unimportant features can be removed to improve the XGBoost model's prediction accuracy, while simultaneously reducing the risk of overfitting and computational costs.

To clearly reflect the criteria of feature selection, we have added some descriptions in relevant section.

►     Lines 208-215 of section "2.5.3 Automatic optimization of XGBoost model":

"The initial input features (Table 1) are optimized *through a feature selection step, where MCV serves as the selection criterion. The selection is implemented by recursively removing the feature with the least importance, and reassessing the MCV based on cross-validation (Akhtar et al., 2019). Initially, an XGBoost model is trained with all features, and the importance of each feature is assessed based on its contribution to the model accuracy. The feature with the least importance is removed and the XGBoost model is retrained using the remaining features. The feature importance and MCV are updated accordingly and another feature is removed. This iterative process continues until the optimal number of features is identified, corresponding to the highest MCV achieved during the process.* The overall flowchart of the proposed spatiotemporally *separated* source reconstruction model is shown in Fig. S1."

***(1.2) Impact of varying combinations of observation sites***

We agree that it is important to discuss the impact of varying combinations of observation sites. Briefly speaking, the selection of representative sites is more important for model performance than increasing the number of sites. In this study, we have demonstrated this impact through sensitivity studies with respect to both the number and combination of observation sites.

1)    The number of observation sites

Additional observation sites can better capture environmental variability and impose stronger constraints on the estimation, leading to more robust results. However, the usage of all observation sites may cause overfitting of the XGBoost model and reduce the prediction accuracy of the trained model. Our sensitivity study (Fig. 12) also reveals that locating the source using all observation sites does not achieve the lowest error level, though the error level remains low.

2)    The position of observation sites

Observation sites located at appropriate position can capture environmental variability and provide adequate information for locating the source. Utilizing only these representative sites can alleviate overfitting and enhance the prediction accuracy of the XGBoost model. The sensitivity study demonstrates that the lowest error levels are achieved by a subset of sites, i.e. Site ABD on Oct. 3 and Site BD on Oct. 4. For Oct.3, multi-site estimations with Site B always produce lower error levels, and single-site estimation using Site B also achieves high accuracy. For Oct.4, multi-site estimations with Site BD always achieve relatively low error levels. These results prove the importance of representative sites in source location estimation. In addition, the representative sites (Site B for Oct. 3 and Site BD for Oct. 4) are consistent with the feature selection results in Fig. 4, preliminarily indicating the potential of feature selection to identify representative sites.

To highlight the impact of varying combinations of observation sites, we have added some descriptions in relevant section.

►     Lines 534-546 of section "3.4.4 Sensitivity to the number and combination of observation sites":

"*Figure 13* compares the results obtained with different numbers and combinations of observation sites. The results indicate that the *source location* error may be more sensitive to the position of the observation site than to the number of sites included. *The error level of all-site estimations is relatively low for both days, indicating that increasing the number of observation sites better constrains the solution and help improve the robustness of the model. However, the lowest error levels are achieved by a subset of sites, i.e. Site ABD on Oct. 3 and Site BD on Oct. 4. This is possibly because including all observation sites may cause overfitting and reduce the prediction accuracy. This overfitting can be alleviated by using only representative sites at appropriate position, which capture the environmental variability and*

*provide clear information for locating the source. For Oct.3, multi-site estimations with Site B always produce low error levels, and single-site estimation using Site B also achieves high accuracy. For Oct.4, multi-site estimations with Site BD always achieve relatively low error levels. These results demonstrate the importance of using representative sites for source location estimation. The representative sites (Site B for Oct. 3 and Site BD for Oct. 4) are consistent with the importance calculated in the feature selection step (Fig. 4), preliminarily indicating the potential for feature selection to identify representative sites. In addition, feature selection reduces the mean error level in most cases.*"

**(2) More validation of the method:**

We acknowledge the importance of validating our method against diverse scenarios and weather conditions to assess its robustness and practical applicability.

To address your concern, we have incorporated an additional validation case based on the first release of the European Tracer Experiment (ETEX-1) (Nodop et al., 1998), involving a different type of releases (continental-scale) and more complex meteorological conditions (temporally and spatially varying), to thoroughly assess the method's efficiency and adaptability. During ETEX-1, a total of 340 kg of perfluoromethylcyclohexane (PMCH) was released continuously into the atmosphere from 23 October 1994 at 16:00:00 UTC and 24 October at 1994 03:50:00 UTC. Assuming the release could have occurred between 23 October at 1994 00:00:00 UTC and 28 October 1994 at 00:00:00 UTC, it is viewed as a temporally-varying release, with a release rate of zero outside the actual release window. A total of 3104 available observations (3-h-averaged concentrations) were collected at 168 ground sites. As shown in Fig. 2b, we choose two groups of observation sites: the first comprises four sites (i.e. B05, D10, D16, F02) randomly selected from the sites within the sample zone (Group1, with a total of 92 available observations), and the second involves four sites (i.e. CR02, D15, DK08, S09) randomly selected from the sites beyond the sample zone boundaries (Group2, with a total of 90 available observations).

For the continental-scale ETEX-1 experiment, the proposed method still achieves the lowest source location errors among all methods, which are below 10 km and 20 km (less than the grid size of 0.25°×0.25°) for Group1 and Group2, respectively. Regarding the results of the uncertainty analysis, the mean relative source location errors are 2.42% and 4.97% for Group 1 and Group 2, respectively, lower than the correlation-based and Bayesian methods. The proposed method provides highly accurate mean estimates of release rate for both groups after feature selection, although with a large uncertainty range. These results demonstrate that spatiotemporally separated source reconstruction is feasible and achieves satisfactory accuracy in multi-scale release scenarios, thereby providing a promising framework for reconstructing atmospheric radionuclide releases.

An overview of the ETEX-1 experiment and corresponding source reconstruction results are provided below. Detailed results and discussions have been included in the revised manuscript.

► Lines 15-20 of section "Abstract":

[revised manuscript text omitted]

"

"**Table S4.** *Hyperparameter optimization results of all 50 runs in Group 1 of ETEX-1 experiment.*

| Run | Hyperparameters | | | | | | | |
|---|---|---|---|---|---|---|---|---|
| | *max_depth* | *learning_rate* | *n_estimators* | *min_child_weight* | *subsample* | *colsample_bytree* | *reg_lambda* | *gamma* |
| 1 | 3 | 0.07742 | 63 | 9 | 0.55953 | 0.44389 | 0.35824 | 0.81392 |
| 2 | 4 | 0.05011 | 142 | 3 | 0.70683 | 0.23027 | 4.89201 | 0.98193 |
| 3 | 3 | 0.06110 | 99 | 2 | 0.93364 | 0.37012 | 2.20274 | 0.76370 |
| 4 | 7 | 0.06117 | 64 | 10 | 0.55117 | 0.24878 | 2.34822 | 0.07471 |
| 5 | 3 | 0.07906 | 76 | 9 | 0.71631 | 0.38280 | 1.84573 | 0.22755 |
| 6 | 3 | 0.05067 | 116 | 4 | 0.70755 | 0.37613 | 1.50917 | 0.85795 |
| 7 | 3 | 0.06296 | 86 | 9 | 0.76834 | 0.34797 | 1.04674 | 0.44097 |
| 8 | 3 | 0.05342 | 106 | 5 | 0.53520 | 0.18199 | 4.18213 | 0.89476 |
| 9 | 3 | 0.07584 | 76 | 4 | 0.95540 | 0.53685 | 1.30937 | 0.49309 |
| 10 | 3 | 0.07213 | 84 | 5 | 0.98527 | 0.41048 | 2.61014 | 0.06896 |
| 11 | 3 | 0.05907 | 115 | 6 | 0.74495 | 0.37051 | 1.96059 | 0.27702 |
| 12 | 3 | 0.09471 | 76 | 3 | 0.91290 | 0.51041 | 0.38558 | 0.05232 |
| 13 | 3 | 0.07018 | 103 | 3 | 0.87242 | 0.35732 | 2.77176 | 0.16073 |
| 14 | 3 | 0.07072 | 95 | 10 | 0.98317 | 0.34966 | 4.67025 | 0.95006 |
| 15 | 3 | 0.08357 | 66 | 2 | 0.80913 | 0.37858 | 2.74202 | 0.05494 |
| 16 | 3 | 0.05001 | 121 | 3 | 0.52214 | 0.27101 | 4.30584 | 0.30632 |
| 17 | 3 | 0.09354 | 50 | 6 | 0.64736 | 0.62473 | 2.55863 | 0.35745 |
| 18 | 3 | 0.05486 | 134 | 8 | 0.66206 | 0.71278 | 0.80280 | 0.97413 |
| 19 | 3 | 0.07556 | 102 | 5 | 0.70927 | 0.34789 | 3.12167 | 0.99997 |
| 20 | 3 | 0.07479 | 52 | 9 | 0.79240 | 0.56720 | 1.02323 | 0.32951 |
| 21 | 3 | 0.05518 | 78 | 10 | 0.66309 | 0.99871 | 3.14571 | 0.84078 |
| 22 | 3 | 0.05139 | 111 | 6 | 0.73839 | 0.42603 | 2.49218 | 0.87318 |
| 23 | 3 | 0.10952 | 50 | 5 | 0.97390 | 0.22350 | 0.88047 | 0.35097 |

| | | | | | | | | |
|---|---|---|---|---|---|---|---|---|
| 24 | 3 | 0.06860 | 93 | 8 | 0.66447 | 0.33031 | 3.95098 | 0.67626 |
| 25 | 3 | 0.08670 | 54 | 5 | 0.79857 | 0.39303 | 3.19098 | 0.54197 |
| 26 | 3 | 0.08125 | 67 | 8 | 0.94963 | 0.33151 | 4.40350 | 0.06507 |
| 27 | 3 | 0.08396 | 50 | 2 | 0.93428 | 0.37792 | 3.77359 | 0.13881 |
| 28 | 3 | 0.05418 | 95 | 9 | 0.99598 | 0.25227 | 1.60204 | 0.38791 |
| 29 | 3 | 0.06935 | 89 | 8 | 0.90482 | 0.35876 | 4.15848 | 0.85423 |
| 30 | 3 | 0.06319 | 76 | 2 | 0.91583 | 0.43665 | 3.35600 | 0.88327 |
| 31 | 3 | 0.09009 | 50 | 4 | 0.94183 | 0.48645 | 4.03998 | 0.17582 |
| 32 | 3 | 0.06213 | 148 | 8 | 0.69235 | 0.34063 | 1.38470 | 0.71691 |
| 33 | 3 | 0.05735 | 74 | 10 | 0.83704 | 0.36175 | 2.28311 | 0.93893 |
| 34 | 3 | 0.06437 | 72 | 3 | 0.98748 | 0.31363 | 1.63480 | 0.22685 |
| 35 | 3 | 0.06022 | 60 | 6 | 0.66204 | 0.69317 | 1.21692 | 0.30900 |
| 36 | 3 | 0.09555 | 53 | 5 | 0.80980 | 0.46487 | 1.90000 | 0.60232 |
| 37 | 3 | 0.08434 | 50 | 8 | 0.52774 | 0.26641 | 0.48391 | 0.31574 |
| 38 | 3 | 0.07105 | 51 | 5 | 0.96131 | 0.63725 | 2.01205 | 0.60509 |
| 39 | 4 | 0.05005 | 62 | 10 | 0.77964 | 0.22026 | 3.55884 | 0.74839 |
| 40 | 4 | 0.07553 | 74 | 2 | 0.99323 | 0.36292 | 2.61782 | 0.17595 |
| 41 | 3 | 0.05239 | 77 | 7 | 0.50028 | 0.95751 | 2.41469 | 0.72211 |
| 42 | 3 | 0.08421 | 51 | 5 | 0.99977 | 0.45864 | 2.15063 | 0.54258 |
| 43 | 3 | 0.05414 | 82 | 8 | 0.62226 | 0.76122 | 2.83002 | 0.53414 |
| 44 | 3 | 0.05259 | 111 | 4 | 0.93432 | 0.32197 | 2.04760 | 0.44156 |
| 45 | 3 | 0.09454 | 60 | 4 | 0.72047 | 0.23879 | 4.65624 | 0.75740 |
| 46 | 3 | 0.05013 | 93 | 4 | 0.58250 | 0.40493 | 2.11383 | 0.47864 |
| 47 | 3 | 0.11330 | 50 | 2 | 0.71413 | 0.69524 | 3.50503 | 0.16269 |
| 48 | 3 | 0.08314 | 50 | 10 | 0.96691 | 0.50529 | 2.97909 | 0.95771 |
| 49 | 6 | 0.05978 | 107 | 7 | 0.63666 | 0.20488 | 0.61715 | 0.79254 |
| 50 | 3 | 0.07770 | 71 | 3 | 0.98943 | 0.58108 | 1.17867 | 0.22360 |

**Table S5.** *Hyperparameter optimization results of all 50 runs in Group 2 of ETEX-1 experiment.*

| Run | Hyperparameters | | | | | | | |
| --- | --- | --- | --- | --- | --- | --- | --- | --- |
| | max_depth | learning_rate | n_estimators | min_child_weight | subsample | colsample_bytree | reg_lambda | gamma |
| 1 | 8 | 0.17252 | 127 | 7 | 0.58651 | 0.60647 | 4.02984 | 0.60724 |
| 2 | 8 | 0.22988 | 68 | 9 | 0.54811 | 0.91321 | 2.29180 | 0.12852 |
| 3 | 8 | 0.10002 | 213 | 10 | 0.75238 | 0.64053 | 4.99653 | 0.10778 |
| 4 | 4 | 0.12366 | 147 | 5 | 0.87463 | 0.83119 | 0.63085 | 0.26995 |
| 5 | 5 | 0.23441 | 297 | 7 | 0.97266 | 0.36675 | 0.34489 | 0.99034 |
| 6 | 4 | 0.20533 | 155 | 4 | 0.63974 | 0.56526 | 2.63317 | 0.38177 |
| 7 | 5 | 0.10641 | 151 | 3 | 0.76094 | 0.79390 | 0.98025 | 0.96160 |
| 8 | 4 | 0.17290 | 222 | 4 | 0.55589 | 0.76284 | 1.62191 | 0.47379 |
| 9 | 4 | 0.05855 | 160 | 3 | 0.88818 | 0.79781 | 4.98019 | 0.84983 |
| 10 | 4 | 0.11741 | 184 | 5 | 0.79714 | 0.83203 | 4.43324 | 0.62777 |
| 11 | 4 | 0.16266 | 136 | 2 | 0.75690 | 0.93818 | 3.30854 | 0.55481 |
| 12 | 5 | 0.23134 | 144 | 2 | 0.58373 | 0.60423 | 2.76711 | 0.16986 |
| 13 | 8 | 0.27891 | 193 | 7 | 0.64977 | 0.89059 | 3.88204 | 0.23152 |
| 14 | 7 | 0.18603 | 245 | 8 | 0.77290 | 0.78709 | 3.45149 | 0.01806 |
| 15 | 5 | 0.16915 | 268 | 10 | 0.62385 | 0.49651 | 2.30355 | 0.27120 |
| 16 | 4 | 0.20217 | 64 | 2 | 0.92784 | 0.78470 | 0.94699 | 0.93657 |
| 17 | 5 | 0.10871 | 181 | 9 | 0.70489 | 0.84917 | 4.43678 | 0.07228 |
| 18 | 5 | 0.07394 | 297 | 8 | 0.87839 | 0.62200 | 3.24008 | 0.11160 |
| 19 | 6 | 0.20293 | 216 | 9 | 0.66381 | 0.89210 | 4.08151 | 0.60613 |
| 20 | 7 | 0.20570 | 158 | 4 | 0.50653 | 0.86393 | 3.36667 | 0.79227 |
| 21 | 5 | 0.20083 | 88 | 7 | 0.57460 | 0.62410 | 1.26707 | 0.17321 |
| 22 | 4 | 0.27072 | 50 | 4 | 0.85604 | 0.86560 | 0.16264 | 0.44052 |
| 23 | 8 | 0.15380 | 86 | 5 | 0.67811 | 0.74505 | 4.54334 | 0.93377 |
| 24 | 4 | 0.16205 | 183 | 6 | 0.59364 | 0.93969 | 1.05664 | 0.40669 |
| 25 | 6 | 0.14171 | 288 | 6 | 0.75389 | 0.85527 | 4.65363 | 0.50557 |
| 26 | 7 | 0.21287 | 253 | 9 | 0.59311 | 0.65113 | 2.79234 | 0.83703 |

| | | | | | | | | |
|---|---|---|---|---|---|---|---|---|
| 27 | 4 | 0.15371 | 247 | 5 | 0.77890 | 0.52357 | 4.81584 | 0.67752 |
| 28 | 5 | 0.11665 | 135 | 5 | 0.79729 | 0.86017 | 4.26743 | 0.12912 |
| 29 | 4 | 0.08378 | 192 | 4 | 0.52749 | 0.79980 | 2.64816 | 0.57092 |
| 30 | 6 | 0.13030 | 210 | 3 | 0.50209 | 0.61548 | 3.80894 | 0.64347 |
| 31 | 7 | 0.24148 | 173 | 10 | 0.64711 | 0.79358 | 2.66441 | 0.23023 |
| 32 | 6 | 0.09301 | 204 | 8 | 0.69879 | 0.97301 | 4.67770 | 0.36945 |
| 33 | 5 | 0.12318 | 283 | 6 | 0.93580 | 0.70267 | 2.23369 | 0.17565 |
| 34 | 4 | 0.23289 | 227 | 7 | 0.60924 | 0.76662 | 2.97809 | 0.22066 |
| 35 | 6 | 0.21219 | 162 | 3 | 0.54969 | 0.50796 | 4.01790 | 0.10632 |
| 36 | 5 | 0.10657 | 148 | 3 | 0.77407 | 0.84022 | 4.19435 | 0.53237 |
| 37 | 4 | 0.14220 | 169 | 4 | 0.69411 | 0.90516 | 2.46148 | 0.83182 |
| 38 | 6 | 0.11224 | 239 | 4 | 0.64335 | 0.91879 | 1.53421 | 0.43750 |
| 39 | 5 | 0.08990 | 98 | 5 | 0.83843 | 0.99546 | 3.80815 | 0.86071 |
| 40 | 6 | 0.19006 | 130 | 4 | 0.95749 | 0.88483 | 3.68950 | 0.17261 |
| 41 | 5 | 0.21434 | 93 | 6 | 0.80593 | 0.97025 | 2.23769 | 0.40479 |
| 42 | 7 | 0.07619 | 234 | 6 | 0.62146 | 0.77954 | 4.62217 | 0.85628 |
| 43 | 4 | 0.17377 | 273 | 6 | 0.85218 | 0.79578 | 3.43808 | 0.62076 |
| 44 | 4 | 0.18522 | 135 | 4 | 0.82615 | 0.63563 | 4.24215 | 0.56409 |
| 45 | 4 | 0.14993 | 152 | 8 | 0.60441 | 0.80580 | 2.50467 | 0.09351 |
| 46 | 5 | 0.15229 | 164 | 7 | 0.94667 | 0.83661 | 3.59476 | 0.15891 |
| 47 | 5 | 0.15393 | 116 | 9 | 0.90651 | 0.85377 | 4.60433 | 0.89894 |
| 48 | 5 | 0.22272 | 290 | 9 | 0.86799 | 0.85502 | 4.52637 | 0.79836 |
| 49 | 5 | 0.11275 | 91 | 4 | 0.72730 | 0.75528 | 3.72672 | 0.17298 |
| 50 | 4 | 0.13702 | 299 | 7 | 0.95702 | 0.91622 | 2.93120 | 0.22371 |

"

**Specific comments**

**Comment#1:**

L215 - Figure 2: The axes represent distances and should therefore have identical scales for clarity and accuracy.

**Response to comment#1:**

We appreciate your attention on the different scales of axes on Figure 2. We have revised the figure to ensure that both axes represent distances with identical scales.

► Figure 2 of section "2.6.1 Field experiments" (Lines 240-246):

[Figure]

"

**Figure 2.** Release location and observation sites of two field experiments. *(a) SCK-CEN [41]Ar experiment. The map was created based on the relative positions of the release source and observation sites (Drews et al., 2002). The coordinates of the sample border are (500 m, −200 m) and (1180 m, 580 m) on Oct. 3, and (450 m, 10 m) and (850 m, 450 m) on Oct. 4. This figure was plotted using MATLAB 2016b, rather than created by a map provider; (b) ETEX-1 experiment. The map was created based on the real longitudes and latitudes of the release source and observation sites (Nodop et al.,*

*1998). The coordinates of the sample border are (10°W, 40°N) and (10°E, 60°N). This figure was plotted using the cartopy function of Python, rather than created by a map provider.*"

**Comment#2:**

L222 - Consideration of vertical information could provide a more comprehensive understanding of the dispersion patterns. How does the model account for vertical dispersion?

**Response to comment#2:**

Thank you for your constructive comment. We agree that incorporating vertical information can aid in understanding the dispersion patterns. In the SCK-CEN $^{41}$Ar experiment, the $^{41}$Ar was emitted from a 60-m stack, while the ground-level fluence rates were collected by NaI (Tl) gamma detectors. Due to the lack of vertical observations in this experiment, the vertical dispersion has not been discussed in the manuscript.

The RIMPUFF model is a gaussian puff model that uses the diffusion coefficient in vertical direction to describe the vertical dispersion of each puff. In this study, the Karlsruhe-Jülich diffusion coefficients were used to calculate the vertical dispersion, which has been validated for the SCK-CEN $^{41}$Ar experiment and has shown good accuracy (Li et al., 2019).

In the future, we will try to incorporate the vertical dispersion information into the source parameters.

To ensure clarity, we have added some descriptions of vertical information in the revised manuscript.

► Lines 224-225 of section "2.6.1 SCK-CEN $^{41}$Ar field experiment":

"The 60-s-average *ground-level* fluence rates were continuously collected by an array of NaI (Tl) gamma detectors, with different observation sites used on the two days."

► Line 251 of section "2.6.2 Simulation settings of atmospheric dispersion model":

"*The release height of $^{41}$Ar was assumed to be 60 m.*"

The added ETEX-1 experiment also consider vertical information.

► Lines 231-232 of section "2.6.1 Field experiments":

"*A total of 3104 available observations (3-h-averaged concentrations) were collected at 168 ground sites.*"

► Lines 263-265 section "2.6.2 Simulation settings of atmospheric dispersion model":

"*To rapidly establish the relationship between the varying source locations and the observations, 182 backward simulations were performed using FLEXPART with a time interval of 3 h, grid size of 0.25°×0.25°, and 8 vertical levels (from 100–50000 m). Only the lowest model output layer was used for source reconstruction.*"

**Comment#3:**

L302 - Figure 3: To ensure clarity and accuracy in data representation, the scales on the vertical and horizontal axes must be consistent.

**Response to comment#3:**

We appreciate your attention on the different scales on axes on Figure 3. We have carefully adjusted Figure 3 so that the scales on both the vertical and horizontal axes are now consistent.

► Figure 3 of section "3.1 Filtering performance" (Lines 344-347):

[Figure]

**Figure 3.** Scatter plots of the original (yellow squares) and *filtered* (green squares) observations versus the constant-release simulation results. *SCK-CEN [41]Ar experiment: (a) Oct. 3 (synthetic observations); (b) Oct. 4 (synthetic observations); (c) Oct. 3 (real observations); (d) Oct. 4 (real observations); ETEX-1 experiment: (e) Group 1 (real observations); (f) Group 2 (real observations).*"

**Comment#4:**

L340 - The capability to estimate with greater accuracy than the grid size warrants a discussion. What implications does this have for the model's precision and its practical significance?

**Response to comment#4:**

Thank you for your insightful query. The ability to estimate source locations with accuracy surpassing the grid size can be attributed to the strong fitting capability of the optimized XGBoost model (Chen and Guestrin, 2016; Grinsztajn et al., 2022), which excels in interpolating within the grid size and extrapolating beyond the source location samples. As discussed in our response to the General comments, ETEX-1 experiment also achieved source location accuracy beyond the grid size (Fig. 5c and 5d, the grid size is 0.25°×0.25°), suggesting that the phenomenon is not merely coincidental. In addition, previous studies also achieved similar source location accuracy using traditional methods (Lucas et al., 2017; Tichý et al., 2017). However, this ability, although inherent, does not uniformly manifest across all optimized XGBoost models, as external factors like observation noises and meteorological data inaccuracies can also impact the accuracy of source location estimation. The uncertainty analysis in Sect. 3.3.3 has demonstrated that the source location estimates tend to cluster within several grids surrounding the true source, which is more reasonable and practical in real-world scenarios. Detailed discussions are as follows:

(1) ***Enhanced accuracy through XGBoost***: The high accuracy in locating the source is directly achieved by the XGBoost model, since it establishes the complex nonlinear relationships between the input features and the source location. Utilizing automatic optimization techniques (detailed in Sect. 2.5.3), 8 main hyperparameters of XGBoost and 24 observation series features are finely tuned to achieve an optimized model. This optimization not only mitigates the risk of overfitting but also enhances the model's ability for interpolation within the grid size and extrapolation beyond the source location samples.

(2) ***Validation on ETEX-1 experiment***: The proposed method has been validated through ETEX-1 experiment, as discussed in our response to the General comments. Compared to SCK-CEN $^{41}$Ar experiment, ETEX-1 involves a different type of releases (continental-scale) and more complex meteorological conditions (temporally and spatially varying). As shown in Fig. 5(c) and 5(d), this experiment also achieved source location accuracy beyond the grid size (0.25°×0.25°), suggesting that the phenomenon is not merely coincidental. Providing that the XGBoost model is effectively optimized and the observations are reliable, the model has ability to achieve high accuracy.

[Figure]

**Figure 5.** *Source location estimation* results *of SCK-CEN [41]Ar experiment:* (a) Oct. 3; (b) Oct. 4*; and ETEX-1 experiment: (c) Group 1; (d) Group 2. A detailed enlargement of the region around (2.5°W, 47.5°N) to (1.5°W, 48.5°N) is shown in the bottom right corner in (c) and (d) to highlight the source location estimation results of the proposed method.* The yellow dots denote the maximum correlation points, which are the results of the correlation-based method. The green and red stars represent the results based on XGBoost before and after feature selection, respectively. The cyan diamonds represent the results based on the Bayesian method.

(3) ***Uncertainty analysis of XGBoost hyperparameters***: An uncertainty analysis of the XGBoost hyperparameters (Fig. 7) has revealed that not all source location estimates achieve greater accuracy than the grid size. Instead, source location estimates tend to cluster within several grids surrounding the true source. This phenomenon highlights the

practical significance of the proposed method.

[Figure]

**Figure 7.** *Spatial distribution of 50 source location estimates of SCK-CEN $^{41}$Ar experiment: (a) Oct. 3; (b) Oct. 4; and ETEX-1 experiment: (c) Group 1; (d) Group 2. Each circle denotes an individual estimate as detailed in Sect. 2.8.5, with colour variations indicating the respective method employed. Histograms along the axes represent the frequency distribution of the estimates along the respective axis.*

To avoid confusion, we have added some discussions in the revised manuscript to explain the greater accuracy than the grid size. The revised manuscript has also included the reconstruction results of ETEX-1 experiment (see our response to the General comments), which will further prove the model's ability.

► Lines 391-397 of section "3.3.1 Source locations":

"The estimates without feature selection are only 10.65 m (Oct. 3) and 20.62 m (Oct. 4) away from the true locations. Feature selection further reduces these errors to 6.19 m (Oct. 3, *a relative error of 0.60%*) and 4.52 m (Oct. 4, *a relative error of 0.80%*), which are below the grid size (10 m×10 m) of the *atmospheric dispersion* simulation. *The ability to*

*estimate the source locations with accuracy surpassing the grid size can be attributed to the strong fitting capability of the optimized XGBoost model (Chen and Guestrin, 2016; Grinsztajn et al., 2022). However, this capability, although inherent, is not present across all optimized XGBoost models, as external factors such as observation noises and meteorological data inaccuracies can also impact the accuracy of source location estimation."*

**Comment#5:**

L346 - Figure 5: As these axes represent distances, maintaining identical scales on both axes is crucial for accurate data interpretation.

**Response to comment#5:**

    Thank you for pointing out this issue. We have revised Figure 5 to ensure that both the horizontal and vertical axes have the same scale.

►     Figure 5 of section "3.3.1 Source locations" (Lines 407-412):

[Figure]

"

**Figure 5.** *Source location estimation* results *of SCK-CEN [41]Ar experiment:* (a) Oct. 3; (b) Oct. 4*; and ETEX-1 experiment: (c) Group 1; (d) Group 2. A detailed enlargement of the region around (2.5°W, 47.5°N) to (1.5°W, 48.5°N) is shown in the bottom right corner in (c) and (d) to highlight the source location estimation results of the proposed method.* The yellow dots denote the maximum correlation points, which are the results of the correlation-based method. The green and red stars represent the results based on XGBoost before and after feature selection, respectively. The cyan diamonds represent the results based on the Bayesian method."

**Comment#6:**

L374 - Figure 7: Given that both axes represent distances, their scales should be identical. The complexity of the graphs necessitates a detailed explanation within the figure caption to aid in interpretation.

**Response to comment#6:**

Thank you for your constructive comment. We have revised Figure 7 to ensure that both axes are now on identical scales and expanded the figure caption to include a detailed explanation of the graph's components.

► Figure 7 of section "3.3.3 Uncertainty range" (Lines 451-454):

[Figure]

"

**Figure 7.** *Spatial distribution of 50 source location estimates of SCK-CEN $^{41}$Ar experiment: (a) Oct. 3; (b) Oct. 4; and ETEX-1 experiment: (c) Group 1; (d) Group 2. Each circle denotes an individual estimate as detailed in Sect. 2.8.5, with colour variations indicating the respective method employed. Histograms along the axes represent the frequency distribution of the estimates along the respective axis.*"

**Thanks again for such a thorough review!**

---

## Author Response (AR2)

Dear Referee,

Thank you very much for taking the time to review our revised manuscript (GMD-2023-173) once again and for providing further valuable feedback. Your suggestive comments have played a crucial role in enhancing the quality and clarity of our paper. We have made some modifications to the manuscript, and the responses are listed below. To guide the review process, comments from the referee and original texts in the manuscript are presented in black, our responses are in blue, and any text modifications made to the manuscript are highlighted in red italics. The line numbers mentioned in this response correspond to those in the revised manuscript. Links are provided below for easy navigation in the document.

**General comments**

**Minor points**

We are looking forward to your reply.

Best regards,

Yours sincerely

Sheng Fang

**General comments**

The authors addressed most of the concerns raised to the earlier version of the manuscript. I appreciate their efforts in making significant changes and adding new contents to the paper. The responses are mostly satisfying.

I would recommend publication after the authors could further address some of the minor issues listed below.

**Response to general comments:**

Thank you for your valuable feedback and suggestive comments on our manuscript. Your appreciation of our efforts is highly encouraging. We are particularly grateful to hear that our revisions meet most of your expectations. To refine the manuscript, we have carefully reviewed the minor issues you have listed and have addressed them in the revised manuscript.

**Minor points**

**Comment#1:**

Abstract, line 17: The definition of the relative source location errors is not universal. Only presenting the values without explaining how the relative errors are defined is not informative.

**Response to comment#1:**

Thank you for your suggestive comments regarding the presentation of relative source location errors in the abstract. I agree that using the term "relative source location errors" may lead to confusion, since the definition is not universally established. To address this issue, I have replaced the results of the relative source location errors with those of the absolute source location errors.

▶ Line 15-17 of section "Abstract":

"This method is validated against the local-scale SCK-CEN $^{41}$Ar field experiment and the first release of the continental-scale European Tracer Experiment, for which the lowest source location errors are *4.52 m* and *5.19 km*, respectively."

**Comment#2:**

Lines 69-74: Inaccuracy of the meteorological fields is another major factor that makes the real-world source reconstruction problems challenging. This needs to be included.

**Response to comment#2:**

Thank you for your suggestive comments. I agree that the inaccuracies of meteorological fields should be highlighted to provide a more comprehensive understanding of the challenges in source reconstruction. These inaccuracies can intensify the challenges of source reconstruction, due to the interaction between the time-varying release characteristics and non-stationary meteorological fields. We have added this point in the revised manuscript.

▶ Lines 69-74 of section "1. Introduction":

"However, previous studies have also demonstrated that real-world applications may be much more challenging, (Kovalets et al., 2020; Tomas et al., 2021; Andronopoulos and Kovalets, 2021; Becker et al., 2007) because the release usually exhibits temporal variations and may experience non-stationary meteorological fields. *In addition, inaccurate calculation of the meteorological field input can further intensify these challenges.* The interaction between the time-varying release characteristics and non-stationary meteorological fields is neglected in the instantaneous-release and constant-release assumptions, leading to inaccurate reconstruction."

**Comment#3:**

Line 101, "observation vector composed of observations at $N$ sequential time steps": N should be the number of observations, not the number of sequential time steps. There could be multiple observations at a single time.

**Response to comment#3:**

We apologize for the incorrect definition of "observation vector". You are correct in pointing out that N should represent the number of observations rather than the number of sequential time steps. To avoid confusion, we have modified it in the revised manuscript.

► Line 101-103 section "2.1 Source reconstruction models":

"where $\boldsymbol{\mu} = [\mu_1, \mu_2, \cdots, \mu_N]^T \in \mathbb{R}^N$ is an observation vector composed of *N observations*, the function **F** maps the source parameters to the observations, i.e. an atmospheric dispersion model, **r** refers to the source location, $\mathbf{q} \in \mathbb{R}^S$ is the temporally varying release rate, and $\boldsymbol{\varepsilon} \in \mathbb{R}^N$ is a vector containing both model and measurement errors."

"

**Comment#4:**

Line 552, "... and the 50th error level is lower than ...": Please add "percentile" after "50th".

**Response to comment#4:**

We appreciate your helpful suggestion regarding the clarification needed on Line 552. To avoid confusion, we have added "percentile" after "50th" in the revised manuscript.

► Lines 551-552 of section "3.4.5 Sensitivity to the meteorological errors":

"For Oct. 3, the estimates generally present a low error level (generally below 10%), and the 50th *percentile* error level is lower than the error of the unperturbed results (4.68%)."

**Thanks again for such a thorough review!**